# An evaluation of global organic aerosol schemes using airborne observations

Sidhant J. Pai[1], Colette L. Heald[1,2], Jeffrey R. Pierce[3], Salvatore C. Farina[3], Eloise A. Marais[4], Jose L. Jimenez[5], Pedro Campuzano-Jost[5], Benjamin A. Nault[5], Ann M. Middlebrook[6], Hugh Coe[7], John E. Shilling[8], Roya Bahreini[9], Justin H. Dingle[9], Kennedy Vu[9]

[1]Department of Civil and Environmental Engineering, Massachusetts Institute of Technology, Cambridge, MA, 02139, USA
[2]Department of Earth, Atmospheric and Planetary Sciences, Massachusetts Institute of Technology, Cambridge, MA, 02139, USA
[3]Colorado State University, Department of Atmospheric Science, Fort Collins, CO, 80523, USA
[4]Department of Physics and Astronomy, University of Leicester, Leicester, LE1 7RH, UK.
[5]Department of Chemistry, and Cooperative Institute for Research in Environmental Sciences (CIRES), University of Colorado, Boulder, CO, 80309, USA
[6]NOAA Earth System Research Laboratory (ESRL) Chemical Sciences Division, 325 Broadway, Boulder, CO 80305, USA
[7]Centre for Atmospheric Science, School of Earth and Environmental Science, University of Manchester, Manchester, M13 9PL, UK
[8]Atmospheric Sciences and Global Change Division, Pacific Northwest National Laboratory, Richland, Washington, USA
[9]Department of Environmental Sciences, University of California, Riverside, CA 92521, USA

*Correspondence to*: Sidhant J. Pai (sidhantp@mit.edu) and Colette L. Heald (heald@mit.edu)

**Abstract.** Chemical transport models have historically struggled to accurately simulate the magnitude and variability of observed organic aerosol (OA), with previous studies demonstrating that models significantly underestimate observed concentrations in the troposphere. In this study, we explore two different model OA schemes within the standard GEOS-Chem chemical transport model and evaluate the simulations against a suite of 15 globally-distributed airborne campaigns from 2008-2017, primarily in the spring and summer seasons. These include the ATom, KORUS-AQ, GoAmazon, FRAPPE, SEAC4RS, SENEX, DC3, CalNex, OP3, EUCAARI, ARCTAS and ARCPAC campaigns and provide broad coverage over a diverse set of atmospheric-composition regimes – anthropogenic, biogenic, pyrogenic and remote. The schemes include significant differences in their treatment of the primary and secondary components of OA – a 'simple scheme' that models primary OA (POA) as non-volatile and takes a fixed-yield approach to secondary OA (SOA) formation, and a 'complex scheme' that simulates POA as semi-volatile and uses a more sophisticated volatility basis set approach for non-isoprene SOA, with an explicit aqueous uptake mechanism to model isoprene SOA. Despite these substantial differences, both the simple and complex schemes perform comparably across the aggregate dataset in their ability to capture the observed variability (with an $R^2$ of 0.41 and 0.44 respectively). The simple scheme displays greater skill in minimizing the overall model-bias (with an NMB of 0.04, compared to 0.30 for the complex scheme). Across both schemes, the model skill in reproducing observed OA is superior to previous model evaluations and approaches the fidelity of the sulfate simulation within the GEOS-Chem model. However, there are significant differences in model performance across different chemical source regimes,

classified here into 7 categories. Higher-resolution nested regional simulations indicate that model resolution is an important factor in capturing variability in highly-localized campaigns, while also demonstrating the importance of well-constrained emissions inventories and local meteorology, particularly over Asia. Our analysis suggests that a semi-volatile treatment of POA is superior to a non-volatile treatment. It is also likely that the complex scheme parameterization overestimates biogenic SOA at the global scale. While this study identifies factors within the SOA

schemes that likely contribute to OA model bias (such as a strong dependency of the bias in the complex scheme on relative humidity and sulfate concentrations), comparisons with the skill of the sulfate aerosol scheme in GEOS-Chem indicate the importance of other drivers of bias such as emissions, transport, and deposition that are exogenous to the OA chemical scheme.

## 1. Introduction

Aerosols in the atmosphere have significant climate impacts through radiative scattering and cloud formation (IPCC, 2013; Ramanathan et al., 2001). Exposure to these particles is also detrimental to human health and is associated with over 4 million premature deaths per year world-wide (Pope and Dockery, 2006; Cohen et al., 2017). Organic aerosol (OA) often accounts for a large portion of the total fine aerosol burden (Jimenez et al., 2009), a fraction that has been increasing over time, particularly in regions where sulfur dioxide controls have reduced anthropogenic sources of

sulfate (Marais et al., 2017). Characterizing aerosol impacts on air quality and climate thus requires a comprehensive understanding of the lifecycle of organic aerosol in the atmosphere.

Organic aerosol that is emitted directly into the atmosphere from anthropogenic or natural sources is called primary organic aerosol (POA). A significant fraction of primary organic emissions have been shown to be semi-volatile, partitioning between the gas and particle phase depending on ambient temperature and background organic aerosol

concentration (Grieshop et al., 2009; Lipsky and Robinson, 2006; Robinson et al., 2007; Shrivastava et al., 2006). As these compounds are dispersed through the atmosphere, they are oxidized (in both gas and particle phase) and typically form lower volatility products. In addition to the primary component, organic aerosol is also generated dynamically in the atmosphere from volatile organic compound (VOC) and intermediate volatility organic compound (IVOC) precursors that are both anthropogenic (e.g. benzene, toluene, xylene) and biogenic (e.g. isoprene, monoterpenes,

sesquiterpenes). These gas-phase precursors undergo multi-phase, multi-generational oxidation processes that result in a complex array of semi-volatile species that partition into organic aerosol under conducive conditions. This class of aerosol products is called secondary organic aerosols (SOA). Both POA and SOA are important drivers of climate and air quality, often influencing regions far removed from their original source locations (Kanakidou et al., 2005).

Primary organic aerosol has traditionally been modeled as non-volatile (e.g. Chung & Seinfeld, 2002), but recent

studies have incorporated a semi-volatile treatment that allows the aerosol species to dynamically partition between the condensed-phase and gas-phase, while simultaneously undergoing gas-phase oxidation to form organic compounds of lower volatility (Donahue et al., 2006; Robinson et al., 2007; Huffman et al., 2009; Pye and Seinfeld, 2010). There has been a similar evolution in the methods to model the formation and chemical processing of SOA in the atmosphere. Initial global modeling efforts often simulated SOA as a species that is directly formed upon emission

of various precursors based on a fixed yield from laboratory or field studies (Chin et al., 2002; Kim et al., 2015; Pandis et al., 1992; Park et al., 2003). Many earth system models continue to use this simple approach (Tsigaridis et al., 2014). The two-product absorptive reversible partitioning scheme was then developed to account for the semi-volatile nature of SOA, using lumped oxidation products from precursor VOCs (Odum et al., 1996; Pankow, 1994). Advances in computational resources have enabled recent studies to more effectively capture the volatility-distribution of organics, using a volatility basis set (VBS) of volatility-resolved semi-volatile surrogates that absorptively partition based on dry ambient OA concentrations (Donahue et al., 2006; Pye et al., 2010). There have also been more explicit chemical treatments of organic aerosol formation, such as those involving the implementation of a master chemical mechanism coupled with equilibrium absorptive-partitioning and reactive surface uptake mechanisms (Li et al., 2015; Xia et al., 2008) or the explicit description of irreversible aqueous OA formation pathways (Fisher et al., 2016; Lin et al., 2012; Marais et al., 2016).

The wide range of VOC precursors, the complexities of the various formation pathways, and the limited laboratory constraints on these processes make accurately modeling OA concentrations highly challenging. Previous model studies have identified large underestimates in the simulated OA when compared to observations (e.g. Heald et al., 2011; Volkamer et al., 2006). Over the past decade, the treatment of organic aerosol in chemical transport models has grown in complexity with models showing improved regional skill at simulating OA over areas like the southeast US (Marais et al., 2016; Budisulistiorini et al., 2017). However, studies that have evaluated OA model simulations against globally distributed measurements have demonstrated a consistent model inability to capture the magnitude and variability of observed OA concentrations (Heald et al., 2011; Tsigaridis et al., 2014). In particular, the evaluation by Heald et al. (2011) that used a two-product OA scheme revealed significant deficiencies in model skill and suggested that the GEOS-Chem model underestimated both the sources and sinks of OA at the global scale. The complex nature of OA formation and loss mechanisms in the atmosphere has thus made it difficult to constrain global models using a bottom-up approach, particularly given the uncertainties inherent in the various emission inventories and chemical mechanisms. Here, we use a top-down approach, leveraging a suite of 15 aircraft campaigns to evaluate the two different organic aerosol schemes implemented within the standard GEOS-Chem chemical transport model in order to assess their relative strengths and weaknesses over a wide range of chemical and spatial regimes.

## 2. Model Description

We use the chemical transport model GEOS-Chem (www.geos-chem.org) to simulate organic aerosol mass concentrations along the flight tracks of a suite of airborne campaigns described in Sect. 3. In order to contrast the different approaches to modeling organic aerosol and its precursors in the atmosphere, we perform a series of simulations from 2008 to 2017 using two distinct model schemes that vary based on their treatment of organic aerosol (see Sect. 2.1 and Table S1).

These simulations were performed with the GEOS-Chem model version 12.1.1 (https://doi.org/10.5281/zenodo.2249246) with a horizontal resolution of 2° x 2.5° and 47 vertical hybrid-sigma levels that extend from the surface to the lower stratosphere. A series of nested simulations, over North America and Asia,

were performed at a higher spatial resolution of 0.5° x 0.625° using boundary conditions from the 2° x 2.5° global run. The model is driven by the MERRA-2 assimilated meteorological product from the NASA Global Modeling and Assimilation Office (GMAO) with a transport time-step of 10 minutes as recommended by Philip et al. (2016). The model includes a coupled treatment of $HO_x$-$NO_x$-VOC-$O_3$ chemistry (Mao et al., 2013; Travis et al., 2016; Miller et al., 2017) with integrated Cl-Br-I chemistry (Sherwen et al., 2016) and uses a bulk aerosol scheme with fixed log-

normal modes (Martin et al., 2003). GEOS-Chem simulates sulfate aerosol (Park, 2004), sea-salt (Jaeglé et al., 2011), black carbon (Park et al., 2003) and mineral dust (Fairlie et al., 2007; Ridley et al., 2012). Ammonium and nitrate thermodynamics are described using the ISORROPIA II model (Fountoukis and Nenes, 2007). Deposition losses are dictated by aerosol and gas dry deposition to surfaces based on a resistor-in-series scheme (Wesely, 1989; Zhang et al., 2001) and wet deposition from scavenging by rainfall and moist convective cloud updrafts (Amos et al., 2012;

Jacob et al., 2000; Liu et al., 2001). More details on the deposition schemes are provided in the SI.

### 2.1 Organic Aerosol Simulations

This study evaluates the two standard organic aerosol schemes within the GEOS-Chem model. The 'complex scheme' represents a more detailed, recently updated treatment of organic aerosol in the atmosphere based on numerous laboratory studies and an explicit chemical mechanism for the oxidation of isoprene. The 'simple scheme' is designed

to serve as a computationally efficient alternative for approximating tropospheric OA concentrations without attempting to model the formation and fate of the various aerosol species mechanistically and without explicit thermodynamic partitioning. We note that the simple scheme was developed independently from the complex scheme and should not be regarded as a reduced version of the complex scheme. These schemes are described below and are graphically illustrated in Fig. 1.

**The Simple scheme** treats all organic aerosol as non-volatile. The POA consists of a hydrophobic 'emitted' component (EPOA) with an assumed organic mass-to-organic carbon (OM:OC) ratio of 1.4 and a hydrophilic 'oxygenated' component (OPOA) with an assumed OM:OC ratio of 2.1. Of the organic carbon emitted from primary sources, 50% is assumed to be hydrophilic (OPOA) to simulate the near-field oxidation of EPOA. The atmospheric aging of EPOA is modeled by its conversion to hydrophilic aerosol (OPOA) with an atmospheric lifetime of 1.15

days, with no explicit dependence on local oxidant levels (Chin et al., 2002; Cooke et al., 1999). The EPOA and OPOA species are represented within the GEOS-Chem model using the variable names 'OCPO' and 'OCPI' respectively. In addition, GEOS-Chem includes an online emission parameterization for sub-micron non-volatile marine primary organic aerosol (MPOA) as described in Gantt et al. (2015). The marine POA is emitted as hydrophobic (M-EPOA) and is aged in the atmosphere by its conversion to hydrophilic aerosol (M-OPOA), with the same 1.15-day lifetime.

For the purpose of this study, the hydrophobic and hydrophilic components have been lumped together under the MPOA moniker.

The simple scheme uses a lumped SOA product (SOAS) with a molecular weight of 150 g mol⁻¹ and an SOA precursor (SOAP) that is emitted from biogenic, pyrogenic and anthropogenic sources with fixed OA yields: 3% from isoprene (Kim et al., 2015) and 10% from both monoterpenes and sesquiterpenes (Chin et al., 2002). SOA precursor emissions

from combustion sources are estimated using CO emissions as a proxy, with a 1.3% scaled co-emission of SOAP from

fire and biofuel CO, and a 6.9% SOAP co-emission from fossil fuel CO (Cubison et al., 2011; Hayes et al., 2015; Kim et al., 2015). For biogenic SOA from isoprene, monoterpenes and sesquiterpenes 50% is emitted directly as SOAS to account for the near-field formation of secondary organic aerosol. The SOAP converts to SOAS based on a first order rate constant with a lifetime of 1 day as it is transported through the troposphere (Fig. 1).

For the purpose of this study, the default simple scheme in GEOS-Chem was modified to individually simulate 14 OA lumped model tracers from anthropogenic, biogenic, marine, and pyrogenic sources. These consisted of 6 POA tracers, 4 SOA tracers and 4 SOA precursor tracers, allowing for the independent adjustment of parameters such as emission rates, yields, chemical lifetimes and deposition rates, enabling a robust testing of various sensitivities, and OA source attributions.

**The Complex scheme,** based primarily on Pye et al. (2010) and Marais et al. (2016), is graphically described in Fig. 1. The primary organics are treated as semi-volatile and allowed to reversibly partition between aerosol (EPOA) and gas (EPOG) phase using a two-product reversible partitioning model while simultaneously undergoing oxidation with OH in the gas phase to form oxidized primary organic gases (OPOG) which, in turn, reversibly partition to oxidized primary organic aerosols (OPOA). Primary semi-volatile organic vapors that are oxidized to form lower volatility products are sometimes classified as secondary organic aerosol (Murphy et al., 2014). However, for the purpose of this study, we define SOA as being formed exclusively from volatile precursors, while classifying the OA resulting from the oxidation of primary organic compounds as OPOA, in order to be consistent with previous model studies using GEOS-Chem (Pye et al., 2010). Model EPOG emissions are based on the EPOA inventories used in the simple scheme and have been scaled up by 27% to account for semi-volatile organic matter emitted in the gas-phase (Pye et al., 2010; Schauer et al., 2001). As in the simple scheme, the EPOA and OPOA are assumed to have an OM:OC ratio of 1.4 and 2.1 respectively. The complex scheme also includes the non-volatile MPOA simulation as described above.

SOA formation from anthropogenic, pyrogenic and select biogenic precursors is based on the VBS outlined in Pye et al. (2010) that oxidizes gas-phase SOA precursors (with oxidants - OH, $O_3$) to form alkyl peroxy ($RO_2$) radicals that react with either $HO_2$ or NO. The SOA formed from these second-generation products depends on the $NO_x$ regime –

with high and low $NO_x$ yields and partitioning coefficients based on experimental fits from laboratory studies. The resulting products are classified into two categories based on the origins of their precursors, Anthropogenic SOA (ASOA; formed from the oxidation of light aromatic compounds) and Terpene SOA (TSOA; formed from the oxidation of monoterpenes and sesquiterpenes) that dynamically partition between the aerosol and gas phases based on their saturation vapor pressures and ambient aerosol concentrations. Aerosol formed from intermediate volatility

organic compounds (IVOCs) is modelled using naphthalene as a proxy which, when oxidized, contributes to the ASOA lumped product. A comprehensive overview of the VBS scheme can be found in Pye et al., 2010.

The complex scheme builds on this VBS framework by incorporating aerosol formed irreversibly from the aqueous phase reactive uptake of isoprene (Marais et al., 2016) and the formation of organo-nitrates (Org-Nit) from both isoprene and monoterpene precursors (denoted in Fig. 1) based on work by Fisher et al. (2016). These mechanisms

replace the 'pure-VBS' treatment of isoprene SOA (ISOA) and organic nitrates (formed from the oxidation of isoprene and monoterpenes by $NO_3$) from Pye et al. (2010). The total organic aerosol loadings in the complex scheme are thus

comprised of the EPOA, OPOA, ASOA and TSOA species in addition to the various products resulting from the isoprene and monoterpene organo-nitrate oxidation pathways (organic nitrates from isoprene and monoterpene precursors, aerosol-phase glyoxal, methylglyoxal, isoprene epoxydiols (IEPOX), $C_4$ epoxides, organo-nitrate hydrolysis products, second-generation hydroxy-nitrates and low-volatility non-IEPOX products of isoprene hydroxy hydroperoxide oxidation), lumped here as ISOA and Org-Nit. ISOA and Org-Nit are generated exclusively through the aqueous uptake pathway and do not include any 'non-aqueous' OA. The model does not explicitly consider cloud processing of SOA. More information on the treatment of OA in the complex scheme can be found in the SI.

In order to conduct a comparison with a VBS treatment of isoprene SOA (as described in Pye et al., 2010), an analysis was also conducted with the isoprene SOA forming exclusively through the VBS (referred to here as 'pure VBS').

**2.2 Emissions**

Global annual mean emissions of key species for a single simulation year (2013) are shown in Table 1. The corresponding emissions (and atmospheric sources) for OA species are shown in Table 2. Year-specific pyrogenic emissions are simulated at a 3-hour resolution from the GFED4s satellite derived global fire emissions database (van der Werf et al., 2017). Global anthropogenic emissions follow the Community Emissions Data System (CEDS) inventory (Hoesly et al., 2018). Anthropogenic IVOC emissions are estimated using naphthalene as a proxy (see SI for more information), which is assumed to have the same spatial distribution as benzene and is scaled from the CEDS inventory using the same approach as Pye and Seinfeld (2010). These emissions are overwritten with regional inventories when available, such as the National Emissions Inventory (NEI 2011) for the US (as described by Travis et al., 2016), the Big Bend Regional Aerosol and Visibility Observational (BRAVO) inventory for Mexico (Kuhns et al., 2005), the Criteria Air Contaminants (CAC) inventory for Canada ([https://www.canada.ca/en/environment-climate-change.html](https://www.canada.ca/en/environment-climate-change.html)), the European Monitoring and Evaluation Programme (EMEP) inventory for Europe ([http://www.emep.int/](http://www.emep.int/)), the Diffuse and Inefficient Combustion Emissions (DICE) inventory for Africa (Marais and Wiedinmyer, 2016) and the MIX inventory for Asian emissions (Li et al., 2017). In addition to the anthropogenic and pyrogenic inventories listed above, nitrogen oxides are also emitted from lightning (Murray et al., 2012; Ott et al., 2010), soil (Hudman et al., 2012) and ship (Holmes et al., 2014) sources. Biogenic emissions for isoprene and terpene species in GEOS-Chem are based on the coupled ecosystem emissions model MEGAN (Model of Emissions of Gases and Aerosols from Nature) v2.1 (Guenther et al., 2012). All emissions are managed via the Harvard-NASA Emissions Component (HEMCO) module (Keller et al., 2014). We note that given the inter-annual variability in emissions, particularly from fires, the emissions for years other than 2013 may differ somewhat from the values shown in Table 1 and Table 2.

In the simple scheme, 50% of the primary OA is emitted as EPOA and 50% is emitted as OPOA to approximate the near-field aging of EPOA. Total OC emissions are 31.2 TgC. Given the OC:OM ratios of 1.4 and 2.1 assumed for EPOA and OPOA respectively, total POA emissions in the simple scheme are 21.8 Tg EPOA and 32.8 Tg OPOA for a total annual POA emission of 54.6 Tg. We note that OPOA emissions in the simple scheme are a subset of the sources listed in Table 2 since they do not include atmospheric formation through the oxidative aging of EPOA. In the complex scheme, all POA is emitted as gas-phase EPOG after scaling the same inventory used in the simple

scheme by 27% to account for the extra gas-phase material. Total primary emissions in the Complex Scheme are thus exclusively from EPOG gas phase emissions and amount to 55.4 Tg yr[-1]. Both schemes emit an additional 7.0 Tg yr[-1] of OA from marine sources. The simple scheme also directly emits 71.7 Tg yr[-1] of SOA (in the form of SOAS and SOAP), over half of which come from anthropogenic sources. The total OA source (POA + SOA; includes direct emissions and atmospheric formation) in both the complex and simple schemes (150.1 Tg yr[-1] and 145.3 Tg yr[-1] respectively; Table 2) is greater than the ensemble median OA source of around 100 Tg yr[-1] calculated by Tsigaridis et al. (2014) across a set of various global models.

**2.3 Model Evaluation**

Two primary metrics have been used through this study evaluate model performance compared to ambient observations (see Sect. 3) – the coefficient of determination ($R^2$) and the normalized mean bias (NMB). The coefficient of determination is defined by the regression fit using Eq. (1) and can be interpreted as the proportion of the variance in the observational data that is accurately captured by the model. The normalized mean bias is calculated using Eq. (2). A positive NMB indicates that the model is biased high on average and vice versa.

$$Coefficient\ of\ Determination\ (R^2) = 1 - \frac{Residual\ Sum\ of\ Squares}{Total\ Sum\ of\ Squares} \tag{1}$$

$$Normalized\ Mean\ Bias\ (NMB) = \frac{\sum_1^n (Model - Observation)}{\sum_1^n (Observation)} \tag{2}$$

**3. Description of Observations**

For the purposes of evaluating the GEOS-Chem model, we focus on airborne data which provides regional 3D sampling and reduces the challenges associated with model representation error at single sites. We further define a set of observations that make use of a single measurement technique, were publicly-accessible, and that do not extend beyond the last decade. The resulting observations are from 15 aircraft campaigns conducted between 2008 and 2017 and cover a wide range of geographic locations and chemical regimes. Table 3 provides a brief overview of the various campaigns included here and Fig. 2 shows the spatial extent of the individual flight tracks. Aerosol concentrations were measured using Aerosol Mass Spectrometers (AMS) (Jayne et al., 2000; Canagaratna et al., 2007) with small variations in the instrumentation and aircraft inlet configurations between the different campaigns (as referenced in Table 3). The AMS measures sub-micron non-refractory dry aerosol mass and is estimated to have an uncertainty of 34-38%, depending on the species (Bahreini et al., 2009). All concentration measurements in this study have been converted to standard conditions of temperature and pressure (STP: 273 K, 1 atm), denoted as µg sm$^{-3}$. In addition to organic aerosol mass loadings, concentrations of other species such as nitrogen oxides, carbon monoxide, isoprene, and sulfate are used in this study to validate chemical regimes (see Sect. 4.2). Table S2 provides an overview of the instrumentation and associated primary investigators for the organic aerosol and trace gas observations. Environmental and meteorological measurements such as temperature and relative humidity are also used in the analysis.

Observations are gridded to the GEOS-Chem model resolution of 2° x 2.5° (or alternatively to 0.5° x 0.625° for comparisons with nested simulations) and are averaged over the model time-step of 10 minutes in cases where multiple

observations were conducted within the span of a single timestep (see SI for more details on model sampling). In order to limit the impact of localized plumes, in particular from fires, we filter the observations to remove concentrations over the 97[th] percentile for each campaign, eliminating measurements that can often exceed 500 µg sm$^{-3}$. This enables a more fair comparison with the model by disregarding the impact of sub-grid features that cannot be reproduced by an Eulerian model (Rastigejev et al., 2010). Following the averaging process, we obtain a merged dataset of over 25,000 unique points, with a broad spatial extent (Fig. 2) covering a variety of chemical regimes representing anthropogenic, pyrogenic, biogenic, and remote environments. Despite the large temporal range of the observational dataset, most of the campaigns analyzed in this study were conducted during the spring and summer seasons, limiting the ability to perform a seasonal analysis.

Based on the proximity to emission sources and exposure to long-range pollutants, there is significant variation in the observed mean, medians and standard deviations across the different campaigns (Table 3; Fig. S1). The campaigns are also influenced by different OA sources depending on their sampling region. The EUCAARI campaign over western Europe (Morgan et al., 2010), KORUS-AQ over the Korean peninsula (Nault et al., 2018), CalNex over California (Ryerson et al., 2013) and DC3 (Barth et al., 2014) and FRAPPE over the central US (Dingle et al., 2016) sample over regions that are heavily influenced by anthropogenic emissions. In contrast, the GoAmazon campaigns during the wet and dry seasons (Martin et al., 2016; Shilling et al., 2018) over the Manaus region in the Amazon and the OP3 campaign (Hewitt et al., 2010) over equatorial forests in southeast Asia are heavily influenced by biogenic emissions, although the GoAmazon campaign in the dry season is also strongly influenced by biomass burning. Additionally, data from both seasons of the GoAmazon campaign are influenced by anthropogenic urban outflow from Manaus (Shilling et al., 2018). Campaigns like SENEX (Warneke et al., 2016) and SEAC4RS (Toon et al., 2016) that conducted measurements over the southeast US are influenced by both anthropogenic and biogenic emissions while the ARCPAC campaign (Brock et al., 2011) during the spring and the ARCTAS (Jacob et al., 2010) campaign during the spring and summer over the northern latitudes are strongly influenced by pyrogenic emissions from forest fires (particularly during the summer) and aged anthropogenic and biogenic emissions over the Arctic region. The KORUS-AQ campaign also includes a short deployment over California. However, for the purpose of this study, we restrict observations from this campaign to those over the Korean peninsula. Lastly, the dataset includes measurements from the ATom-1 and ATom-2 campaigns (Wofsy et al., 2018). We divide the ATom campaigns into two datasets using a land-mask in order to separate the observations of remote, well-mixed air masses over the Atlantic and the Pacific from near-source measurements over North America.

Figure 3 demonstrates that organic aerosol accounts for a significant portion (52% on average) of the total non-refractory aerosol mass loadings measured by AMS across all of the campaigns. The GoAmazon measurements during the dry season have the highest contribution of OA to the total submicron aerosol loading (77%) while the ARCTAS campaign during the spring has the lowest OA contribution of any campaign (31%).

## 4. Results and Discussion

### 4.1 Simulated OA Budget

Figure 4 shows the global annual-mean simulated surface OA concentrations and global annual-mean burdens using the simple and complex schemes for the year 2013 (burden numbers are provided in Table 2). The complex scheme simulates a larger annual-mean OA burden than the simple scheme (2.37 Tg compared to 1.94 Tg). This is largely due to the scaled emissions of the primary organic gases in the complex scheme (greater by a factor of 27%) as well as the semi-volatile treatment of the EPOA/EPOG and OPOA/OPOG species that substantially extends their tropospheric residence time, due to the longer lifetime of the gas-phase component in the boundary layer. As a result, the complex scheme simulates a larger POA burden (EPOA + OPOA + MPOA) of 1.46 Tg, compared to 0.92 Tg POA in the simple scheme. The majority (91.4%) of the POA in the complex scheme consists of oxidized POA and oxidized MPOA (M-OPOA) that, given its aged and chemically processed nature, is often indistinguishable from secondary organic aerosol with typical AMS measurements (Jimenez et al., 2009). Consequently, 94.7% of the global OA burden in the complex scheme is oxygenated organic aerosol (OOA = OPOA + M-OPOA + SOA; Table 2). Similarly, 91.7% of the total POA burden and 96.1% of the total OA burden are oxygenated in the simple scheme.

Both the complex and simple schemes simulate comparable global SOA burdens (0.91 Tg and 1.02 Tg respectively). However, the complex scheme produces more isoprene-derived SOA (ISOA) and biogenic organo-nitrates (Org-Nit) than the simple scheme (Fig. 4d), particularly over areas with elevated isoprene and anthropogenic sulfate concentrations (such as the southeast US and southeast Asia) since the ISOA formation is acid-catalyzed. The explicit aqueous uptake mechanism for the isoprene-derived SOA products also results in substantially larger global isoprene SOA burdens (0.31 Tg) when compared to the 'pure-VBS' treatment of isoprene-derived SOA that simulates an annually averaged ISOA burden of 0.12 Tg. This is consistent with other comparisons that have shown the VBS treatment in GEOS-Chem under-predicts observed ISOA concentrations compared to the complex treatment (Jo et al., 2019). Despite the different treatments, both the complex and simple schemes have similar terpene-derived SOA (TSOA) burdens at 0.19 Tg and 0.18 Tg, respectively (Table 2).

Anthropogenic SOA (ASOA) is a particularly important global OA source in the simple scheme, accounting for almost a third of the total OA burden. The simple scheme, with its near-field formation of SOA proportional to anthropogenic CO emissions, simulates a substantially larger ASOA burden than the complex scheme (0.63 Tg vs 0.10 Tg; Table 2), particularly over industrialized regions in Asia (Fig. 4c). Previous studies that have constrained global SOA burdens using observed mass loadings have proposed a missing model SOA source over anthropogenic regions (Spracklen et al., 2011), as have recent regional studies (Schroder et al., 2018; Shah et al., 2019). The simple scheme appears to capture a greater fraction of this missing burden. However, we note that ASOA yields in the simple scheme are based on a lumped parameterization over the Los Angeles basin (Hayes et al., 2015) and might not be representative of global yields across different chemical regimes. The global ASOA burden of 0.63 Tg is 4 times greater than the ASOA burden proposed by Spracklen et al. 2011, but well within the 'anthropogenically controlled' SOA burden proposed by the same study. This suggests that the simple parametrization in its current form might unintentionally represent some anthropogenically-controlled biogenic SOA. Additionally, while the simple scheme includes separate SOA yield

parameters for fossil fuel and biofuel combustion, the emissions inventories used in this study do not always explicitly differentiate between the two sources. As a consequence, biofuel is often lumped together with fossil fuel CO, potentially leading to an overestimate in ASOA yields from biofuel emissions.

Pye and Seinfeld (2010) performed a similar analysis of tropospheric OA burdens using a semi-volatile POA treatment and a 'pure-VBS' treatment of SOA (i.e. all SOA treated in the VBS, including isoprene) with the GEOS-Chem model
(v8.01.04). Their model simulated 0.03 Tg EPOA, 0.81 Tg OPOA and 0.80 Tg SOA, compared to 0.11Tg EPOA, 1.27 Tg OPOA and 0.91 Tg SOA for the complex scheme and 0.06 Tg EPOA. 0.78 Tg OPOA and 1.02 Tg SOA for the simple scheme in this study. When compared to an analysis of organic aerosol loadings from 31 different chemical transport and general circulation models (Tsigaridis et al., 2014), the primary OA burden from the complex scheme (EPOA + MPOA + OPOA) is substantially higher than most of the models surveyed, while the SOA burden falls
below the mean but above the median of the distribution. The simple scheme, with a much smaller POA burden, is approximately on par with the Tsigaridis et al. (2014) ensemble mean. The simple SOA burden is roughly equivalent to the Tsigaridis et al. model mean (but significantly greater than the median) for global SOA loadings.

Aerosol lifetimes are calculated using the ratio between the mass burden and the physical loss rates due to dry and wet deposition (Table 2). POA in the complex scheme has an average lifetime to physical loss of 6.1 days ($\tau_{EPOA}$ ~ 11.5
days, $\tau_{OPOA}$ ~ 6.3 days, $\tau_{MPOA}$ ~ 3.0 days) in the atmosphere while SOA has a lifetime of 5.3 days on average ($\tau_{ASOA}$ ~ 7.9 days, $\tau_{TSOA}$ ~ 5.3 days, $\tau_{ISOA}$ ~ 5.1 days, $\tau_{ORG-NIT}$ ~ 4.9 days). POA in the simple scheme has an average global lifetime of 4.6 days ($\tau_{EPOA}$ ~ 7.8 days, $\tau_{OPOA}$ ~ 4.6 days, $\tau_{MPOA}$ ~ 3.0 days), while the parameterized SOA species have an average lifetime of 5.2 days ($\tau_{ASOA}$ ~ 5.6 days, $\tau_{TSOA}$ ~ 4.3 days, $\tau_{ISOA}$ ~ 4.7 days). POA lifetimes in both the complex and simple schemes are similar to the simulated POA lifetimes from Tsigaridis et al. (2014) who calculated an
ensemble mean POA lifetime of approximately 5 days. SOA lifetimes from this study are lower than the ensemble-mean of 8 days calculated by Tsigaridis et al. (2014). The range in aerosol lifetimes can be attributed to several different factors. The hydrophobic nature of EPOA leads to longer lifetimes against wet-deposition since the particles are unaffected by rainout. The spatial distribution of the different aerosol types also plays an important role in determining their lifetimes, with species emitted over marine / tropical regions experiencing a higher likelihood of
being deposited via wet deposition than aerosol over drier regions. Surface land types also affect dry deposition velocities, impacting aerosol lifetimes. In addition, there is a marked difference in lifetimes between the semi-volatile species in the complex scheme and non-volatile species in the simple scheme. Due to the temperature dependent partitioning, the semi-volatile aerosol species are often in the gas-phase in the warmer parts of the troposphere and are advected to higher altitudes before they partition to aerosol. The non-volatile species do not simulate this process and
are more likely to be deposited before they can be transported to higher altitudes.

### 4.2 Regime Analysis

We use the observations from the 15 field campaigns described in Sect. 3 as a single coherent dataset. Given the wide range of chemical regimes sampled by the various field campaigns, a method for classifying the observations is needed to better inform the model-measurement comparisons. While the chemical composition of the observed OA can
provide some insight into source types or aging, a comprehensive classification is not possible using only the

observations, requiring that we rely on the model for such a segmentation. In this analysis, we use the relative dominance of the different OA species within the GEOS-Chem simple scheme simulation to classify the measurements into different regimes (described in Table S3 in Supplementary Information). The sorting algorithm weights the relative importance of the three OA source types – Anthropogenic (A), Biogenic (B), and Pyrogenic (F), based on their relative contribution by mass to the total OA loading in the model. Any data point with a source contribution greater than 70% of the total organic mass loading is categorized as being dominated by that source (such as A for Anthropogenic). Although this threshold limit is somewhat arbitrary, an analysis of different threshold values between 60%-80% shows that the resulting classifications are not particularly sensitive to changes within this range. Data points without a single dominant source but with two large sources, contributing greater than 85% of the total OA mass, are classified into a second type of regime category (such as AB for Anthropogenic/Biogenic) and points without any dominant OA source types are classified into the mixed regime category (AFB). Points with an aggregate OA mass concentration below 0.2 $\mu$g sm$^{-3}$ across the three source types are classified as 'Remote / Marine'. Points where MPOA contributes over 50% of the mass are also categorized under the 'Remote / Marine' regime.

While we expect these model-based categories to adequately reflect source influences (i.e. biogenic emissions over the Amazon vs. anthropogenic emissions over Asia), the relative mass contributions simulated by the model are subject to large uncertainties in OA formation and lifetime. As noted in Sect. 3, sampling conditions over the regions can vary significantly from the model treatment (such as the sampling of the Manaus anthropogenic plume or biomass burning plumes during the 'biogenic' GoAmazon campaign). Due to the coarse model resolution, the regime segmentation described above is incapable of accurately categorizing some of these data points. We therefore compare the relative concentrations of observed NO$_x$, CO and isoprene to independently validate the segmentation approach. For instance, mean observed NO$_x$ values over the Anthropogenic regime approach 1 ppb, compared to 0.36 ppb over the AB regime and 0.17 ppb over the Biogenic regime, consistent with the expected chemical signature over these regions. Similarly, averaged isoprene observations over the biogenic regime are over 20 times greater than average measurements over the Anthropogenic regime.

Median concentrations over anthropogenic regions are markedly lower than those over other sources. Fire influenced regions display the highest variability, consistent with the expected source profile. Table S1 provides an overview of the observational data-sets used for this validation. An overview of the resulting segmentation, validation, and regime categories is provided in Table S2. Figure 5 provides a spatial representation of the regime categorization for all the flight data. We note that a large proportion of the observations from the GoAmazon and OP3 campaigns are densely co-located over the Amazon and Borneo and are thus difficult to discern in the figure. We also note that the 'remote' points over the south-east US represent observations in the upper troposphere and are plotted over points in the lower troposphere making them difficult to distinguish. Figure S2 provides a spatial characterization of the different regimes differentiated by altitude for further clarity. While the regime analysis provides useful insight into the primary sources of OA over the region, the classifications are intended to be broad and do not, for instance, distinguish between fresh and aged aerosol contributions from the same source. For example, a number of points over the northern Atlantic and Pacific oceans that are classified as anthropogenic because they are composed of a minimum of 70% anthropogenic OA from continental sources and are high enough in concentration to not be classified as 'remote'.

**4.3 Evaluation of Model Simulations against Airborne Measurements**

Here we evaluate the two model schemes against the suite of airborne observations described in Sect. 3. Despite the substantial differences described in Sect. 2.1, both schemes reproduce the broad distribution (Fig. 6a) of OA observations. While the schemes exhibit slight offsets in their peaks near the lower end of the distribution, there is no evidence of a large systematic skew compared to observations, suggesting that there is not an obvious mode of formation or loss of OA that the model fails to capture. Differences between the two model distributions are also relatively small and both exhibit fairly comparable skill. The simple scheme is less biased than the complex scheme on average with median OA values of 0.81 μg sm$^{-3}$ and 0.86 μg sm$^{-3}$ respectively, compared to the observational median of 0.68 μg sm$^{-3}$. An analysis of the model-observation distributions for the individual campaigns (see Fig. 7) demonstrates that both model schemes appear to overestimate OA mass at the low and high ends of the distribution for several campaigns (as seen in the case of KORUS, GOAMA-W and OP3), while underestimating organic aerosol loadings in the middle of the distribution, suggesting a potential mischaracterization of aerosol sources and lifetimes over these regions. This might also be the result of the coarse model resolution in regions with a high spatial variance in source strengths. Both model schemes underestimate the lowest concentrations and overestimate the highest concentrations over the ocean (ATOM1-W and ATOM2-W). However, Fig. 6a suggests that these are not pervasive issues with the OA simulation at the global scale. We note, however, that this could be due to an averaging effect. Figure 6b shows the same comparison for sulfate, as a benchmark for a species that is generally well simulated by the GEOS-Chem model (Fisher et al., 2011; Heald et al., 2011; Kim et al., 2015). While the comparison suggests that there continues to be further scope for improvement within the OA chemical schemes, the model simulations are approaching the skill of the sulfate simulation both in terms of bias (the sulfate simulation normalized mean bias of 0.20 is similar to the model OA bias outlined above) and captured variability (with an $R^2$ of 0.62 for the model sulfate scheme relative to the observations, compared to an $R^2$ of 0.41 and 0.44 for the simple and complex OA schemes respectively). This suggests the potential importance of other drivers of variability common to both sulfate and organic aerosol, such as emissions and transport, in controlling aerosol concentrations.

Figure 8 shows that both the complex and simple schemes exhibit substantial skill in capturing the vertical OA profile across the aggregate dataset, with a vertical $R^2$ of 0.97 and 0.95 across the complex and simple schemes respectively. Despite significant differences in the treatment of OA formation and atmospheric processing (and thus the source of simulated OA), both schemes appear to have similar skill in reproducing the observed vertical profile across the individual regimes with the exception of the Remote regime (driven largely by ATOM1-W and ATOM2-W) where both schemes struggle somewhat to reproduce the variability in the observed vertical profile (Fig. S3). This result is not surprising given the low concentrations and the potential for uncertainties in transport and chemical processing to be exacerbated in the remote regime. Overall, the schemes display similar skill at capturing the vertical variability across the different regimes, highlighting that much of this variability is likely driven by the prescribed transport and vertical mixing and is independent of the OA chemical scheme.

When compared in aggregate, the simple scheme is less biased in the lower troposphere, while the complex scheme is less biased in the upper troposphere (Fig. 8; Fig. S3). This could be due to the partitioning mechanism in the complex

scheme that is able to model semi-volatile OPOA and SOA with greater sophistication using the VBS framework. There are also various regime-specific differences in model performance. For instance, the complex scheme significantly overestimates OA in the lower troposphere over fire-influenced regions, likely due to the 27% increase in primary OA emissions to account for the dynamic partitioning between gas and aerosol phase POA. However, both the complex and simple schemes underestimate OA loadings in the mid-troposphere over these same regions. This bias may be due to fire injection from large fires into the free troposphere, particularly over boreal regions (Turquety et al., 2007) that is not captured by the model (all emissions from fires are assumed to be in the boundary layer). This shortcoming is also evident over regions influenced by both anthropogenic and fire emissions (AF Regime). Figure 8 also demonstrates that lower tropospheric concentrations cannot be reproduced over oceans without the inclusion of a marine source of POA, although the comparisons suggest that the marine POA source may be a factor of ~2 too high. While the model appears to capture the vertical profile of OA in anthropogenic regions reasonably well (Fig. 8), there are regional differences (Fig. 9), with large model underestimates of OA in the lower troposphere over California (CalNex), the central US (DC3) and Europe (EUCAARI) and large over-estimates over Korea and parts of the Pacific influenced by outflow from Asia (Fig. 9; Fig. S4). These differences are consistent across both the simple and complex schemes, highlighting the importance of accurate anthropogenic emission inventories. The overestimate in the Asian outflow region might specifically point to the importance of constraining Asian IVOC emissions, given that recent studies have suggested that SOA from IVOCs account for a major fraction of the total OA burden across China (Zhao et al., 2016). In regions influenced by both anthropogenic and biogenic emissions (AB Regime) the complex scheme is less biased than the simple scheme, which underestimates the observed concentrations. This difference in bias is likely due to the more sophisticated treatment of isoprene-derived SOA yields (through the aqueous uptake and organic nitrate formation mechanisms) in the complex scheme. The $NO_x$-dependent yields of isoprene and terpene-derived SOA in the complex scheme might also be a source of increased model skill, given that organic nitrates and oxidized isoprene products account for a dominant fraction of the total modelled OA in the complex scheme over these regions. The relative skill of the complex scheme is unsurprising given that the vast majority of the AB Regime is over the southeast US, for which the complex scheme was developed and validated. However, the model skill over the AB regime may be fortuitous, given that recent studies have demonstrated a significant fraction of the observed OA over the southeast US is generated from monoterpene precursors, rather than isoprene (Xu et al., 2018; Zhang et al., 2018). This potentially suggests that monoterpene SOA yields over the southeast US are low in the model. This may also contribute to the underestimate of OA observed during EUCAARI, which is influenced by the forests of Northern Europe (Fig. 9; Fig. S4). Recent work has also demonstrated that organo-nitrates contribute a significant fraction of the total OA mass over certain parts of Europe (Kiendler-Scharr et al., 2016), potentially indicating a model underestimate in organo-nitrate formation over the region. In contrast to its skill over the US, the complex scheme displays a large positive bias over biogenic (B) regions (such as the Amazon), primarily driven by an overestimate in terpene SOA, potentially suggesting that the scheme may not accurately capture global biogenic SOA burdens and needs to be better constrained. The overestimate of OA in both schemes in the boundary layer over the Amazon and Borneo is accompanied by an under-estimate in the upper-troposphere (Fig. 9), potentially indicating overly-rapid model SOA formation or a failure to capture vertical mixing in the region.

We note that while the observations used in this study have a large spatial range, they are temporally limited and might not be representative of the mean state. Atypical meteorological conditions during the different campaigns may contribute significantly to the model-observation bias. For example, the EUCAARI campaign was characterized by a westward flow across Germany and southern UK (Morgan et al., 2010), capped by a strong inversion that limited vertical mixing. Similarly, differences in sampling priorities might impact the chemical composition of the observations in a manner that deviates from climatology. For instance, the GoAmazon campaign was partially oriented toward sampling anthropogenic outflow from the city of Manaus (Shilling et al., 2018), impacting the OA measurements in a manner that the model is ill-equipped to reproduce. However, despite the various gaps in model fidelity, this analysis suggests that both schemes are relatively skilled at capturing the observed magnitude and vertical variability across the different regimes. A previous comparison of observed vertical profiles by Heald et al. (2011) concluded that the 2-product SOA with non-volatile POA model used in earlier versions of GEOS-Chem required additional sinks and sources in order to match observations, suggesting the need for photochemical sinks from photolysis and fragmentation pathways. Figure 8 indicates no obvious need for large additional sinks for either scheme in aggregate, although specific regions may benefit from a more sophisticated treatment of SOA formation and loss.

An analysis of the coefficients of determination ($R^2$) and the normalized mean biases (NMB) across the different regimes (Fig. 10) and campaigns (Fig. 11) indicates that the complex scheme marginally outperforms the simple scheme across the aggregate dataset in its ability to reproduce the observed OA variability (with an $R^2$ of 0.44 compared to an $R^2$ of 0.41 for the simple scheme), with small differences in performance over the different regimes. The simple scheme is more skilled at minimizing bias over the aggregate dataset and most source regimes, but is biased low over the AB and AFB regimes. Figure S4 provides a spatial context to the model-measurement comparisons discussed here. The result that both the complex and simple schemes slightly over-estimate OA in the aggregate dataset is distinct from the conclusion drawn by Heald et al. (2011) who demonstrated a consistent model underestimate of OA over most regions. In this study, median modelled concentrations are within 1 μg sm$^{-3}$ of the observations for 14 out of the 17 datasets analyzed with both schemes. Figure S5 provides distributions of the ratio and bias between the observed and modelled organic aerosol concentrations for both model schemes across the different campaigns.

When compared to the simple scheme, the complex scheme does a superior job at minimizing the bias over much of the US. However, there continues to be an underestimate in OA loadings in both schemes (Fig. 9; Fig. S4). The bias is likely driven by a variety of factors that need to be explored on a regional basis. For instance, a previous model analysis of FRAPPE observations over Colorado suggested that an underestimate of anthropogenic emissions from the oil and gas sector contributed to an underestimate of ASOA in the region (Bahreini et al., 2018). Both schemes over-estimate OA loadings in the northern latitudes (over parts of Alaska and Canada), likely due to an overestimate in POA from fires (Fig. 9; Fig. 11; Fig. S3; Fig. S4). The complex scheme is also biased high over the Amazon rainforest, due to the large mass loadings of terpene SOA and various isoprene and monoterpene-derived organo-nitrates. Conversely, the simple scheme assumes an identical SOA yield from both monoterpenes and sesquiterpenes, likely degrading its skill. Both schemes are biased low over Europe but high over the Korean peninsula, both anthropogenically influenced regions, potentially arising from the different regional inventories (EMEP and MIX) used by the model. Both schemes overestimate the OA concentrations observed during the winter ATom-2 deployment

(Fig. 9; Fig. 11) driven largely by an overestimate in anthropogenic OA, particularly in the North Pacific (Fig. S4); a similar bias is not apparent in the summertime ATom-1 deployment, suggesting a potential seasonal overestimate in anthropogenic emissions in Asia that may warrant further study. In comparison to the complex scheme, simulations conducted using a 'pure-VBS' treatment of SOA were significantly less skilled at capturing OA variability and minimizing model bias over the aggregate dataset, demonstrating the value of an explicit description of isoprene SOA over the non-mechanistic VBS treatment.

**4.4 Exploring the model-measurement differences in OA**

There are many factors that contribute to the model performance over individual campaigns or regions and investigating the specific drivers of regional differences is not the goal of this work. However, here we explore general features of the model-measurement comparisons to identify issues that may inform the development of future model OA schemes.

There is a large spread in the model-observation bias both within and across the individual campaigns. A comparison of OA metrics (such as $R^2$ and NMB) with the corresponding model sulfate simulations for the same campaigns demonstrates a similar variance (Fig. 11). This suggests that the lack of model skill over certain campaigns could be due to physical processes, such as transport and deposition, that impact both OA and sulfate species and are independent of the chemical scheme utilized.

A comparison between the simulated and observed coefficients of variation (CV; defined as the ratio of the standard deviation to the mean) for the different campaigns indicates that both the complex and simple schemes are relatively skilled at capturing the range of observations within the individual campaigns, with the CV from the simple scheme and the complex scheme both showing a high degree of correlation when compared to the observed CV ($R^2$ of 0.7; Fig. 12). The CV provides a measure of statistical dispersion. Figure 12 highlights how localized campaigns such as GoAmazon and FRAPPE have low CVs. Both schemes demonstrate a lack of ability to accurately capture intra-campaign variability (described above by the campaign $R^2$ in Fig. 11). The coarse model resolution and the resulting inability to resolve sub-grid concentration and emission gradients is likely an important barrier to model skill, particularly across more localized campaigns (with low CVs) with smaller dynamic ranges and/or spatial extents, like OP3, KORUS-AQ and FRAPPE. To explore this, additional simulations (not shown) were conducted using a nested 0.5° x 0.625° grid with the simple scheme (while maintaining all other model parameters) over North America for the FRAPPE campaign, and over Asia for the KORUS-AQ and OP3 campaigns. The nested simulations performed significantly better at capturing the observed variability in OA for FRAPPE (with a change in $R^2$ from 0.19 to 0.34). However, the nested KORUS-AQ simulations resulted in a decrease in model skill, with a change in $R^2$ from 0.37 to 0.25. This result suggests that uncertainties in emission inventories and meteorology over Asia may degrade higher-resolution comparisons, consistent with recent work demonstrating deficiencies in emission inventories in the region (Goldberg et al., 2018). The nested simulations also did nothing to improve model fidelity for the OP3 campaign over Borneo (with a change in $R^2$ from 0.49 to 0.48). Biogenic emissions and chemical conditions are likely relatively uniform over this region, and therefore a higher resolution simulation does not lead to a distinct improvement in the simulation.

To compare the underlying source signatures for the ambient OA concentrations over different regimes, we analyze the relationship between OA and CO concentrations across both the model schemes and the observational data-set (Fig. S6). Generally, the model underestimates the observed OA:CO slope, but captures the relative difference in OA:CO slopes observed in different environments. The two schemes are broadly consistent, and the model skill in reproducing this relationship is not notably better or worse over most regimes or environments, providing little insight into model scheme deficiencies. However, there is a notable difference between the observed and modelled OA:CO slope over the anthropogenic regime (though it is not consistent over all regions), potentially warranting further exploration of regional anthropogenic OA yields within the simple scheme.

Model-bias is also evaluated as a function of a suite of observed parameters (Relative Humidity, Temperature $NO_x$, sulfate, isoprene, CO) to identify any salient relationships (Fig. S7). We find that the model-observation bias in the complex scheme displays a robust positive correlation with the observed relative humidity and sulfate concentrations (Fig. 13). This suggests that the aqueous uptake of isoprene oxidation products in the complex scheme is overestimated in conditions of high humidity and high acidity, and that further work is needed to constrain this formation pathway under a range of ambient environmental conditions. It also suggests that large additional pathways of aqueous SOA formation are unlikely to be missing from the model. In-cloud processing of SOA is not explicitly considered in the complex scheme, with Marais et al. (2016) estimating that the pathway accounts for a minor fraction of the total SOA. However, studies have suggested that cloud chemistry can significantly impact SOA concentrations during certain cloud-cycling events (Brégonzio-Rozier et al., 2016; Giorio et al., 2017), indicating the need for more research to constrain the regional relevance of such systems.

The simple and complex schemes differ significantly in their treatment of primary organic aerosol. The simple scheme simulates POA using two non-volatile primary species while the complex scheme uses two semi-volatile primary species that partition between the gas and aerosol phase. This is an important difference, because aerosol partitioning in the semi-volatile species is sensitive to ambient temperature and organic aerosol concentration, influencing concentrations far away from the original source. Given the differences in POA treatment, an analysis of model skill (in terms of its ability to minimize bias and capture observational variability) was conducted by considering the effects of combining EPOA and OPOA loadings from the complex scheme with SOA loadings from the simple scheme (and vice versa). With an $R^2$ of 0.46 and an NMB of 0.03, this model configuration (complex scheme POA with simple scheme SOA) outperformed both the simple and complex schemes over the aggregate dataset in its ability to capture the observed variability and minimize observational bias, supporting the need to explicitly model the semi-volatile nature of POA (Fig. S8). We note that this analysis assumes a parameterized enthalpy of vaporization of 50 kJ mol$^{-1}$ to estimate saturation vapor pressures for semi-volatile partitioning in the complex scheme, an assumption that needs to be more rigorously examined in field and modeling studies.

Based on the results from the simple scheme, an offline analysis was conducted to optimize the various model parameters by running a multi-variate linear regression in combination with a gradient descent optimizer that used a weighted cost-function to maximize the coefficient of determination and minimize the normalized mean bias. This was done across multiple parameter classes (such as emission rates and yields) in order to ascertain a set of optimized

model parameters. The optimized parameters improved the model coefficient of determination by only up to 5% in most cases. This is perhaps unsurprising given that this simplistic analysis assumes that simulated OA concentrations are linearly correlated with changes in emissions and yields, an assumption that is not truly representative of the model treatment which includes non-linear effects such as wet deposition loss. More work is required to optimize these parameter classes using an online analysis.

We also incorporated a rudimentary $NO_x$ and sulfate dependency into the biogenic SOA yields for the simple scheme, using offline monthly-averaged $NO_x$ and sulfate concentrations from a full-chemistry GEOS-Chem simulation for the year 2013. Isoprene-derived SOA was modelled as having a negative $NO_x$ dependency, ranging from a 3% yield in low-NOx conditions to a 2.25% yield at high-$NO_x$. Monoterpene SOA was also modelled as having a negative $NO_x$ dependency – ranging from a 10% yield under low-$NO_x$ conditions to a 7.5% yield under high-$NO_x$ conditions.
Sesquiterpene SOA yields were simulated as having a positive NOx dependence, ranging from 10% under low-NOx conditions to 20% under high-NOx conditions. These yields were determined based on an analysis of relevant literature (e.g. Kroll et al., 2006; Ng et al., 2007) coupled with various offline optimizations from this study. ISOA was also modelled as having a positive $SO_4$ dependence (from a yield of 1.5% in clean conditions to a high of 4.5% in extremely polluted conditions with high sulfate) based on previous work (Marais et al., 2016) that demonstrated the
importance of the acid-catalyzed SOA formation pathway for isoprene.

The NOx-dependent parameterization did not meaningfully improve model skill. However, the sulfate parametrization improved model performance by a few percentage points, bringing the aggregate $R^2$ to within 0.01 of the complex scheme, demonstrating the potential to further improve model performance. The analysis also points to the limitations of the simple scheme in its current form. For instance, OA yields have also been shown to be highly variable by region
and source, particularly in the case of fires (Jolleys et al., 2014), a facet that is not currently captured within the simple scheme. Chemical processing lifetimes are also highly dependent on the ambient regime, with observational studies finding that OA in urban environments (e.g. Jimenez et al., 2009) is often oxidized at timescales that are significantly faster than the 1.15 days assumed in the simple scheme. Our rudimentary optimization of the simple scheme with a sulfate dependency demonstrates the potential to further improve model performance, although additional work is
needed to conduct a more rigorous optimization of the various model parameters.

## 5. Conclusions

In this study, we use a suite of observations that represent a variety of spatial and chemical regimes to undertake a comprehensive evaluation of the two standard organic aerosol schemes in the GEOS-Chem chemical transport model, both with very different treatments of OA. The simple scheme, which uses non-volatile tracers to model primary
organic aerosol, simulates a total annual POA burden that is approximately two-thirds of the comparable burden simulated by the complex scheme that treats POA as semi-volatile. While the total SOA burdens are similar, the simple scheme simulates an anthropogenic SOA burden that is over 6 times greater than the complex scheme. Conversely, the complex scheme simulates a global burden of biogenic SOA that is roughly 2.5 times greater than the comparable burden in the simple scheme, largely due to higher isoprene SOA and organo-nitrate mass loadings. Due to the lack

of well-differentiated fossil-fuel and biofuel emissions, the simple parameterization likely overestimates ASOA from biofuel sources. We note that the simple ASOA parameterization as applied in this study might also capture some 'anthropogenically controlled' SOA formed from biogenic VOC precursors, potentially accounting for some of the disparities noted above. More work is needed to constrain these yields across different chemical regimes at a global scale.

Despite the substantial difference in the complexity of these OA schemes and the relative magnitudes of their sources, differences in their ability to capture observed airborne OA concentrations from around the world are modest. The simple scheme appears to slightly out-perform the more sophisticated complex scheme in terms of its ability to minimize bias over the aggregate dataset while the complex scheme is slightly more skilled in its ability to capture the observed variability. When compared spatially to the simple scheme, the complex treatment is less biased over the

southeast US and certain regions in North America and Europe, while displaying reduced skill in pyrogenic regimes over the northern latitudes and biogenic regimes in the Amazon, where it produces large overestimates. When comparing vertical profiles, both schemes overestimate OA loadings in the lower troposphere. However, the complex scheme is more skilled at capturing the mid-tropospheric burden, likely due to the more sophisticated semi-volatile treatment of primary OA. Both schemes underestimate mid-tropospheric OA loadings over fire influenced campaigns,

pointing to the potential importance of fire injection into the free troposphere in those regions, which was not modeled in this study. The overestimate of OA in the tropical boundary layer and the underestimate aloft similarly indicates model failure to capture the chemical lifetimes of biogenic SOA formation, or points to deficiencies in its ability to capture vertical mixing in these regions. Our analysis of nested simulations over North America and Asia also points to the importance of constraining regional emissions and local meteorology over Asia in order to improve model

fidelity. As a result of our analysis, we recommend that (1) POA be modelled as semi-volatile, (2) fire POA emissions not be scaled up by 27% in the complex scheme and (3) marine POA be included in the simulation of marine-influenced regions. Further explorations of fire injection heights of aerosols (e.g. Zhu et al., 2018) and anthropogenic emissions of OA precursors, particularly in Asia, are needed. However, despite these deficiencies, both model schemes generally capture the magnitude of the observed OA. This is particularly true, given the 38% uncertainty associated

with the AMS OA observations; 33% of the modeled data-points fall within this observed uncertainty, demonstrating significant progress since the first airborne analysis of OA simulated in the GEOS-Chem model, which revealed up to an order of magnitude biases (Heald et al., 2005).

  The surprising result that both the simple and complex schemes perform comparably across the aggregate data-set challenges our expectations that a more complex and mechanistic description of OA should outperform a highly

parameterized scheme. This may suggest that accurately capturing the source influence (i.e. emissions of OA and its precursors) is a more crucial limitation on current model skill than the specific details pertaining to OA formation. Alternatively, it may suggest that there remain substantial deficiencies in our understanding of the mechanistic formation of OA, as represented in the complex scheme (for example, associated with the oxidation of aromatics). The VBS oxidation of monoterpenes and sesquiterpenes in the complex scheme uses $NO_x$ dependent yields to

determine the formation of second generation oxidation products. However, these yields are uncertain and recent studies have suggested the importance of accounting for interactions between multi-generational oxidation products

when determining these yields, demonstrating that such interactions can significantly depress SOA formation under ambient conditions (McFiggans et al., 2019). Recent work has also demonstrated the importance of $RO_2$ autoxidation pathways in the formation of SOA (e.g. Crounse et al., 2013; D'Ambro et al., 2017; Pye et al., 2019). A more sophisticated, explicit treatment that accounts for these oxidation product interactions under different chemical regimes could thus improve model fidelity (but with an associated computational cost). Additionally, the underlying mechanisms (and related uptake coefficients) behind the treatment of isoprene in the complex scheme were developed and validated primarily using data from campaigns over the southeast US; more work is needed to constrain these coefficients under different chemical regimes outside this region. Finally, the lack of model fidelity could also indicate the importance of better constraining the physical processes inherent to both schemes, such as transport and deposition, or point to the salience of photochemical loss, atmospheric aging and fragmentation loss which are not represented in either scheme (Heald et al., 2011; Hodzic et al., 2016). In addition to these factors, the observational comparison with model sulfate suggests that the large drivers of unexplained model variability might be exogenous to the OA chemical scheme.

At the global scale, the computational advantages and relative skill of the simple scheme make it an attractive tool. Our analysis demonstrates that this computational benefit is accompanied by a relatively limited decline in model skill. However, caution should be exercised when applying such a scheme that fails to incorporate the mechanistic responses necessary to ensure predictive skill (eg: for climate studies). There is thus a need to improve upon both the simple parameterized approach as well as the more sophisticated mechanistic scheme in order to further our understanding of organic aerosol in the atmosphere.

This study highlights the critical need to develop new methods to translate experimental studies on the formation and fate of OA into global models, in order to identify the key processes that are required to reproduce observed atmospheric OA concentrations. The study also indicates the importance of additional observational constraints to benchmark and improve model fidelity. The AMS observations offer a rich mass-differentiated dataset that could be further leveraged using factor ratios and clustering analyses to inform future model evaluations. Standardized reporting of AMS data during future campaigns could enable further model evaluation using a more comprehensive range of the instrument's capabilities. In addition, observations of organic aerosol would be particularly useful in understudied regions such as India, China, Central Asia and Africa. Recent campaigns over these regions (such as the 2016 DACCIWA, 2018 ORACLES and 2016 SWAAMI campaigns) could also be leveraged to study the relevant chemistry. Due to the relative paucity of airborne AMS observations, this study does not include an analysis of seasonal trends. Additional aircraft campaigns over the fall and winter seasons (such as the 2015 WINTER campaign over the north-eastern US) could enable a more comprehensive intra-annual analysis which could provide insight into seasonal sources. There is also a need for more field observations at a regional-scale, as opposed to localized sampling, in order to better constrain and improve the treatment of organic aerosol in large-scale regional and global models. Finally, this analysis, while a comprehensive model evaluation of OA, is limited to two schemes within one model and does not include any surface constraints. An on-going, meticulous evaluation of new OA model schemes against globally distributed datasets is paramount to the advancement of the simulation of the air quality and climate impacts of aerosols.

### 6. Author Contributions

CLH designed the study. SJP modified the code, performed the simulations and led the analysis. JRP, SCF, and EAM contributed to the GEOS-Chem organic aerosol simulation. JLJ, PC, BAN, AMM, HC, JES, RB, JHD, KV provided organic aerosol measurements used in the analysis. SJP and CLH wrote the paper with input from the co-authors and acknowledged individuals below.

### 7. Competing Interests

The authors declare that they have no conflict of interest

### 8. Acknowledgements

This work was supported by the National Science Foundation (AGS-1564495). The authors would like to acknowledge Katherine R. Travis, Jesse H. Kroll, Jeffrey L. Collett, Jr. and Taehyoung Lee for useful discussions and inputs. We also acknowledge the following investigators who provided measurements of $NO_x$, CO, and isoprene: Andrew J.
Weinheimer, Armin Wisthaler, Bruce C. Daube, Carsten Warneke, Chelsea R. Thompson, David J. Knapp, Denise D. Montzka, Donald R. Blake, Eric A. Kort, Eric C. Apel, Frank M. Flocke, Glen Sachse, Glenn S. Diskin, Ilana B. Pollack, Jeffrey Peischl, John E. Shilling, John S. Holloway, Joost A. de Gouw, Lisa Kaser, Markus Müller, Martin Graus, Philipp Eichler, Rebecca S. Hornbrook, Roisin Commane, Sally E. Pusede, Stephen R. Springston, Steven C. Wofsy, Teresa L. Campos, Thomas B. Ryerson, Tomas Mikoviny. RB was supported by the Colorado Department of
Public Health and Environment. JES was supported by the U.S. Department of Energy Office of Biological and Environmental Research as part of the ARM and ASR programs. The Pacific Northwest National Laboratory is operated for DOE by Battelle Memorial Institute under contract DE-AC05-76RL01830. JRP was supported by the U.S. NSF(AGS-1559607) and the NOAA 31(NA17OAR430001). Aerosol measurements from the EUCAARI and OP3 campaigns was collected under NERC grants NE/D013690/1, NE/D004624/1 and NE/E01108X/1. The CU-
Boulder group was supported by NASA (NNX15AH33, NNX15AT96G, and 80NSSC18K0630), EPA STAR (83587701-0), and DOE (DE-SC0016559). This manuscript has not been reviewed by the EPA, and thus no endorsement should be inferred. PTR-MS measurements during DC3 and KORUS-AQ were supported by the Austrian Federal Ministry for Transport, Innovation and Technology through the Austrian Space Applications Programme (ASAP) of the Austrian Research Promotion Agency (FFG). GoAmazon data was obtained from the Atmospheric
Radiation Measurement (ARM) user facility (doi: 10.5439/1346559), a U.S. Department of Energy (DOE) Office of Science user facility managed by the Office of Biological and Environmental Research. The art for Fig. 1 was obtained and modified from public domain images.

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

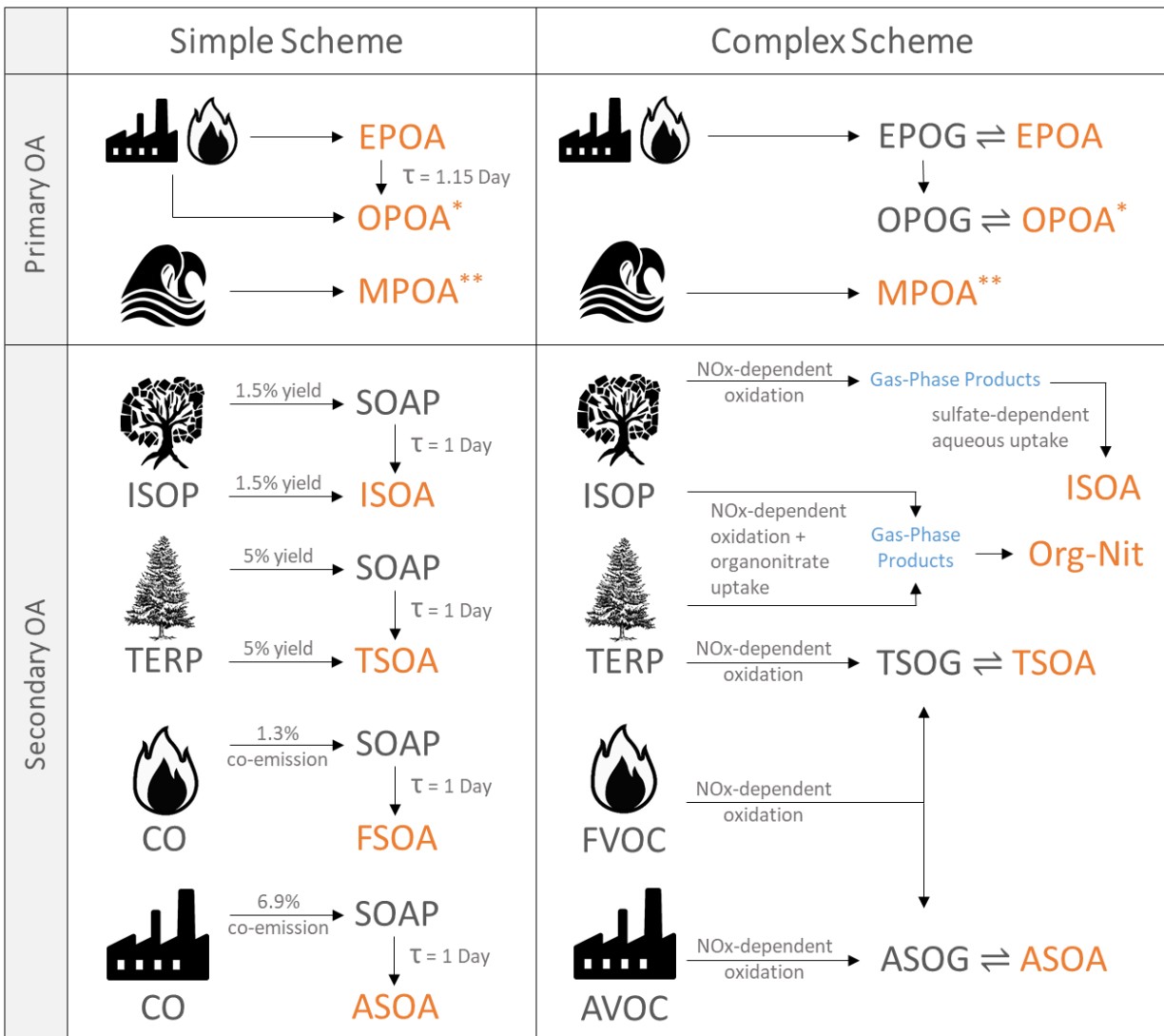

**Figure 1 – A graphical overview of the two organic aerosol model schemes in GEOS-Chem. TERP denotes monoterpenes and sesquiterpenes. Pyrogenic VOCs (FVOC) denote the various volatile and semi-volatile organic compounds emitted from fires while anthropogenic VOCs (AVOC) are comprised of benzene, toluene, xylene and various intermediate-volatility organic compounds that are modelled using naphthalene as a proxy. OPOA\* is sometimes classified as secondary organic aerosol from SVOCs. MPOA\*\* denotes lumped marine POA consisting of both fresh (M-EPOA) and oxidized (M-OPOA) components. Species in orange contribute to OA. See Sect. 2.1 text for details.**



| Species | Annual Global Emissions (Tg year$^{-1}$) |
|---|---|
| **Total Aromatics** | **25.6** |
| - Anthropogenic | 23.5 |
| - Pyrogenic | 2.1 |
| **IVOCs** | **5.43** |
| **Isoprene** | **385.3** |
| **Terpenes** | **153.6** |
| **Total CO** | **891.2** |
| - Anthropogenic | 593.0 |
| - Pyrogenic | 298.2 |
| **Total NO$_x$** | **111.7** |
| - Anthropogenic | 70.7 |
| - Pyrogenic | 12.1 |
| - Lightning | 12.7 |
| - Soil and Fertilizer | 16.2 |

**Table 1. Global annual mean emissions of SOA precursors and relevant species used in the GEOS-Chem simulation for the year 2013.**

| | Complex | | | | | Simple | | | | |
|---|---|---|---|---|---|---|---|---|---|---|
| | Source (Tg yr$^{-1}$) | Burden (Tg) | Lifetime (days) | Dry Dep. (Tg yr$^{-1}$) | Wet Dep. (Tg yr$^{-1}$) | Source (Tg yr$^{-1}$) | Burden (Tg) | Lifetime (days) | Dry Dep. (Tg yr$^{-1}$) | Wet Dep. (Tg yr$^{-1}$) |
| **Total POA** | **87.3**[†] | **1.46** | **6.1** | **14.7** | **72.6** | **73.8**[†] | **0.92** | **4.6** | **13.2** | **60.6** |
| - Emitted POA | 55.4[††] | 0.11 | 11.5 | 2.1 | 1.4 | 21.8 | 0.06 | 7.8 | 1.4 | 1.4 |
| - Marine EPOA | 7.0 | 0.02 | 4.3 | 0.9 | 0.8 | 7.0 | 0.02 | 4.3 | 0.9 | 0.8 |
| - Oxygenated POA* | 74.1[†] | 1.27 | 6.3 | 10.2 | 63.9 | 61.3[†] | 0.78 | 4.6 | 9.4 | 51.9 |
| - Marine OPOA* | 8.0[†] | 0.06 | 2.8 | 1.5 | 6.5 | 8.0[†] | 0.06 | 2.8 | 1.5 | 6.5 |
| **Total SOA** | **62.9**[†] | **0.91** | **5.3** | **7.3** | **55.6** | **71.7** | **1.02** | **5.2** | **9.4** | **62.2** |
| - Anthropogenic SOA | 4.6[†] | 0.10 | 7.9 | 0.6 | 4.0 | 41.0 | 0.63 | 5.6 | 6.2 | 34.8 |
| - Terpene SOA | 13.1[†] | 0.19 | 5.3 | 1.5 | 11.6 | 15.2 | 0.18 | 4.3 | 1.6 | 13.6 |
| - Isoprene SOA | 22.2[†] | 0.31 | 5.1 | 2.3 | 19.9 | 11.6 | 0.15 | 4.7 | 1.1 | 10.4 |
| - Organic Nitrates | 23.0[†] | 0.31 | 4.9 | 2.9 | 20.1 | - | - | - | - | - |
| - Pyrogenic SOA | - | - | - | - | - | 3.9 | 0.06 | 5.7 | 0.5 | 3.4 |
| **Total OOA**** | **145.0**[†] | **2.24** | **5.6** | **19.0** | **126.0** | **140.9**[†] | **1.86** | **4.8** | **20.3** | **120.6** |
| **Total OA** | **150.1**[†] | **2.37** | **5.8** | **21.9** | **128.2** | **145.3**[†] | **1.94** | **4.9** | **22.6** | **122.8** |

*SVOCs from primary sources that are oxidized in the atmosphere, sometimes classified as SOA

**OOA (Oxygenated Organic Aerosol) = OPOA + M-OPOA + SOA

[†]Calculated based on a steady-state assumption with depositional losses in order to account for atmospheric formation.

[††]Primary organic emissions in the complex scheme are in gas phase (EPOG) while primary organic emissions in the simple scheme are in the form of non-volatile particulate. An OM:OC ratio of 1.4 is assumed for the EPOG and EPOA species while an OM:OC ratio of 2.1 is assumed for the OPOA species.

**Table 2. Annual mean simulated global source, burden, lifetime (against physical deposition) and wet and dry deposition rates for the individual OA species averaged over 2013 for the complex and simple schemes.**

| Campaign | Dates (UTC) | Region | Abbreviation | Measurement Technique | Mean / Median / SD |
|---|---|---|---|---|---|
| **ARCPAC** (Brock et al., 2011) | 2008 Spring (03/29 - 04/24) | Arctic / North America | - | C-ToF-AMS | 1.9 / 0.9 / 2.1 |
| **ARCTAS** (Jacob et al., 2010) | 2008 Spring (04/01 − 04/20) | Arctic / North America | ARCTAS-SP | HR-ToF-AMS | 0.7 / 0.4 / 0.9 |
| **ARCTAS** (Jacob et al., 2010) | 2008 Summer (06/18 − 07/13) | Arctic / North America | ARCTAS-SU | HR-ToF-AMS | 3.2 / 0.9 / 5.1 |
| **EUCAARI** (Morgan et al., 2010) | 2008 Spring (05/06 - 05/22) | North West Europe | - | C-ToF-AMS | 2.5 / 2.4 / 2.0 |
| **OP3** (Hewitt et al., 2010) | 2008 Summer (07/10 - 07/20) | Borneo | - | C-ToF-AMS | 0.4 / 0.1 / 0.5 |
| **CalNex** (Ryerson et al., 2013) | 2010 Spring and Summer (04/30 − 06/22) | South West US | - | C-ToF-AMS | 1.3 / 0.8 / 1.4 |
| **DC3** (Barth et al., 2014) | 2012 Spring and Summer (05/18 - 06/23) | Central US | - | HR-ToF-AMS | 2.5 / 1.4 / 2.4 |
| **SENEX** (Warneke et al., 2016) | 2013 Summer (06/03 − 07/10) | South East US | - | C-ToF-AMS | 5.3 / 4.7 / 3.7 |
| **SEAC4RS** (Toon et al., 2016) | 2013 Summer and Fall (08/06 − 09/24) | South East / West US | - | HR-ToF-AMS | 3.2 / 0.6 / 4.6 |
| **GoAmazon** (Shilling et al., 2018) | 2014 Wet Season (02/22 − 03/23) | Amazon | GOAMA-W | HR-ToF-AMS | 1.0 / 0.9 / 0.6 |
| **FRAPPE** (Dingle et al., 2016) | 2014 Summer (07/26 − 08/19) | Central US | - | C-ToF-mAMS | 2.7 / 2.5 / 1.4 |
| **GoAmazon** (Shilling et al., 2018) | 2014 Dry Season (09/06 − 10/04) | Amazon | GOAMA-D | HR-ToF-AMS | 4.6 / 4.6 / 1.8 |
| **KORUS-AQ** (Nault et al., 2018) | 2016 Spring and Summer (05/03 − 06/10) | South Korea | KORUS | HR-ToF-AMS | 4.8 / 2.4 / 5.5 |
| **ATom** (Wofsy et al., 2018) | 2016 Summer (07/29 − 08/20) | Remote Ocean North America | ATOM1-W ATOM1-L | HR-ToF-AMS | 0.1 / 0.1 / 0.2 0.5 / 0.2 / 0.8 |
| **ATom** (Wofsy et al., 2018) | 2017 Spring (01/26 − 02/21) | Remote Ocean North America | ATOM2-W ATOM2-L | HR-ToF-AMS | 0.1 / 0.1 / 0.1 0.1 / 0.1 / 0.1 |
| Aggregate | | | | | 2.4 / 0.7 / 3.6 |

**Table 3. Aircraft measurements of organic aerosol used in this analysis. The statistical metrics for OA provided above (Mean / Median / Standard Deviation) are based on filtered data for each campaign (as discussed in the text) and are in units of μg m$^{-3}$**


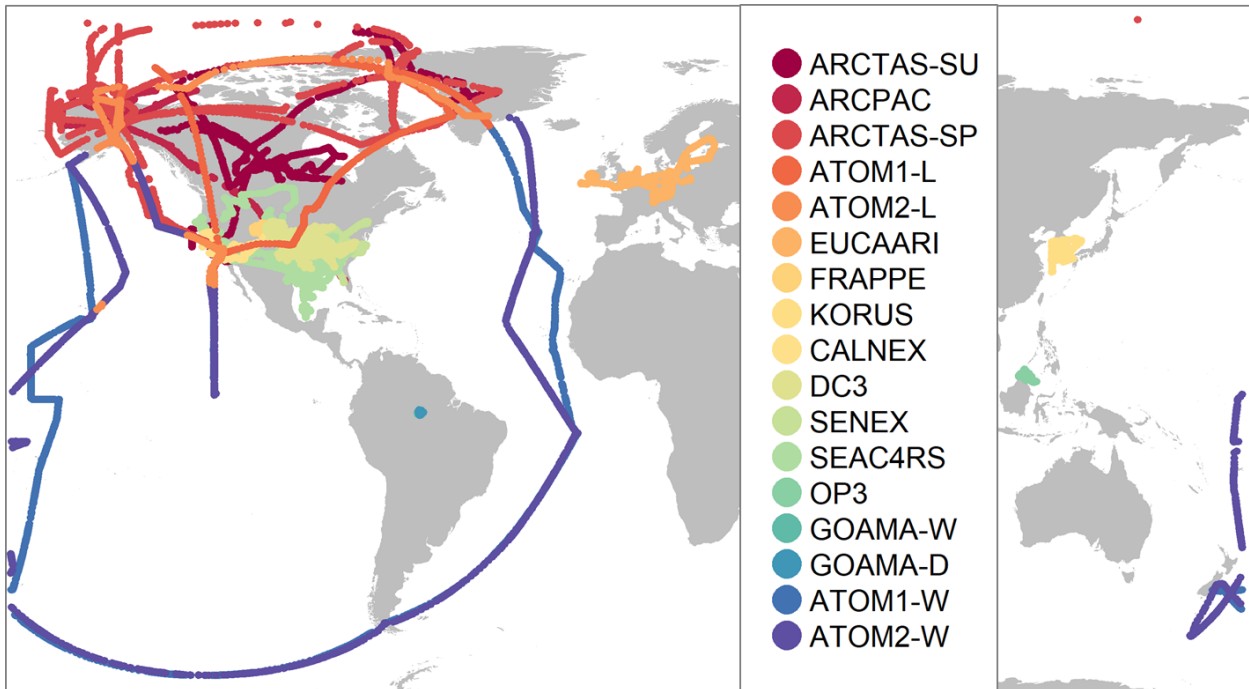

**Figure 2. Location of flight tracks for the airborne field campaigns.**

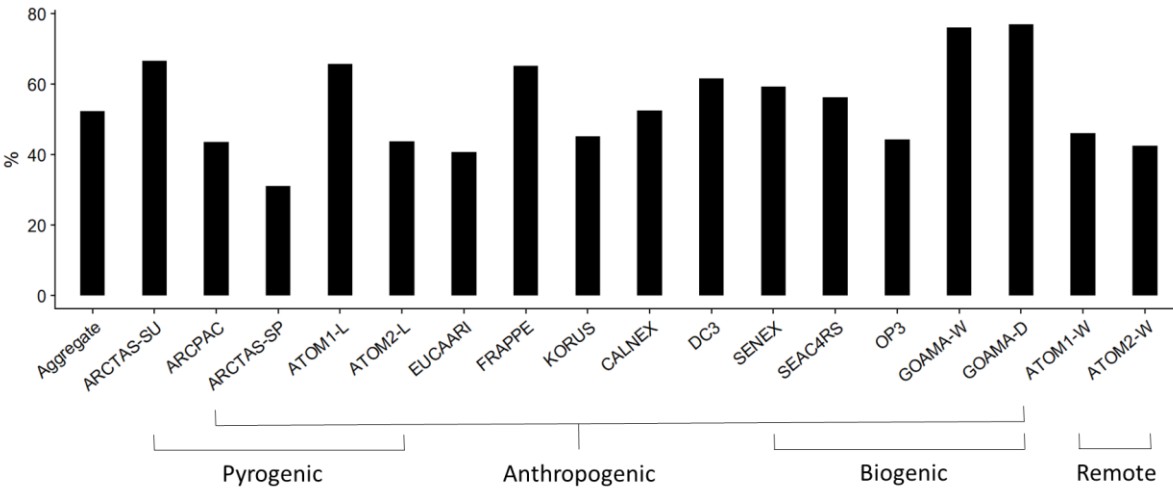

**Figure 3. The percentage contribution of organic aerosol by mass to the total observed non-refractory mass concentrations measured by the AMS, organized by campaign. This includes aerosol mass from organic aerosol, sulfate, nitrate and ammonium. Campaigns are broadly organized based largely on model characterized source influence. However, as noted in the text, this characterization is often not indicative of the true sampling profile. For instance, the GoAmazon campaigns sampled heavily from fire and anthropogenic sources in addition to being strongly influenced by biogenic sources.**

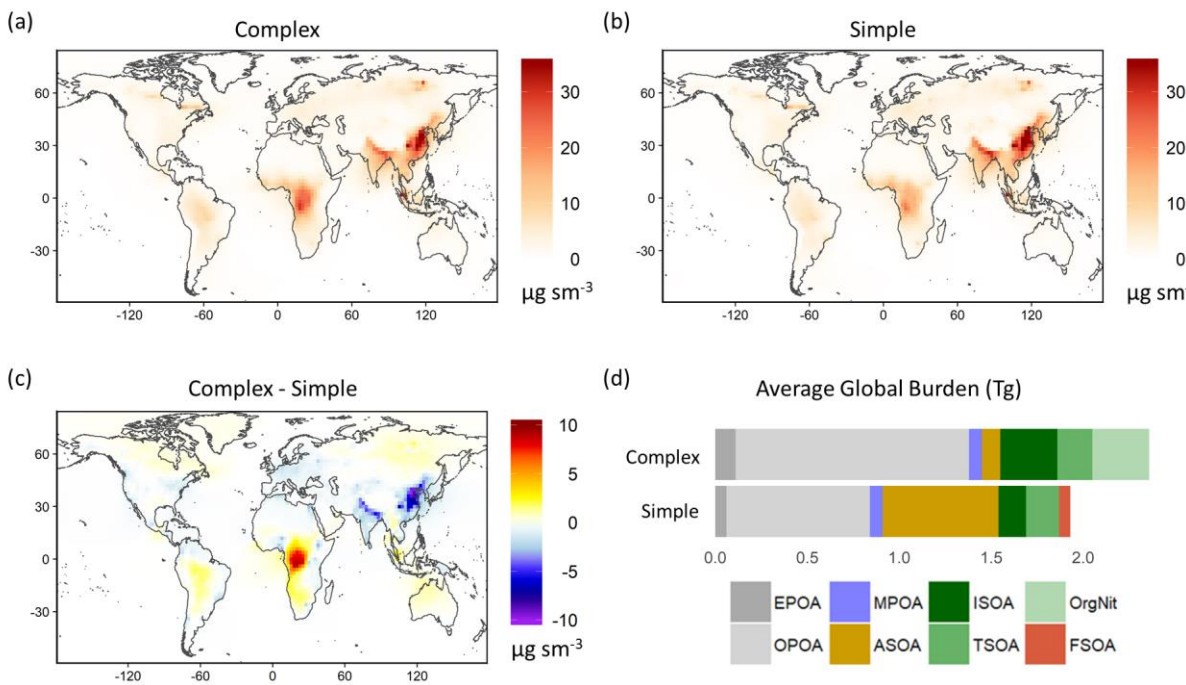


**Figure 4. Global map of simulated OA surface concentrations in 2013 for the (a) complex and (b) simple schemes; Panel (c) illustrates the difference in OA surface loadings between the complex and simple schemes. Panel (d) displays the total global burden for the individual OA species from both schemes averaged over 2013. Refer to Sect. 3 for details on model sampling and averaging.**


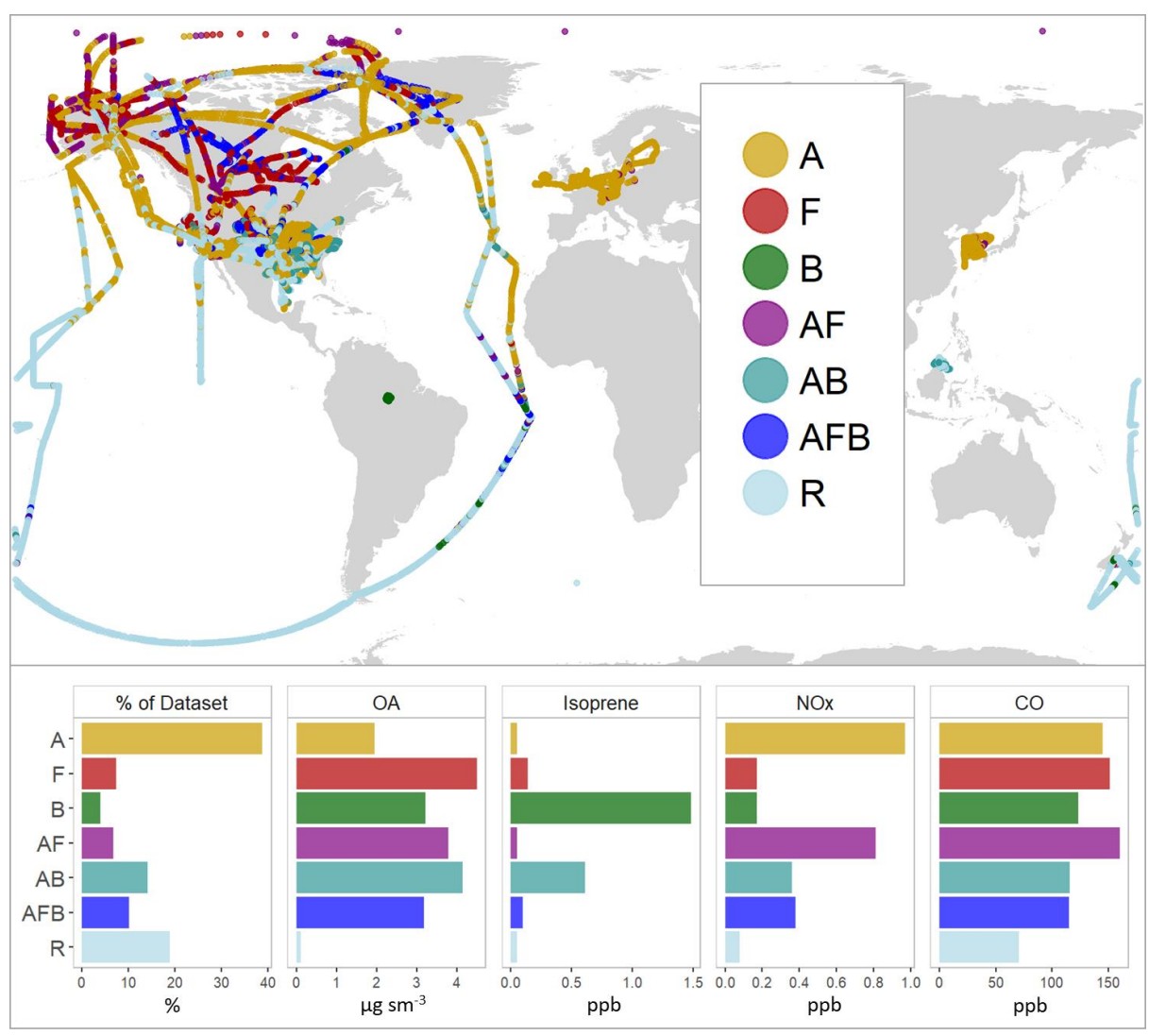

Figure 5. Flight tracks colored by regime type (top). The bar plots (bottom) compare observed mean values for various species across the different regimes. Mean values for OA are in units of μg sm$^{-3}$. Mean values of isoprene, nitrogen oxides and carbon monoxide are in units of parts per billion (ppb). The Regimes are as follows – Anthropogenic (A), Pyrogenic (F), Biogenic (B), Anthropogenic + Pyrogenic (AF), Anthropogenic + Biogenic (AB), Mixed (AFB) and Remote / Marine (R). Refer to Sect. 3 for details on model sampling and averaging. See Fig. S2 for altitude differentiated maps.

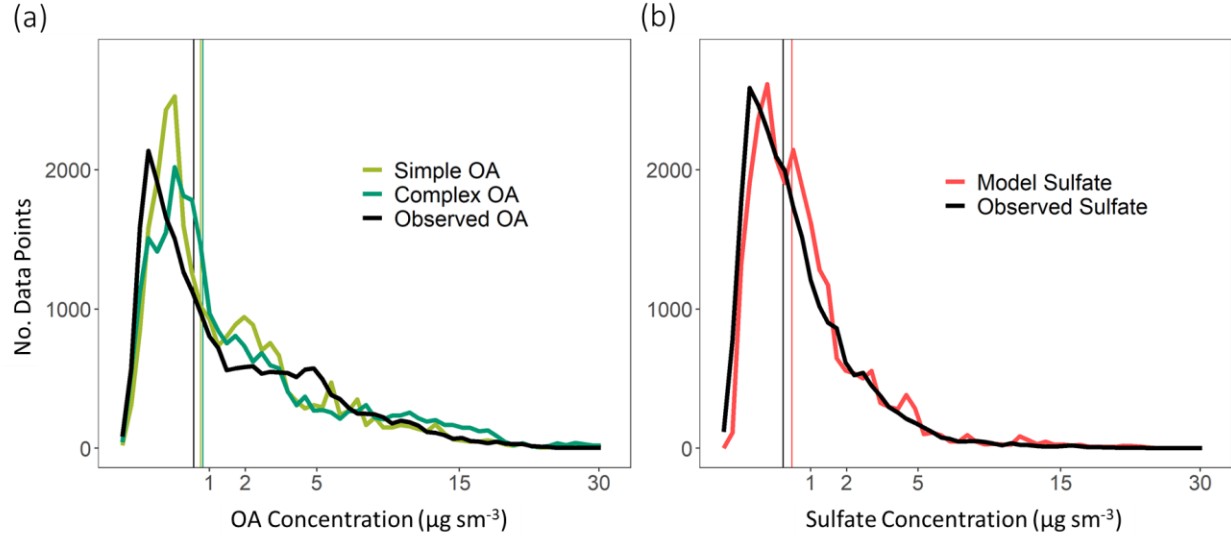

**Figure 6. (a) Distribution plots of OA mass concentrations for the complex scheme (dark green), simple scheme (light green), and AMS observations (black). The x-axis has been transformed using a square-root function. Vertical lines represent median values for the different distributions. (b) Distribution plots of sulfate mass concentrations for the model (red) and AMS observations (black). Refer to Sect. 3 for details on model sampling and averaging.**


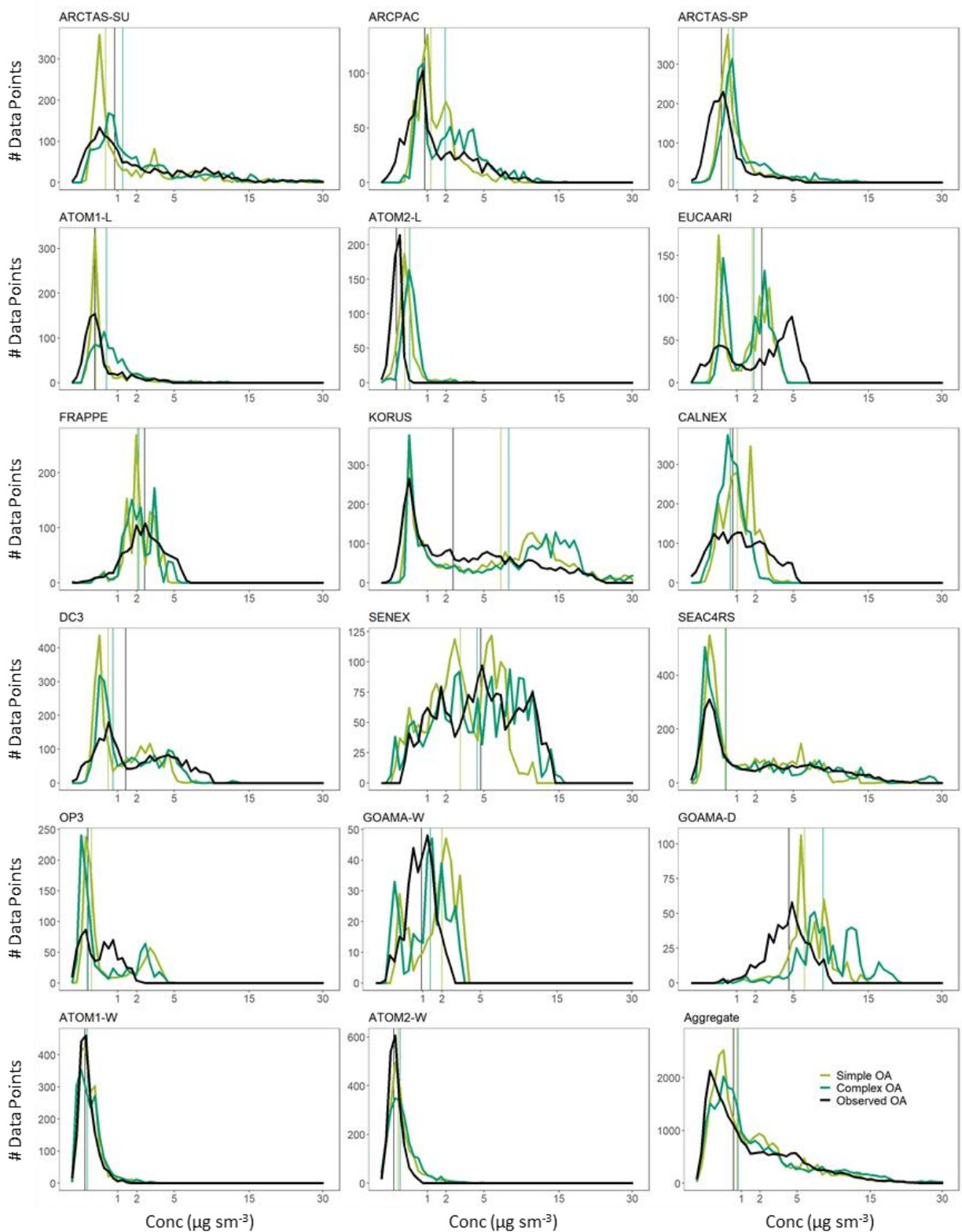

Figure 7. Superimposed distributions from the complex (dark green) and simple (light green) schemes with the observations in black for the different campaigns. Vertical lines represent median values for the different distributions.

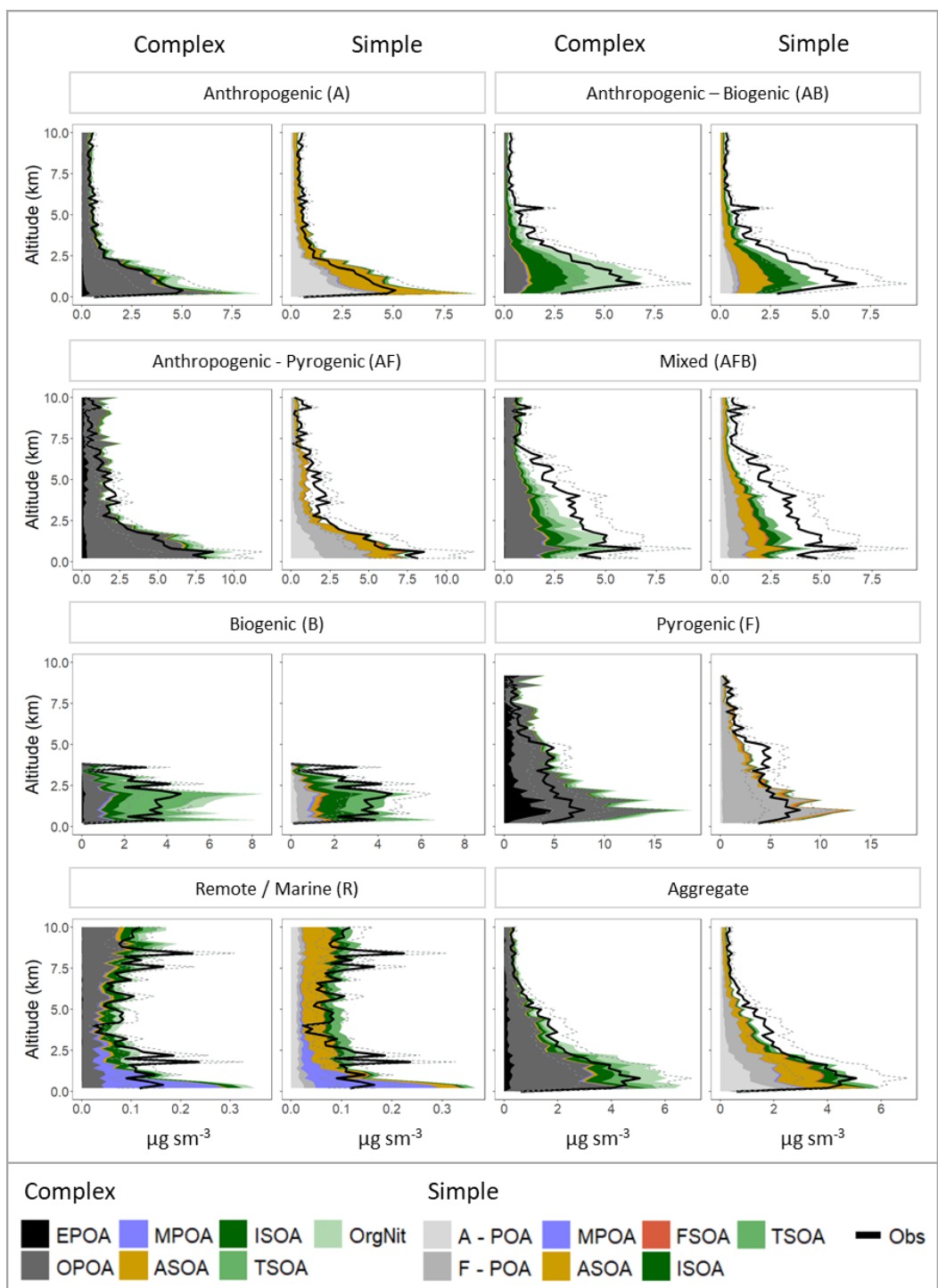

**Figure 8.** Mean vertical profiles (in kilometers) comparing the observed (black) and simulated (colored) OA mass concentrations classified into the different regimes. The dashed lines represent the uncertainty in the observed OA mass loadings. The profiles are binned at 200m intervals. For the simple scheme, A-POA represents anthropogenic POA and F-POA represents pyrogenic POA. Refer to text for other OA categories and details on model sampling.


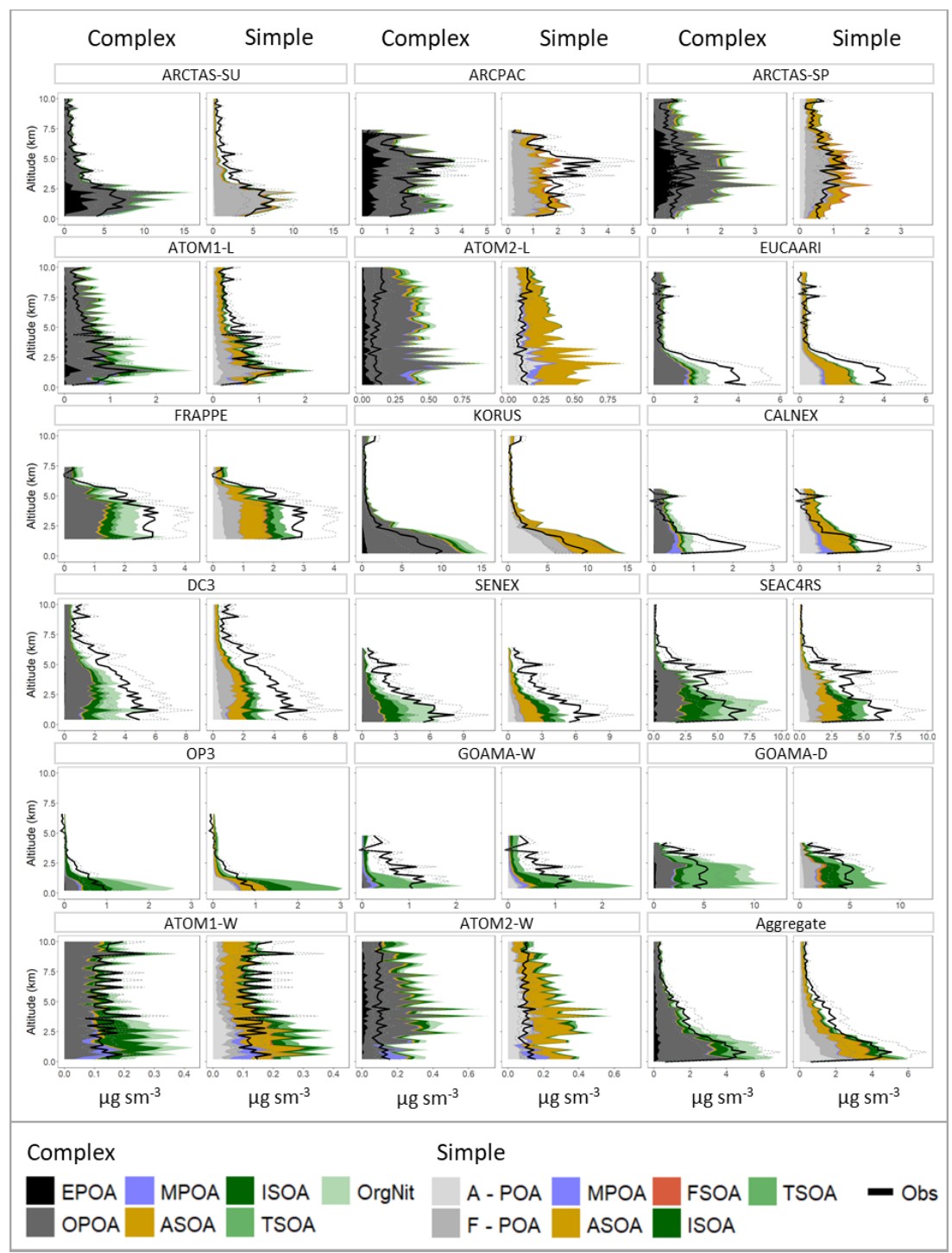

**Figure 9.** Mean vertical profiles (in kilometers) comparing the observed (black) and simulated (colored) OA mass concentrations across the different campaigns. The dashed lines represent the uncertainty in the observed OA mass loadings. The profiles are binned at 200m intervals. For the simple scheme, A-POA represents anthropogenic POA and F-POA represents pyrogenic POA. Refer to text for other OA categories and details on model sampling.

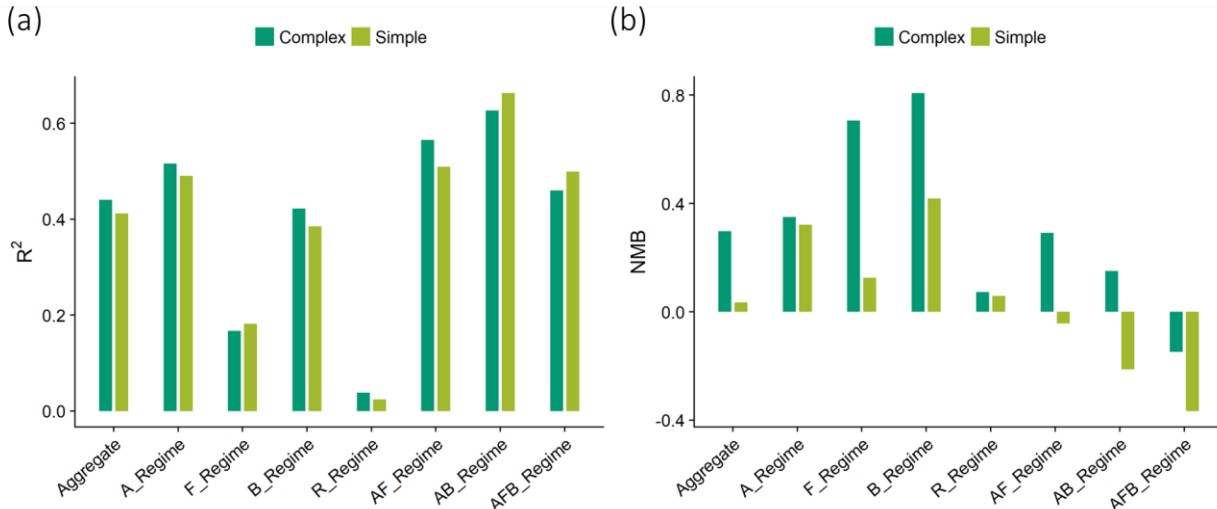

**Figure 10. Statistical evaluation of the OA model skill for the complex (dark green) and simple (light green) scheme against observations shown as (a) the coefficients of determination ($R^2$) and (b) the normalized mean bias (NMB) across the segmented regimes. A positive normalized mean bias indicates that the model over predicts OA loadings.**

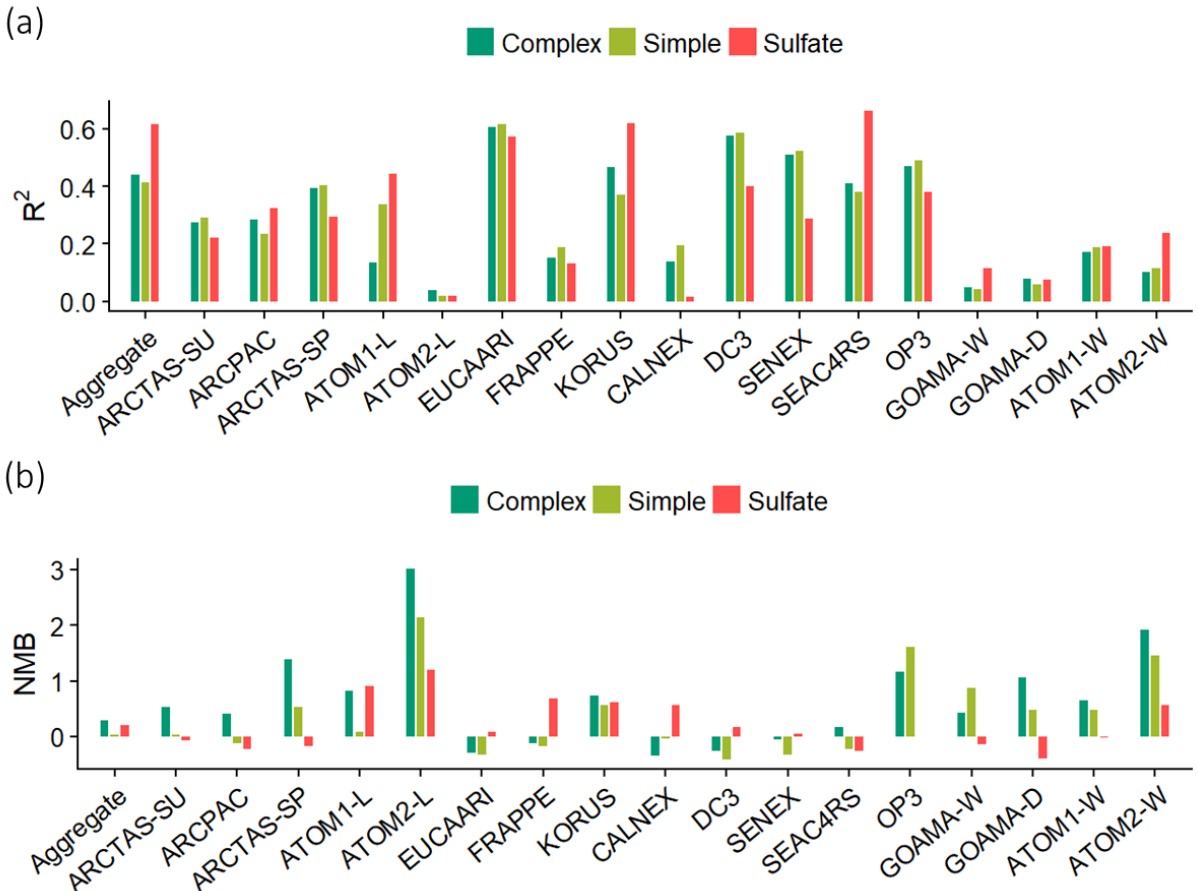

**Figure 11. Statistical evaluation of the model skill against observations shown as (a) the coefficients of determination ($R^2$) and (b) the normalized mean bias (NMB) across the individual field campaigns. The complex (dark green) and simple (light green) OA schemes are compared to the sulfate simulation (red).**

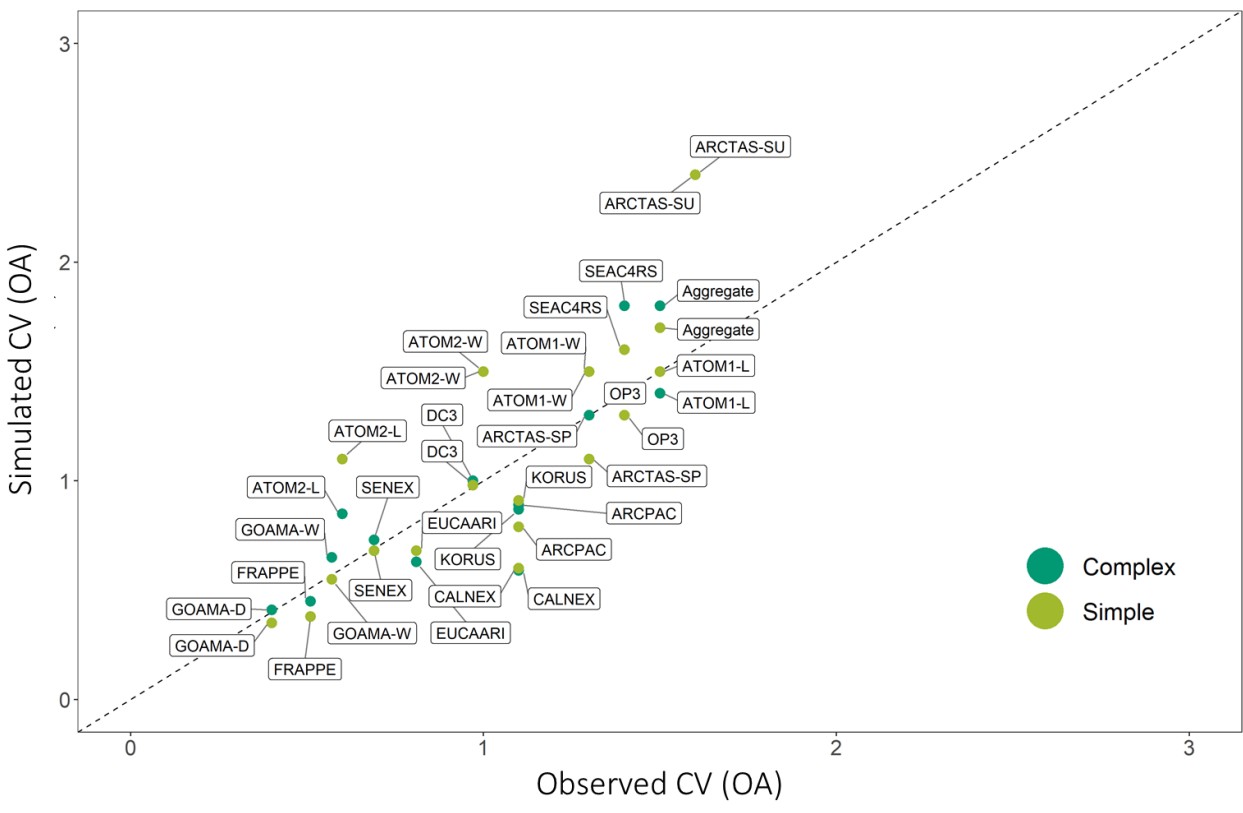

**Figure 12. A comparison of the simulated (GEOS-Chem) coefficient of variation (CV, the ratio of the standard deviation to the mean) for complex (dark green) and simple (light green) OA schemes against the observed CV for each airborne campaign. The one-to-one line is shown as a dashed black line.**

(a)

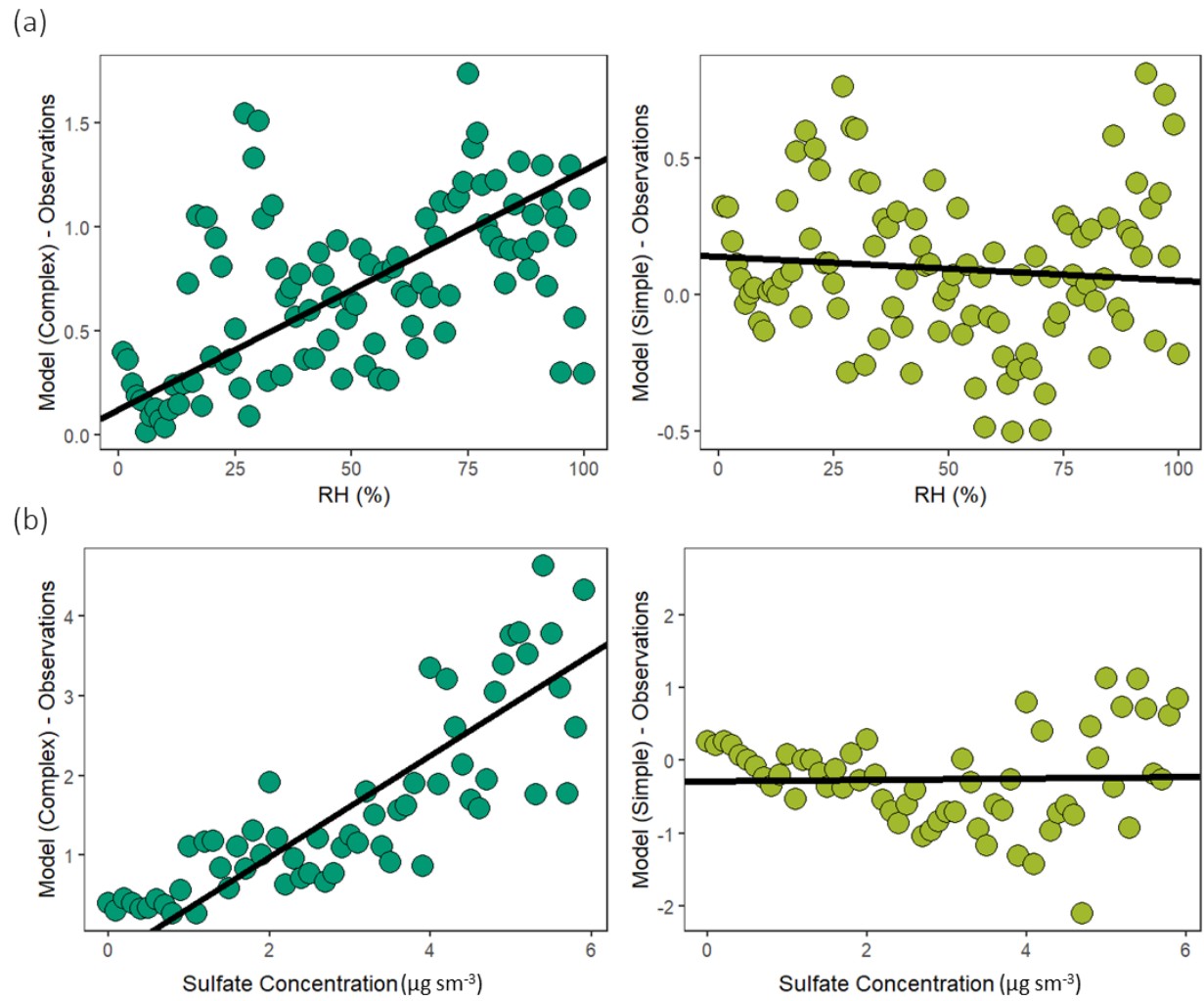

(b)

**Figure 13. A comparison of model-observation OA bias and observed a) relative humidity and b) sulfate mass concentrations for the complex (left panel - dark green) and simple (right panel - light green) OA schemes across the aggregate dataset (observations are binned by intervals of 1% for RH and 0.1 μg sm$^{-3}$ for sulfate). The best fit line is shown in black.**

1215