# Peer review of "An evaluation of global organic aerosol schemes using airborne observations"

_Atmospheric Chemistry and Physics, 2019_

## Referee Comment (RC1) · Anonymous Referee #1 · 30 May 2019

**Review of "An evaluation of global organic aerosol schemes using airborne Observations"**

Two organic aerosol schemes in GEOS-chem are evaluated against observations. A comprehensive set of aircraft measurements of aerosol composition, mainly using the ToF-AMS instrument, is assembled and classified into regimes in which the model is evaluated. A simple scheme with non-volatile primary organic aerosol and with fixed yields for secondary organic aerosols is compared with a complex scheme, in which semivolatile primary and secondary organics are produced in a volatility basis set framework and aqueous reactive uptake for isoprene is included. The performance of the two schemes is similar. High-resolution (or higher-resolution, at least) nested simulations and some tweaks to the simple scheme are used to gain further insights into the behavior of the mechanisms. The paper builds on previous work by many of the same authors, published in Heald et al 2011.

The paper is a valuable addition to the literature. It is well-written, insightful, and will attract attention from the community. I recommend it for publication in ACP. I have the following minor suggestions for improvements.

**Specific comments:**

Introduction: I think it's worth remarking (with appropriate references) that the "simple scheme" is fairly similar to what is used in most climate and earth system models submitted to CMIP6 (NorESM, HadGEM3/UKESM1, I think also GFDL AM4), while a couple of climate models (CESM and E3SM as far as I know) have configurations that are more similar in at least some important respects to the complex scheme (such as including the volatility basis set for semivolatile SOA).

L103 "we perform a series of simulations from 2008 to 2017 using two distinct model scheme": It would be helpful to include a table summarizing the simulations that were performed (one simple, one complex, then some simulations in which the simple scheme was modified, the simple SOA with complex POA, etc), and a more detailed explanation of which periods were simulated – just the times of the flights, or the whole year. Also, at or around L223: "The observations were averaged over the model grid-boxes and timestep." It would be great to be a bit more explicit about how the comparison was done -  for example, was the model output also diagnosed and written out to a file on every timestep, or every few hours?

L112 "A standard bulk aerosol scheme": which one? Also please put into context the subsequent sentence "GEOS-chem also simulates sulfate aerosol…" – is this somehow a separate issue from the bulk aerosol scheme?

Figure 4: It's rather unfortunate that the differences between the two schemes are greatest in areas where you have no measurements: central Africa and inland in China. This is pointed out in the conclusions, but perhaps Africa should be added to the list on line 612 – you could also point out that there do exist some datasets that might already help with resolving the large discrepancies there (DACCIWA, ORACLES, for example)?

"The explicit aqueous uptake mechanism for the isoprene-derived SOA products also results in substantially larger global isoprene SOA burdens (0.31 Tg) when compared to the 'pure-VBS' treatment of isoprene-derived SOA that simulates an annually averaged ISOA burden of 0.12 Tg" - -so was there an additional simulation performed with the aqueous uptake turned off? Can you be a bit more detailed about what differences this causes, maybe adding

another row to Table 2 where isoprene SOA is split into aqueous and non-aqueous contributions?

L295. It is stated in the abstract that the model skill is superior to previous model evaluations, and in this section at line 295 the model is compared to an ensemble from Tsigaridis et al 2014. However, the reasons why the model differs from the ensemble probably vary from model to model. For the GEOS-chem model, the authors already include some comments about how the current model differs from that in Heald et al 2011 at lines 427 and 438. Can the authors identify whether it is changes to the emissions inventories since the 2011 paper, or changes to the OA schemes, that are responsible for the differences? Also, perhaps it is worth saying why some of the campaigns from Heald et al 2011 were used in the current study, but not others (presumably this was just to avoid running the simulations for unfeasibly many years)?

L343: The regime analysis is interesting and very useful in the following interpretation, but needs some further explanation, or possibly further tuning of the classifications, because some features of Figure 5 are rather surprising. In Figure 5, many regions that must be at least relatively pristine compared to the eastern United States are categorized as anthropogenic (large portions of the North Atlantic and North Pacific ATOM flights, much of the Canadian Arctic) and perhaps this can explain the sentence "Median concentrations over anthropogenic regions are markedly lower than those over other sources"?

Then, it looks like most of the eastern USA is classified as "remote". Is "remote" being plotted on top of "anthropogenic" for example, so the high volume of data would cause misleading results where only the last plotted regime shows up? Or is the North Atlantic characterized as anthropogenic because the aerosol mass concentration can be quite high due to a lot of dust (and the North Pacific, potentially, due to volcanic sulfur). Could other aerosol types have been included in the 'aggregate OA mass concentration' threshold of 0.2mugsm-3?

The classification would presumably be quite different if the figure was remade using regime types from the complex scheme rather than the simple scheme. Perhaps this would be interesting to try, just to reproduce Figure 5 (no need to repeat the whole analysis!) It would overcome the shortcoming of the simple scheme already mentioned in the text that it tends to count (for example) anthropogenically influenced biogenic SOA as anthropogenic SOA.

Another way to make this figure 5 less surprising might be to introduce separate categories for 'remote anthropogenic' and 'polluted anthropogenic' based on another mass threshold.

Figures 7 and 8 show that in the remote/marine region, the two schemes also disagree radically on whether the aerosol is primary or secondary, above 2km altitude. This is well discussed in the first paragraph of the conclusion but also seems worthy of greater emphasis and discussion in the paper around line 505. It is remarkable how well both schemes reproduce observations despite this (at least in Figure 7 and in summer, and even the lack of variability noted at line 378 does not seem to be a large effect in Figure S2). It makes sense that semivolatile POA gets to high altitude more effectively than non-volatile POA, so it stands to reason that the complex scheme is doing well. So then it seems surprising that overall the model with complex POA and simple SOA, from fig S7, seems to underestimate OA in the remote region (negative NMB)- if I understand how the NMB is defined, it should overestimate it. Similarly, the reverse arrangement, with simple POA and complex SOA, should underestimate, but the NMB is positive. What does the altitude profile look like?

There are several ways to calculate NMB – please can the authors include an equation somewhere in the text to define it?

In Figure 8, ATOM-2-W shows both models substantially overestimate SOA at high altitude, while ATOM-1-W is fine. This is explained in the text as a seasonal effect, L455. Does the overestimate square with the near-perfect agreement in Figure 7 remote/marine?

I realise this is outside the scope of the current study, but do the authors intend to make use of a fuller range of capabilities of the ToF-AMS in tracking signatures of different aerosol sources – for example signatures of biomass burning (f60), SOA (f44) etc, relevant fragment ratios, etc, etc in future work? Or even PMF factors? I'm not an instrumentation expert, but my understanding is that the ToF-AMS can provide much richer information than simply OA, sulfate, and total mass concentrations, and this could be used in future model evaluations to great effect. It is also one reason why the observation dataset is substantially improved relative to Heald et al 2011. I think this merits a comment in the conclusions alongside the comments about the importance of additional observational constraints from new campaigns, since the expense of new observations would be much easier to justify once the existing datasets have been fully exploited.

**Technical comments**

Figure 6, 10, and S2: the colors are confusing compared to the AMS conventions, please use red for sulfate and green for complex OA.

In Table 3, the standard deviation is often greater than the mean and median, yet negative concentrations aren't possible. This is clearly a matter of opinion and a pretty minor point, but maybe presenting the interquartile range (or better still, the upper and lower quartiles separately) would be more instructive? Or a figure like figure S4, but just to represent variability in observations?

L524 I know it should be obvious but it may be worth saying "biogenic SOA yields for the simple scheme" as presumably this wasn't done for the complex scheme as the dependence is already present.

---

## Referee Comment (RC2) · Anonymous Referee #2 · 3 Jun 2019

The paper evaluates two organic aerosol modeling schemes on a global scale. The use of the results of a series of campaigns in different parts of the world is a major strength of the paper. The authors do attempt to gain insights about the ability of the model to properly simulate different processes and do manage to reach a number of useful conclusions. However, the paper has a number of serious weaknesses that need to be addressed before it can be accepted for publication.

**Use of ground measurements.** It is not clear why the authors restrict their model evaluation to airborne measurements when there are also ground measurements available in the same areas (US, Europe) for the simulated periods. This is especially problematic because they conclude that the simple modeling scheme is especially attractive for health studies. They also suggest that based on the airborne measurements both

schemes appear to overpredict OA in the lower troposphere. This comparison (at least with the IMPROVE and EUCAARI measurements) should be added to the revised paper.

**Treatment of POA.** It is not clear how the POA emissions and the POA atmospheric chemistry are treated in the two models. Some major issues:

(a) Why is the POA emission rate (based on Table 2) in the simple model 21.8 Tg/yr and in the complex model 55.4 Tg/yr? Based on the paper there is a 27 percent increase of the emissions while these numbers suggest a 150 percent increase.

(b) What is the assumed saturation concentration of the POA in the complex model? What is its enthalpy of evaporation?

**Anthropogenic SOA.** How is this simulated in the complex model? Why is so much less than in the simple model? The production of SOA from IVOCs in the complex model is not described well. There is little information about how the IVOC emissions (shown in Table 1) have been estimated and the corresponding yields used. The sentence regarding the use of naphthalene as a proxy (lines 169-171) does not clarify this issue.

**POA lifetime**: Line 306. This is quite difficult to understand and it appears to be counterintuitive. The information about the POA treatment is limited. For example, the saturation concentrations of the two products used (lines 152-153) are not given. Also the conversion of the EPOG to the OPOG is not explained and the corresponding parameters for the reactions and volatilities are missing. Finally, there is no information about the removal of the corresponding organic vapors (e.g., assumed Henry's law coefficients). The authors should explain physically how a 27 percent increase in emissions (with some of them in the gas phase) leads to a more than 50 percent increase of the burden of particulate POA.

**Lifetimes of OA components**: A wide range of global average lifetimes (from 3 to

11.5 days) is predicted. This is rather surprising for particulate matter components that should have similar size distributions. A discussion of how removal is parameterized and an explanation of these unexpected differences among different components are needed.

**Uncertainty of measurements.** The authors site a 34-38 percent uncertainty of the measurements that they use. However, this is not taken in the evaluation of the two models. Could the model errors and biases be due to the corresponding measurement errors? Could most of the $R^2$ difference from unity just be due to these errors? These issues should be addressed and taken into account in the corresponding conclusions. For example, if one of the model was perfect but the measurements had a 35 percent uncertainty what would be the $R^2$ between measurements and predictions?

**Averaging times.** The evaluation of any model does depend on the timescale investigated. This timescale appears to be 10 min. If this is the case, it should be mentioned in all graphs, tables but also in the text. An analysis at different timescales (say 1 hr and campaign average) could provide some additional insights. This would allow the authors to include all their measurements (avoid excluding the highest values).

**Model performance statistics.** The discussion is based on $R^2$ and normalized mean bias (NMB). I think that the errors (both absolute and normalized) should be included in the analysis. For example, they appear to be quite useful in the discussion related to Figure 12.

**Role of enthalpy of evaporation.** Given that a lot of the measurements were collected at relatively low temperatures the predicted partitioning of the semi-volatile OA components will depend on the assumed enthalpy of evaporation. It is not clear which values are used in the present work. The sensitivity of the conclusions of the paper to these assumed values should be discussed.

**Underestimation of OA.** Is it still appropriate to say (abstract and introduction) that models significantly underestimate observed OA concentrations in the troposphere?

The most recent studies including both models here tend to overestimate OA levels. Some additional discussion of the recent OA modeling work is needed here.

**Sources of bias.** The complex model tends to overestimate OA levels. It is not entirely clear which attributes of this model (at least compared to the simple one) are contributing to this tendency for overestimation. Is it the isoprene SOA that is included in different ways in the two models? Something else? Some additional discussion is needed.

**Sesquiterpenes.** It is not clear how the models treats SOA from sesquiterpenes. In Figure 1 this pathway is missing. In line 168 they are mentioned as a source of SOA. The yields used are surprisingly low at least for the simple model: they are equal to those of the monoterpenes, despite the much larger size of these molecules. The yields used in the complex model are not clear. The same applies to their importance as SOA precursors in these simulations.

**Organonitrates.** The treatment of organonitrates requires some additional explanation given that they are implicitly included in the SOA formed in the parameterizations of Pye et al. (2010). Do they represent additional SOA mass or do they describe a subset of the SOA predicted by the parameterizations used? Could this be contributing to the overestimations? Finally, there has been considerable work quantifying the organonitrate PM levels in Europe (Kiendler-Scharr et al., GRL, 2016) during the simulated EURAARI period that is not used here.

**Model performance in Europe.** The parameterizations used in both models have been based, to a large extent, on US field measurements. The relatively poor performance of both models in Europe during EUCAARI requires additional attention. The comparison with the ground sites would be quite helpful here. What is the source of the high predicted concentrations which are inconsistent with the measurements?

**Model behavior during the winter.** This is significant weakness of the paper given that similar modeling exercises suggest that the models have significant problems in

the winter. The authors can use ground measurements to close this gap. They could also change the title of their paper to specify that this evaluation is only for the summer and spring.

**Evaluation for CO and other gases.** Given the importance of CO for the parameterization used in the simple model, an evaluation of the ability of the performance to reproduce measured CO levels is needed. A similar evaluation for isoprene (in the appropriate regions) would be helpful.

**Characterization of the two models.** I found the characterization of the two models as simplex and complex rather misleading. Their differences are mainly in the processes that they simulate, the emissions that they use, and the parameters that they use. They are by no means a simplified version of the other. While I understand the need for names, I also think that the paper should include a table with the main differences in the two simulations. These differences should be the emissions (Table 1 is confusing right now), the processes simulated in one and not the other, the differences in the simulation of the same process (e.g., assumed effective yields for a given OA level), and the different parameters used. If the authors can attribute the differences in predictions of the two models to simplicity or complexity, this should be done carefully and should be properly supported.

**Some additional points**

Lines 42-43. This sentence is confusing.

Lines 73-74. The formation of SOA in some of these models was by no means instantaneous. The VOC precursors were required to react first.

Lines 131-132. The names of the species in the code are not needed here.

Lines 141-143. The use of the term yield is rather confusing. Does this mean for example for fires that the assumed SOAP emission is 1.3 percent of the emitted CO? Something else? In Figure 1 it appears that CO is producing SOA making this even

more confusing.

Lines 142-143. Please explain the biogenic sources discussed here. I am assuming that this means isoprene and the monoterpenes. Referring to Figure 1 could help.

Lines 147-150. Please explain what do these tracers represent and how their use allows the independent adjustment of model parameters.

Table 2. I find this table confusing. For example, the four POA emission rates do not add up to the total. It also mixes actual emission inputs with estimates from steady-state calculations. May be the information could be clearer if it was replaced by two tables.

Line 202: The 54.3 Tg/yr of POA emissions in the simple model cannot be found Table 2.

Figure S1 should probably be moved in the main paper. If needed it can replace of the figures with the vertical profiles.

The conclusion regarding the overestimation of the aqueous uptake of isoprene should be added to the abstract.

―――――――――――――――――

---

## Short Comment (SC1) · 1 Jul 2019

I have several questions for this paper and would appreciate the response from the authors.

Line 126-132 and Line 152-155: It seems to me that 'OCPI' in Simple SOA scheme and 'OPOA' in Complex SOA scheme are both called as "OPOA" in this paper. Because OCPI represents non-volatile hydrophilic OC and OPOA represents semivolatile products of SVOC oxidation, they are very different. It might be better not to confuse readers about that.

Line 155-159: OPOA seemed being classified as POA in this paper. However, in Pye et al. [2010] and related field and lab studies, OPOA is regarded as SOA. Please clarify.

[Figure]

Line 210-212: Large differences between the two schemes in Figure 2 occur in highly polluted areas like China and India. The northeastern US show similar differences. Is it possible to use surface measurements from, for example, the IMPROVE datasets to evaluate the model performance in polluted areas?

Line 346: Why are the pristine areas in the Canadian Arctic and Greenland classified as the anthropogenic regime in Figure 5?

Line 398-399: I don't think that the consistent differences are caused by inaccurate emission inventories only. How about the lacked aging processes and the different abilities to reproduce ASOA and BSOA in the two schemes?

Line 513: I am confused that in Figure S7, the Complex POA + Simple SOA simulation shows lower OA concentrations than other combinations in the anthropogenic regime. Because the Complex POA + Simple SOA double-counts the contribution from SVOC oxidation to OA, the concentrations should be overestimated.

---

## Author Comment (AC3) · 12 Aug 2019

**Response to Comment for "An evaluation of global organic aerosol schemes using airborne observations"**

Thank you for your comments. We have responded to them below (in red)

**Line 126-132 and Line 152-155:** It seems to me that 'OCPI' in Simple SOA scheme and 'OPOA' in Complex SOA scheme are both called as "OPOA" in this paper. Because OCPI represents non-volatile hydrophilic OC and OPOA represents semi-volatile products of SVOC oxidation, they are very different. It might be better not to confuse readers about that.

We have added more information to the main-text and to the supplemental material to provide greater detail on the differences between OPOA treatment in the simple and complex schemes to prevent any confusion. The simple scheme parameterization of OCPO to OCPI is intended to broadly capture the aging of primary OA and is not linked explicitly to heterogeneous chemistry or partitioning. We have thus chosen to use OPOA to refer to both 'simple OCPI' and 'complex OPOA' because they are both intended to represent aged primary OC.

**Line 155-159:** OPOA seemed being classified as POA in this paper. However, in Pye et al. [2010] and related field and lab studies, OPOA is regarded as SOA. Please clarify.

The OPOA product is formed by the oxidation of EPOA. In the simple scheme, this process is approximated by a fixed lifetime of 1 day with no direct dependence on oxidant concentrations. In the complex scheme, EPOA is oxidized with OH to form oxygenated primary organic vapors. Many previous studies in the literature have represented the aerosol formed from these vapors as Oxygenated POA (Donahue et al., 2009; Pye et al., 2010; Shrivastava et al., 2008) but the nomenclature has been the topic of some contention, with other studies preferring to use the terminology of Secondary Organic Aerosol (SOA) to represent this aerosol product (Hayes et al., 2015; Murphy et al., 2014). For the purpose of this study we have chosen to refer to aerosol resulting from the oxidation of primary organic matter that is already semi-volatile as OPOA and reserve the term SOA exclusively for aerosol formed from the oxidation of volatile organic vapors. We are further motivated to maintain the OPOA label given that this is how it is described in the GEOS-Chem model and the relevant model paper (Pye et al., 2010). We have separated the OPOA contribution and discussion whenever possible in this study to allow the reader to interpret the results as desired and have also added a detailed discussion to the SI.

**Line 210-212:** Large differences between the two schemes in Figure 2 occur in highly polluted areas like China and India. The northeastern US show similar differences. Is it possible to use surface measurements from, for example, the IMPROVE datasets to evaluate the model performance in polluted areas?

We chose to limit our analysis to aircraft observations for a few different reasons. Ground networks are only available in a few locations (US and Europe) and these networks use different instrumentation; our goal was to use consistent AMS measurements. We could have used globally distributed AMS observations, such as in Jimenez et al., 2009, however comparisons of global models with surface sites

are more susceptible to representation errors and sub-grid meteorology that are both challenging to address. We specifically designed the study to focus on the regional constraints offered by airborne measurements around the world that sample OA under a range of conditions. Since health impacts are not the focus of this work (i.e. we do not focus on surface concentrations or exposures), observational exploration throughout the full troposphere seemed best suited for exploring the OA budget. We commented on the need for airborne measurements in these regions in our conclusions.

**Line 346:** Why are the pristine areas in the Canadian Arctic and Greenland classified as the anthropogenic regime in Figure 5?

We have modified the main-text to explicitly state that our classification of anthropogenic OA (and indeed all other categories) includes both 'fresh' and 'aged' regions. With regards to the North Pacific, we track much of that pollution to east Asian emissions from China and the surrounding countries, while the North Atlantic is influenced by both European and African emissions.

**Line 398-399:** I don't think that the consistent differences are caused by inaccurate emission inventories only. How about the lacked aging processes and the different abilities to reproduce ASOA and BSOA in the two schemes?

We do not imply that these differences are exclusively due to inaccurate emissions inventories but only that they likely play an important role. We have included various other contributing factors in the main-text.

**Line 513:** I am confused that in Figure S7, the Complex POA + Simple SOA simulation shows lower OA concentrations than other combinations in the anthropogenic regime. Because the Complex POA + Simple SOA double-counts the contribution from SVOC oxidation to OA, the concentrations should be overestimated.

The complex POA consists of EPOA, OPOA and MPOA while the simple SOA consists of ASOA, ISOA, TSOA and FSOA. In the case referred to above, the OA resulting in the oxidation of SVOCs (represented in the model as primary organic gases – POGs and oxidized POGs) is only represented in the complex POA and is not included in the simple ASOA component.

---

## Author Response (AR1)

We thank the referees for their consideration of our manuscript. Below are our responses (in red) to each of the **comments (in black)**, followed by the proposed changes and additions to our revised manuscript and supplementary information (with tracked changes in response to the comments highlighted in red).

**Response to Anonymous Referee #1**

**Introduction:** I think it's worth remarking (with appropriate references) that the "simple scheme" is fairly similar to what is used in most climate and earth system models submitted to CMIP6 (NorESM, HadGEM3/UKESM1, I think also GFDL AM4), while a couple of climate models (CESM and E3SM as far as I know) have configurations that are more similar in at least some important respects to the complex scheme (such as including the volatility basis set for semi-volatile SOA).

This is a good point; we have added a statement in the introduction with an appropriate reference from the 2014 Tsigaridis model intercomparison that can point readers to various ESMs that use similar OA schemes.

**L103** "we perform a series of simulations from 2008 to 2017 using two distinct model scheme": It would be helpful to include a table summarizing the simulations that were performed (one simple, one complex, then some simulations in which the simple scheme was modified, the simple SOA with complex POA, etc.), and a more detailed explanation of which periods were simulated – just the times of the flights, or the whole year.

We have added a table summarizing the various model simulations to the supplementary information in order to provide greater detail.

**Also, at or around L223:** "The observations were averaged over the model grid-boxes and timestep." It would be great to be a bit more explicit about how the comparison was done - for example, was the model output also diagnosed and written out to a file on every timestep, or every few hours?

We have reworded the statement in Section 3 to be more explicit about the process and have also included more information on the plane-flight diagnostic in the supplementary materials

**L112** "A standard bulk aerosol scheme": which one? Also please put into context the subsequent sentence "GEOS-chem also simulates sulfate aerosol…" – is this somehow a separate issue from the bulk aerosol scheme?

We have added an appropriate reference to the aerosol scheme used in this study and have reworded the paragraph to limit confusion.

**Figure 4:** It's rather unfortunate that the differences between the two schemes are greatest in areas where you have no measurements: central Africa and inland in China. This is pointed out in the conclusions, but perhaps Africa should be added to the list on line 612 – you could also point out that there do exist some datasets that might already help with resolving the large discrepancies there (DACCIWA, ORACLES, for example)?

We have added Africa to the list and have mentioned some relevant campaigns.

"The explicit aqueous uptake mechanism for the isoprene-derived SOA products also results in substantially larger global isoprene SOA burdens (0.31 Tg) when compared to the 'pureVBS' treatment of isoprene-derived SOA that simulates an annually averaged ISOA burden of 0.12 Tg" - -so was there an additional simulation performed with the aqueous uptake turned off? Can you be a bit more detailed about what differences this causes, maybe adding another row to Table 2 where isoprene SOA is split into aqueous and non-aqueous contributions?

The isoprene SOA in the complex scheme does not include any non-aqueous contribution and is an explicit scheme that does not include any reversible partitioning. The 'pure VBS' simulation models isoprene SOA using the VBS (no explicit mechanism) as in Pye et al. (2010), and was used in the paper in order to provide context for the complex scheme results. We have updated our model description to explicitly clarify this in order to limit any confusion and have also added a separate category to the model descriptions in the SI.

**L295**. It is stated in the abstract that the model skill is superior to previous model evaluations, and in this section at line 295 the model is compared to an ensemble from Tsigaridis et al 2014. However, the reasons why the model differs from the ensemble probably vary from model to model. For the GEOS-chem model, the authors already include some comments about how the current model differs from that in Heald et al 2011 at lines 427 and 438. Can the authors identify whether it is changes to the emissions inventories since the 2011 paper, or changes to the OA schemes, that are responsible for the differences? Also, perhaps it is worth saying why some of the campaigns from Heald et al 2011 were used in the current study, but not others (presumably this was just to avoid running the simulations for unfeasibly many years)?

While it is likely that the model improvement is largely due to the combination of changes in OA schemes and emissions, it is difficult to attribute model improvement between the different versions given the large number of changes to model code and inputs without running an extensive series of sensitivity experiments with the older code. This is thus regrettably beyond the scope of the work.

We selected a list of representative campaigns to conduct this study and, given the number of simulations required per campaign and our limited computational resources, we decided to focus on campaigns with AMS observations, within the last decade, and that were publicly-available when we started this project in 2017. Text to this effect has been added to Section 3.

**L343:** The regime analysis is interesting and very useful in the following interpretation, but needs some further explanation, or possibly further tuning of the classifications, because some features of Figure 5 are rather surprising. In Figure 5, many regions that must be at least relatively pristine compared to the eastern United States are categorized as anthropogenic (large portions of the North Atlantic and North Pacific ATOM flights, much of the Canadian Arctic) and perhaps this can explain the sentence "Median concentrations over anthropogenic regions are markedly lower than those over other sources"?

Another way to make this figure 5 less surprising might be to introduce separate categories for 'remote anthropogenic' and 'polluted anthropogenic' based on another mass threshold.

We acknowledge the reviewer's concern and have modified the paragraph to explicitly state that our classification of anthropogenic OA (and indeed all other categories) includes both 'fresh' and 'aged' regions which, particularly in the case of anthropogenic SOA, explains the lower median. The regime analysis is imperfect but is intended to provide broad classification and we have found that adding additional categories over the ones currently in the study can be overwhelming while adding little to the underlying classification that is based on a relative weighting as opposed to an absolute one. With regards to the North Pacific, we track much of that pollution to east Asian emissions from China and the surrounding countries, while the North Atlantic is influenced by both European and African emissions.

Then, it looks like most of the eastern USA is classified as "remote". Is "remote" being plotted on top of "anthropogenic" for example, so the high volume of data would cause misleading results where only the last plotted regime shows up? Or is the North Atlantic characterized as anthropogenic because the aerosol mass concentration can be quite high due to a lot of dust (and the North Pacific, potentially, due to volcanic sulfur). Could other aerosol types have been included in the 'aggregate OA mass concentration' threshold of 0.2mugsm3?

As the reviewer points out, the reason there appear to be a number of remote points over the eastern US is due to the density of observations, resulting in observations in the upper troposphere that are below 0.2 µg sm$^{-3}$ (and thus, 'remote') being plotted over points lower in the troposphere. We have added a few lines to the paragraph in order to explicitly clarify this. The regime analysis has been conducted exclusively with OA concentrations and is thus not influenced by other aerosol types. The reason portions of the North Atlantic are classified as anthropogenic is because they do not meet the threshold for remote concentrations (0.2 µg sm$^{-3}$) and are composed of a minimum of 70% anthropogenic OA (from various continental sources)

The classification would presumably be quite different if the figure was remade using regime types from the complex scheme rather than the simple scheme. Perhaps this would be interesting to try, just to reproduce Figure 5 (no need to repeat the whole analysis!) It would overcome the shortcoming of the simple scheme already mentioned in the text that it tends to count (for example) anthropogenically influenced biogenic SOA as anthropogenic SOA.

In an attempt to address the reviewer's comment, we experimented with source regimes using the complex scheme, however, given the complexities involved in partitioning the POA it is difficult to separate the contribution from anthropogenic and fire sources without large structural changes to the code along with the addition of a number of new tracers (and repeating a suite of simulations).

**Figures 7 and 8** show that in the remote/marine region, the two schemes also disagree radically on whether the aerosol is primary or secondary, above 2km altitude. This is well discussed in the first paragraph of the conclusion but also seems worthy of greater emphasis and discussion in the paper around line 505. It is remarkable how well both schemes reproduce observations despite this (at least in Figure 7 and in summer, and even the lack of variability noted at line 378 does not seem to be a large effect in Figure S2). It makes sense that semivolatile POA gets to high altitude more effectively than non-volatile POA, so it stands to reason that the complex scheme is doing well. So then it seems surprising that overall the model with complex POA and simple SOA, from fig S7, seems to underestimate OA in the remote region (negative NMB)- if I understand how the NMB is defined, it should overestimate it. Similarly, the reverse arrangement, with simple POA and complex SOA, should underestimate, but the NMB is positive. What does the altitude profile look like?

There are several ways to calculate NMB – please can the authors include an equation somewhere in the text to define it?

We agree that it is surprising that the simple and complex scheme broadly track each other given the large differences in OA contributions from the different sources. We have added a short discussion on this topic. We have also included an additional section on why we chose $R^2$ and NMB as representative metrics in the main text and have included the equations for how we calculate both metrics. The simple SOA is non-volatile and is thus less sensitive to the altitude profile than the complex SOA. This results in a larger complex SOA loading at higher altitudes and leads to a larger NMB.

**In Figure 8**, ATOM-2-W shows both models substantially overestimate SOA at high altitude, while ATOM-1-W is fine. This is explained in the text as a seasonal effect, L455. Does the overestimate square with the near-perfect agreement in Figure 7 remote/marine?

The overestimated ATom points are classified as Anthropogenic, not remote. This is why the overestimate is not present in the remote comparison.

I realise this is outside the scope of the current study, but do the authors intend to make use of a fuller range of capabilities of the ToF-AMS in tracking signatures of different aerosol sources – for example signatures of biomass burning (f60), SOA (f44) etc, relevant fragment ratios, etc, etc in future work? Or even PMF factors? I'm not an instrumentation expert, but my understanding is that the ToF-AMS can provide much richer information than simply OA, sulfate, and total

mass concentrations, and this could be used in future model evaluations to great effect. It is also one reason why the observation dataset is substantially improved relative to Heald et al 2011. I think this merits a comment in the conclusions alongside the comments about the importance of additional observational constraints from new campaigns, since the expense of new observations would be much easier to justify once the existing datasets have been fully exploited.

We wholeheartedly agree with the reviewer! Unfortunately, such tracers, PMF factors, and P-ToF size distributions are typically not provided in publicly-accessible datasets. We have added a statement in the conclusions addressing this point and stating that the standardized reporting of chemical signatures from AMS data could enable further model evaluation.

Figure 6, 10, and S2: the colors are confusing compared to the AMS conventions, please use red for sulfate and green for complex OA.

We apologize for the confusion; given our use of two OA schemes, we could not use standard 'AMS green' for organics, so decided to differentiate these throughout with red and blue. As a result, we cannot use red for sulfate. However, we agree that using green could be confusing and have changed the sulfate plots to purple so as to be clearly differentiated from the AMS color scheme.

In Table 3, the standard deviation is often greater than the mean and median, yet negative concentrations aren't possible. This is clearly a matter of opinion and a pretty minor point, but maybe presenting the interquartile range (or better still, the upper and lower quartiles separately) would be more instructive? Or a figure like figure S4, but just to represent variability in observations?

In order to prevent a skew in our analysis, we have not excluded negative observations from our dataset. We chose to use the mean and median as metrics to allow for ease of interpretation but have added a figure to the SI to represent the variability in the observations.

L524 I know it should be obvious but it may be worth saying "biogenic SOA yields for the simple scheme" as presumably this wasn't done for the complex scheme as the dependence is already present.

We have made the change to clarify the statement.

**Response to Anonymous Referee #2**

**Use of ground measurements.** It is not clear why the authors restrict their model evaluation to airborne measurements when there are also ground measurements available in the same areas (US, Europe) for the simulated periods. This is especially problematic because they conclude that the simple modeling scheme is especially attractive for health studies. They also suggest that based on the airborne measurements both schemes appear to overpredict OA in the lower

troposphere. This comparison (at least with the IMPROVE and EUCAARI measurements) should be added to the revised paper.

We agree that a comparison with ground observations would provide a more constrained understanding about exposure. However, this is not the focus of this study and we chose to limit our analysis to aircraft observations for the following reasons:

1. Ground networks are only available in a few locations (US and Europe)
2. These ground networks use different instrumentation; our goal was to use consistent AMS measurements.
3. We could have used globally distributed AMS observations, such as in Jimenez et al. (2009), however comparisons of global models with surface sites are more susceptible to representation errors and sub-grid meteorology that are both challenging to address. We specifically designed the study to focus on the regional constraints offered by airborne measurements around the world that sample OA under a range of conditions.
4. Health impacts are not the focus of this work (i.e. we do not focus on surface concentrations or exposures), therefore observational exploration throughout the full troposphere seemed best suited for exploring the OA budget.

We have modified the main text by adding more details explaining our rationale.

**Treatment of POA.** It is not clear how the POA emissions and the POA atmospheric chemistry are treated in the two models. Some major issues:

(a) Why is the POA emission rate (based on Table 2) in the simple model 21.8 Tg/yr and in the complex model 55.4 Tg/yr? Based on the paper there is a 27 percent increase of the emissions while these numbers suggest a 150 percent increase.

In the simple scheme, 50% of the POA is emitted as EPOA (Emitted Primary Organic Aerosol) and 50% is emitted as OPOA (Oxygenated Primary Organic Aerosol) to approximate the near-field aging of EPOA. Total OC emissions are 31.2 TgC. Thus, given the OC:OM ratios of 1.4 and 2.1 assumed for EPOA and OPOA respectively, total POA emissions in the simple scheme are 21.8 Tg EPOA and 32.8 Tg OPOA for a total annual POA emission of 54.6 Tg. In the complex scheme, all POA is emitted as gas-phase EPOG after scaling the same inventory used in the simple scheme by 27% to account for the extra gas-phase material. Total primary emissions in the Complex Scheme are thus solely from EPOG gas phase emissions and amount to 55.4 Tg. We agree that these details can be difficult to follow and have therefore added more information to the main text (and corrected the simple emissions from 54.3 Tg yr$^{-1}$ to 54.6 Tg yr$^{-1}$). We have also included more information in the SI.

(b) What is the assumed saturation concentration of the POA in the complex model? What is its enthalpy of evaporation?

As described in Pye et al. (2010), EPOG saturation concentrations are 1646 and 20 μg m$^{-3}$ at 300 K. As EPOG ages to OPOG, the volatility of the reaction products decreases by a

factor of 100 (Pye et al., 2010; Grieshop et al., 2009). An enthalpy of vaporization of 50 kJ mol$^{-1}$ is assumed in this study. We have added these details in the SI with the appropriate references.

**Anthropogenic SOA.** How is this simulated in the complex model? Why is so much less than in the simple model? The production of SOA from IVOCs in the complex model is not described well. There is little information about how the IVOC emissions (shown in Table 1) have been estimated and the corresponding yields used. The sentence regarding the use of naphthalene as a proxy (lines 169-171) does not clarify this issue.

The anthropogenic SOA in the complex scheme is as described in Pye et al. (2010). To address the reviewer's questions, we have added more detailed information on the complex OA scheme (and the formation of SOA from IVOCs and aromatics) to the SI and have also expanded the discussion in the main text to provide greater clarity and address the discrepancy.

**POA lifetime:** Line 306. This is quite difficult to understand and it appears to be counterintuitive. The information about the POA treatment is limited. For example, the saturation concentrations of the two products used (lines 152-153) are not given. Also the conversion of the EPOG to the OPOG is not explained and the corresponding parameters for the reactions and volatilities are missing. Finally, there is no information about the removal of the corresponding organic vapors (e.g., assumed Henry's law coefficients). The authors should explain physically how a 27 percent increase in emissions (with some of them in the gas phase) leads to a more than 50 percent increase of the burden of particulate POA.

As with the previous points, we refer the reader to the original paper on this scheme (Pye et al., 2010) for a complete description. In light of the reviewer's comments, we have expanded the discussion in the main text and have included the information below in the SI:

EPOG saturation concentrations are 1646 and 20 µg m$^{-3}$ at 300 K. As EPOG reacts with OH and ages to OPOG, the volatility of the reaction products decreases by a factor of 100. The reaction rate for POG to OPOG ($K_{OH}$) is 2 x 10$^{-11}$ cm$^3$ molec$^{-1}$ s$^{-1}$

The 27% increase in emissions are all in the form of gas-phase POG (with a OM:OC of 1.4) that is hydrophobic and is advected higher in the troposphere before forming OA where it is less likely to be deposited. It is then aged to OPOG (with a OM:OC of 2.1). This sensitivity to temperature (and thus altitude) results in the formation of organic aerosol aloft, in a manner that the simple scheme cannot simulate due to its non-volatile representation, and leads to higher concentrations in the upper-troposphere where the aerosol experiences longer lifetimes.

The fraction of gas-phase OA precursors wet deposited is dictated by the liquid to gas ratio for a grid-box at any given timestep. For a soluble gas 'i', this ratio is calculated based on the following relationship:

$$\frac{C_{i,L}}{C_{i,G}} = K_i^* * L * R * T$$

where $K_i^*$ is the effective Henry's law constant that is calculated using the van't Hoff equation (Jacob et al., 2000), L is the cloud liquid water content, R is the ideal gas constant and T is the local temperature. Each organic gas-phase species has an associated Henry's law solubility constant (in M atm$^{-1}$), volatility constant (in K) and pH correction factor which is defined in the GEOS-Chem species database.

EPOG Henry's Law Solubility Constant = 9.5 M atm$^{-1}$

OPOG Henry's Law Solubility Constant = 1 x 10$^5$ M atm$^{-1}$

**Lifetimes of OA components:** A wide range of global average lifetimes (from 3 to 11.5 days) is predicted. This is rather surprising for particulate matter components that should have similar size distributions. A discussion of how removal is parameterized and an explanation of these unexpected differences among different components are needed.

We agree with the comment and were similarly surprised by the wide range in lifetimes. Closer inspection suggests that this is due to a few different factors. We have expanded our discussion in the main text and the SI to cover these points.

1. Spatial Distribution in Aerosol Types – Species emitted over marine / tropical regions experience a higher risk of being deposited via wet deposition than aerosol over more temperate regions.
2. Hydrophilic nature of the aerosol – EPOA is treated as hydrophobic, rainout is turned off and the rate constant for conversion of cloud condensate to precipitation is halved, leading to a longer lifetime against wet-deposition. Oxygenated OA (OPOA and SOA) are treated as hydrophilic and thus have shorter lifetimes.
3. Vertical Distribution – The complex scheme treats OA as semi-volatile. As a result, both POA and SOA species are often in the gas phase in the warmer parts of the troposphere and are less likely to be deposited. These gaseous vapors are advected to the upper troposphere where they then form OA aloft. These aerosol species then have a longer lifetime against deposition since they are less sensitive to meteorology in the lower troposphere.

We refer the readers to Wesely (1989), Zhang et al. (2001), Jacob et al. (2000), Liu et al. (2001) and Amos et al. (2012) for details on dry and wet deposition in GEOS-Chem and have included more details in the SI.

**Uncertainty of measurements.** The authors site a 34-38 percent uncertainty of the measurements that they use. However, this is not taken in the evaluation of the two models. Could the model errors and biases be due to the corresponding measurement errors? Could most of the R2 difference from unity just be due to these errors? These issues should be addressed and taken into account in the corresponding conclusions. For example, if one of the model was

perfect but the measurements had a 35 percent uncertainty what would be the R2 between measurements and predictions?

This is an interesting point. 33% of the modeled data-points fall within the observed uncertainty. We have added a statement to the conclusion that addresses this. If the model was perfect but you add noise (normally distributed with a mean scale factor of 1 and a SD of 0.14) to the observations, the $R^2$ is 0.97. Thus, measurement uncertainty is unlikely to be a major driving factor in the model-obs $R^2$.

**Averaging times.** The evaluation of any model does depend on the timescale investigated. This timescale appears to be 10 min. If this is the case, it should be mentioned in all graphs, tables but also in the text. An analysis at different timescales (say 1 hr and campaign average) could provide some additional insights. This would allow the authors to include all their measurements (avoid excluding the highest values).

We have added more explicit references to the 10 min timescale in the main text and referenced this in the figure captions. In addition, campaign average statistics are available in Table 2. We chose not to average over a more extended period given that the spatial variance in aircraft observations provides useful information that we prefer not to degrade.

**Model performance statistics**. The discussion is based on R2 and normalized mean bias (NMB). I think that the errors (both absolute and normalized) should be included in the analysis. For example, they appear to be quite useful in the discussion related to Figure 12.

The choice of metrics can be somewhat subjective and an initial draft of our study did include multiple metrics. However, we ultimately chose to limit ourselves to NMB and $R^2$ in order to maintain consistency and readability. Absolute differences and ratios are included in the SI (Figure S4). We found that the conclusions of our manuscript do not depend on the choice of metrics and have added a section to the paper explaining our choice of metrics along with their descriptions.

**Role of enthalpy of evaporation.** Given that a lot of the measurements were collected at relatively low temperatures the predicted partitioning of the semi-volatile OA components will depend on the assumed enthalpy of evaporation. It is not clear which values are used in the present work. The sensitivity of the conclusions of the paper to these assumed values should be discussed.

We assume an enthalpy of vaporization of 50 kJ mol$^{-1}$ and have included that information in the main text and SI.

**Underestimation of OA.** Is it still appropriate to say (abstract and introduction) that models significantly underestimate observed OA concentrations in the troposphere?

The most recent studies including both models here tend to overestimate OA levels. Some additional discussion of the recent OA modeling work is needed here.

Certain recent studies with the GEOS-Chem model (such as Marais et al. 2016) have been shown to match the magnitude of SEUS observations, but since the scheme was developed and optimized for this region, this cannot be viewed as an independent test. The simple scheme as implemented here has not been previously evaluated in the literature (the older OA scheme in GEOS-Chem used different parametrizations) but previous evaluations of OA with GEOS-Chem have demonstrated a tendency to underestimate OA at the global scale (Heald et al., 2011; Spracklen et al., 2011; Hodzic et al., 2016). Our statement is intended to convey a general historical context (particularly in the GEOS-Chem model) as opposed to serving as a blanket statement for all regions.

**Sources of bias.** The complex model tends to overestimate OA levels. It is not entirely clear which attributes of this model (at least compared to the simple one) are contributing to this tendency for overestimation. Is it the isoprene SOA that is included in different ways in the two models? Something else? Some additional discussion is needed.

The discrepancy between the simple and complex scheme is the result of a number of factors and can be attributed to different reasons depending on the location. As Figure 11 shows, the complex model overestimates in many regions, but underestimates in others, and the reasons for these biases depend on the regional factors which are discussed in the main text. In general, Figure 4 shows that the burden in the complex scheme is higher generally because of more isoprene SOA and primary OA.

**Sesquiterpenes.** It is not clear how the models treats SOA from sesquiterpenes. In Figure 1 this pathway is missing. In line 168 they are mentioned as a source of SOA. The yields used are surprisingly low at least for the simple model: they are equal to those of the monoterpenes, despite the much larger size of these molecules. The yields used in the complex model are not clear. The same applies to their importance as SOA precursors in these simulations.

Thank you for pointing out this omission. In the simple scheme sesquiterpenes are assumed to have a 10% SOA yield. In the complex scheme, they are treated using the VBS framework (described in Pye et al. 2010) and have an average yield of 42% at 298K with a $C_{OA}$ of 10 ug m$^{-3}$. In both cases, the resulting SOA is lumped with SOA from other terpenes (TSOA). We have clarified the description in the main text and added the relevant details to the SI.

**Organonitrates.** The treatment of organonitrates requires some additional explanation given that they are implicitly included in the SOA formed in the parameterizations of Pye et al. (2010). Do they represent additional SOA mass or do they describe a subset of the SOA predicted by the parameterizations used? Could this be contributing to the overestimations? Finally, there has

been considerable work quantifying the organonitrate PM levels in Europe (Kiendler-Scharr et al., GRL, 2016) during the simulated EURAARI period that is not used here.

The complex scheme uses an explicit mechanism to model the formation of organo-nitrates based on work by Fisher et al. (2016). The Pye et al. (2010) paper does not explicitly model SOA from this pathway, instead using lumped SOA yields within the VBS framework to model the SOA resulting from oxidation with $NO_3$. The simple scheme parametrization does not explicitly account for organo-nitrate formation. We have outlined the mechanisms used in this study in the Supplementary Information and also point the readers to the appropriate references for further detail.

Regarding new developments in model organo-nitrate treatment, the focus of this work is not specifically on organo-nitrates so we have not incorporated more recent developments in the field. We have added a reference to Kiendler-Scharr et al. (2016) in the discussion, suggesting that a more constrained treatment of organo-nitrates could improve model performance over Europe.

**Model performance in Europe.** The parameterizations used in both models have been based, to a large extent, on US field measurements. The relatively poor performance of both models in Europe during EUCAARI requires additional attention. The comparison with the ground sites would be quite helpful here. What is the source of the high predicted concentrations which are inconsistent with the measurements?

Both schemes largely under-predict OA concentrations over Europe, potentially due to an underestimate in terpene SOA. Unfortunately, to our knowledge there are no publicly accessible airborne AMS observations over Europe to further investigate this bias. We have expanded our discussion on this topic.

**Model behavior during the winter.** This is significant weakness of the paper given that similar modeling exercises suggest that the models have significant problems in the winter. The authors can use ground measurements to close this gap. They could also change the title of their paper to specify that this evaluation is only for the summer and spring.

Given the paucity of airborne AMS observations in the winter (with to our knowledge only one publicly accessible campaign during this season – WINTER 2015) and the computational constraints of running more simulations, as stated in our manuscript we do not feel that we can adequately characterize the full seasonal cycle. Introducing comparisons with ground observations over limited regions would skew our conclusions about wintertime OA, and would be inconsistent with the design of the study (see our response above). We have expanded on this aspect in the conclusions and have included a statement in the abstract in the interest of transparency and clarity.

**Evaluation for CO and other gases.** Given the importance of CO for the parameterization used in the simple model, an evaluation of the ability of the performance to reproduce measured CO levels is needed. A similar evaluation for isoprene (in the appropriate regions) would be helpful.

An evaluation of the model ability to reproduce gas-phase species was not the focus of this study and was deemed to be out of the scope of this analysis, particularly given that these observations were not available for all campaigns. We have provided global emissions numbers for CO, NOx, and isoprene, so that these can generally be compared to other studies.

**Characterization of the two models.** I found the characterization of the two models as simplex and complex rather misleading. Their differences are mainly in the processes that they simulate, the emissions that they use, and the parameters that they use. They are by no means a simplified version of the other. While I understand the need for names, I also think that the paper should include a table with the main differences in the two simulations. These differences should be the emissions (Table 1 is confusing right now), the processes simulated in one and not the other, the differences in the simulation of the same process (e.g., assumed effective yields for a given OA level), and the different parameters used. If the authors can attribute the differences in predictions of the two models to simplicity or complexity, this should be done carefully and should be properly supported.

Thank you for the suggestion. In order to provide clarity and limit any confusion we have added a table with the relevant details to the SI and also included a section in the main text explaining the difference in emissions.

Lines 42-43. This sentence is confusing.

We have modified the statement to limit confusion

Lines 73-74. The formation of SOA in some of these models was by no means instantaneous. The VOC precursors were required to react first.

While some types of SOA were formed via the reaction of VOC precursors, the studies we are referring to use a fixed yield approach to the formation of many types of SOA. We have modified the wording in order to ensure clarity.

Lines 131-132. The names of the species in the code are not needed here.

We include them because the use of EPOA and OPOA are not standard in the GEOS-Chem community when describing non-volatile OA, and we want to ensure that future users are clear about our definitions. Thus, in the interest of clarity, we prefer to retain this.

Lines 141-143. The use of the term yield is rather confusing. Does this mean for example for fires that the assumed SOAP emission is 1.3 percent of the emitted CO? Something else? In Figure 1 it appears that CO is producing SOA making this even more confusing.

We have modified the terminology to 'co-emission' in order to limit any confusion.

Lines 142-143. Please explain the biogenic sources discussed here. I am assuming that this means isoprene and the monoterpenes. Referring to Figure 1 could help.

We have expanded on the statement to provide more detail.

Lines 147-150. Please explain what do these tracers represent and how their use allows the independent adjustment of model parameters.

The default model only includes one SOAP and one SOAS tracer, making it impossible to differentiate the different sources. Our modified scheme separately tracers the SOAP and SOAS from fire, anthropogenic and biogenic sources in order to establish the relative contribution from each pathway.

Table 2. I find this table confusing. For example, the four POA emission rates do not add up to the total. It also mixes actual emission inputs with estimates from steady state calculations. May be the information could be clearer if it was replaced by two tables.

We have added a paragraph to the text to clarify some of the confusion about atmospherically formed vs directly emitted OA

Line 202: The 54.3 Tg/yr of POA emissions in the simple model cannot be found Table 2.

POA emissions (54.6 Tg yr$^{-1}$) are composed of direct EPOA + OPOA emissions but do not include OPOA steady state formation (which is included in the table). We have edited the main text to limit any confusion and corrected an error to change 54.3 Tg yr$^{-1}$ to 54.6 Tg yr$^{-1}$.

Figure S1 should probably be moved in the main paper. If needed it can replace of the figures with the vertical profiles.

We have moved Figure S1 into the main text. It is now Figure 7.

The conclusion regarding the overestimation of the aqueous uptake of isoprene should be added to the abstract.

We have added a statement to the abstract.

[revised manuscript text omitted]

| Simulation | Simulation Periods | Lumped Organic Aerosol Tracers | POA Treatment | SOA Treatment |
|---|---|---|---|---|
| **Simple Scheme** | | • EPOA and OPOA (Non-volatile)
• SOA
• M-EPOA and M-OPOA | Non-volatile POA. | Non-volatile SOA. Emitted with the following emission factors:
• 1.5% SOA precursor (SOAP) and 1.5% SOA from isoprene
• 5% SOAP and 5% SOA from monoterpenes and sesquiterpenes,
• 1.3% SOAP from CO emissions from fire sources
• 6.9% of CO emissions from anthropogenic combustion sources.

SOAP converts to SOA with a fixed lifetime of 1.15 days. |
| **Modified Simple Scheme** | 2007/03/01 – 2008/08/01

2009/11/01 – 2010/07/01

2011/12/01 – 2012/07/01 | • A-EPOA, A-OPOA and ASOA (Non-volatile anthropogenic OA)
• F-EPOA, F-OPOA and FSOA (Non-volatile pyrogenic OA)
• M-EPOA and M-OPOA (Non-volatile marine POA)
• ISOA (Non-volatile isoprene SOA)
• TSOA (Non-volatile terpene SOA) | 50% is emitted as fresh hydrophobic OA (EPOA) and 50% is emitted directly as aged hydrophilic OA (OPOA).

EPOA is aged to OPOA in the atmosphere with a fixed lifetime of 1 day. | The modified scheme individually simulates SOAP and SOA from each source. The default simple scheme lumps SOAP and SOA from all sources. |
| **Pure VBS Scheme** | 2012/06/01 – 2014/01/01

2013/09/01 – 2014/02/01

2014/02/01 – 2014/11/01

2015/11/01 – 2016/09/01

2016/08/01 – 2017/03/01 | • EPOA and OPOA
• ASOA (VBS anthropogenic SOA)
• ISOA (VBS isoprene SOA)
• TSOA (VBS terpene SOA)
• M-EPOA and M-OPOA (Non-volatile marine POA) | Semi-volatile. 49% is emitted as EPOG$_1$ with a saturation concentration (C*) of 1646 μg m$^{-3}$ and 51% is emitted as EPOG$_2$ with C* of 20 μg m$^{-3}$. EPOG$_1$ and EPOG$_2$ reversibly partition to EPOA$_1$ and EPOA$_2$.

EPOG$_1$ and EPOG$_2$ are aged in gas-phase via reaction with OH radical to OPOG$_1$ and OPOG$_2$ with C* of 16.46 μg m$^{-3}$ and 0.2 μg m$^{-3}$ respectively. | Gas-phase SOA precursors (aromatics, IVOCs, terpenes and isoprene) are oxidized with oxidants OH, O$_3$ to form alkyl peroxy (RO$_2$) radicals that react with either HO$_2$ or NO depending on the NOx regime.

The resulting products are classified based on the origins of their precursors into Anthropogenic SOA (ASOA), Isoprene SOA (ISOA) and Terpene SOA (TSOA), that dynamically partition between the aerosol and gas phases based on their saturation vapor pressures and ambient aerosol concentrations. Aerosol formed from intermediate volatility organic compounds (IVOCs) is modelled using naphthalene as a proxy which, when oxidized, contributes to the ASOA lumped product. |
| **Complex Scheme** | | • EPOA and OPOA (Semi-volatile)
• ASOA (VBS anthropogenic SOA)
• ISOA (Aqueous isoprene SOA)
• TSOA (VBS terpene SOA)
• OrgNit (Organic Nitrates)
• M-EPOA and M-OPOA (Non-volatile marine POA) | | The Complex scheme builds on VBS framework but replaces VBS isoprene SOA with isoprene-derived OA formed irreversibly from the aqueous phase reactive uptake of isoprene oxidation products. It also includes an explicit formation mechanism for organo-nitrates from isoprene and monoterpene oxidation pathways. |

**Table S1.** A brief description of the various simulations presented in this study

**S1. Model Sampling with the 'Planeflight Diagnostic'**

Latitude, longitude and timestamp information was extracted from the aircraft campaign data and used in conjunction with the default GEOS-Chem 'Planeflight Diagnostic' to sample the appropriate model gridbox at the appropriate spatial and temporal spot. Model transport timestep was set for 10 minute intervals and chemistry timestep was set at 20 minutes. Diagnostic output from the planeflight sampling was averaged in cases where multiple observations were conducted within the span of a single model timestep within a certain gridbox.

**S2. Organic Aerosol in the Complex Scheme**

**S2.1 Absorptive Partitioning**

The complex scheme simulates both primary and secondary OA as semi-volatile using an absorptive partitioning model (Chung and Seinfeld, 2002; Pye et al., 2010), with each class of organic compound (i) associated with a saturation vapor pressure ($C_i^*$) that determines the fraction of the tracer in both gas and aerosol phase using the following relationship:

$$C_i^* = \frac{[G_i][M_o]}{[A_i]} \qquad (S1)$$

$$[M_o] = \sum [A_i] \qquad (S2)$$

Where [Gi] and [Ai] are the concentrations of the semi-volatile i in the gas and aerosol phase respectively and [Mo] is the concentration of the particle-phase absorptive material into which the semi-volatile i can partition. The saturation vapor pressure is temperature dependent and is dynamically calculated using the following equation:

$$\frac{C_i^*(T_2)}{C_i^*(T_1)} = \frac{T_2}{T_1} \exp\left(\frac{\Delta H_i}{R}\left(\frac{1}{T_2} - \frac{1}{T_1}\right)\right) \qquad (S3)$$

An enthalpy of vaporization of 50 kJ mol$^{-1}$ is assumed to estimate C* over a range of ambient temperatures.

**S2.2 POA**

49% of POA is emitted as EPOG$_1$ with a saturation concentration (C*) of 1646 μg m$^{-3}$ and 51% is emitted as EPOG$_2$ with C* of 20 μg m$^{-3}$. EPOG$_1$ and EPOG$_2$ reversibly partition to EPOA$_1$ and EPOA$_2$. EPOG$_1$ and EPOG$_2$ are aged in gas-phase via reaction with the OH radical (k$_{OH}$ of 2 x 10$^{-11}$) to OPOG$_1$ and OPOG$_2$ with C* of 16.46 μg m$^{-3}$ and 0.2 μg m$^{-3}$ and respectively (Grieshop et al., 2009; Pye et al., 2010)

**S2.3 SOA from Aromatic VOCs and Terpenes (Pye et al., 2010)**

Gas-phase anthropogenic and select biogenic VOCs are oxidized (with oxidants - OH, O$_3$) to form alkyl peroxy (RO$_2$) radicals that then react with either HO$_2$ or NO to form second-generation aerosol products depending on the NOx regime – with high and low NOx yields and partitioning coefficients based on experimental fits from laboratory studies (See Table 1 in Pye et al., 2010). These second-generation products are assigned volatilities with C* ranging from 0.1,1,10 and 100 ug m$^{-3}$ and partition between aerosol and gas phase based on the equations listed above. This framework is referred to as the 'Volatility Basis Set' (VBS) and its implementation in the GEOS-Chem model is outlined in Pye et al. (2010). Aromatic VOCs are simulated using benzene, toluene and xylene, which are oxidized to form 4 lumped semi-volatile products. Terpenoids (monoterpenes and sesquiterpenes) are also oxidized to form 4 lumped products with C* of 0.1,1,10,100. A detailed overview of the second-generation yields can be found in Pye et al. (2010).

**S2.4 SOA from IVOCs (Pye et al., 2010)**

Intermediate Volatility Organic Compounds (IVOCs) such as alcohols and phenols have been shown to form SOA on oxidation (Chan et al., 2009; Pye et al., 2010). Phenol and substituted phenol compounds have been shown to be major contributors to IVOC emissions (Schauer et al., 2001) and exhibit similar behavior to naphthalene in terms of their aerosol yields. Thus, IVOCs are represented as a naphthalene-like surrogate (Pye et al., 2010) and assumed to form SOA in accordance with the parameters derived from the chamber studies of Chan et al. 2009. Global IVOC emissions are uncertain but are assumed to have the spatial distribution of naphthalene. For biofuel and biomass burning, naphthalene emissions are approximated using CO as a proxy, with an emission ratio of 0.0602 and 0.0701 mmol naphthalene / mol CO for biomass and biofuel burning respectively (Andreae and Merlet, 2001; Pye et al., 2010). Anthropogenic IVOC emissions are estimated from the CEDS Inventory and were scaled from benzene emissions using the same scale factors used by Pye et al. (2010).

**S2.5 Explicit Mechanism for SOA from Isoprene (Marais et al., 2016)**

Isoprene oxidation occurs through an explicit mechanism outlined in Marais et al. (2016). In this mechanism most of the isoprene undergoes oxidation via OH to form a peroxy radical which in turn reacts with $HO_2$, NO, other peroxy radicals ($RO_2$) or undergoes isomerization. The $HO_2$ reaction pathway leads to the formation of hydroxyhydroperoxides (ISOPOOH) that are oxidized by OH to isoprene epoxydiol (IEPOX) and several low-volatility products, that are represented in the model as the C5-LVOC lumped product. The high-NOx (NO) pathway results in $C_5$ hydroxy carbonyls, methyl vinyl ketone, methacrolein, and first-generation isoprene nitrates (ISOPN). The first three products react with OH to produce glyoxal (GLYX) and methylglyoxal (MGLY). ISOPN is oxidized with OH to form dihydroxy dinitrates (DHDN) and IEPOX. Reaction of the peroxy radical with $RO_2$ is a minor pathway that ultimately leads to the formation of $C_4$ hydroxyepoxides (MEPOX) as well as GLYX and MGLY. Isomerization is a similarly minor pathway that leads to the formation of a hydroperoxyaldehyde that forms GLYX and MGLY when photolyzed. IEPOX also forms GLYX and MGLY on oxidation with OH.

In addition to the processes above, isoprene also undergoes ozonolysis and reaction with $NO_3$, forming MGLY and second generation hydroxy-nitrates (NT-ISOPN). IEPOX, GLYX, MGLY, C5-LVOC, MEPOX, ISOPN, DHDN, NT-ISOPN form non-volatile aerosols through an irreversible aqueous reactive uptake parametrization. A more detailed overview of the relevant mechanism, yields, reaction rates, branching ratios and uptake coefficients can be found in Marais et al. (2016).

**S2.6 Explicit Mechanism for Organo-nitrates from Terpenes (Fisher et al., 2016)**

Terpene species also form aerosol-phase organo-nitrates through an explicit mechanism defined in Fisher et al. (2016). During the day, terpene precursors react with OH to form peroxy radicals which then react with NO to form first generation monoterpene nitrates with a yield of 18%. These are then further oxidized to form second-generation monoterpene nitrates. At night, these terpenes react with $NO_3$ to form nitrooxy peroxy radicals that either decompose or form a more stable organo-nitrate with a predefined branching ratio based on the precursor. Formation of non-volatile aerosol from gas-phase organo-nitrate is modelled using an irreversible reactive uptake parameterization,

followed by particle-phase hydrolysis. A more detailed overview of the relevant mechanism, yields, reaction rates, branching ratios and uptake coefficients can be found in Fisher et al. (2016).

**S3. OA Loss Processes: Dry and Wet Deposition**

Organic Aerosol is deposited from the atmosphere through both wet deposition and dry deposition. Dry deposition is estimated using a parametrization described in Zhang et al. (2001) that calculates particle deposition velocities as a function of particle size, density and relevant meteorology and accounts for turbulent transfer, Brownian diffusion, impaction, interception, gravitational settling and particle rebound. Particle diameter and density is assumed to be 0.5 μm and 1500 kg m$^{-3}$ respectively. Deposition velocities are calculated using the following relationship:

$$V_d = V_g + \frac{1}{(R_a + R_s)}$$
(S4)

where $V_g$ is the gravitational settling velocity, $R_a$ is the aerodynamic resistance above the canopy and $R_s$ is the surface resistance. A more detailed derivation of the individual terms can be found in Section 2 of Zhang et al. (2001).

Wet deposition occurs through two processes – 'Rainout' defined by in-cloud scavenging and 'Washout' defined by below-cloud scavenging. Rainout scavenges aerosols efficiently and is sensitive to the fraction of the grid-box that experiences precipitation. This fraction is calculated online using the grid-scale precipitation formation rate ($Q_k$), cloud condensed water content (L), the duration of the model timestep, the duration of precipitation over the time step ($T_c$) and rate constant for conversion of cloud water to precipitation ($C_1$). See Liu et al. (2001) for more details. Below-cloud scavenging is calculated using a washout rate applied to the precipitation fraction described above. The model also simulates the release of aerosol during the re-evaporation of precipitation as it falls to the ground. Scavenging of aerosols is also modelled from cloud updrafts in moist convection and the fraction of aerosol tracer scavenged by the convective precipitation in the updraft is defined by the following relationship:

$$Conv_{frac} = 1 - e^{-\alpha \Delta z}$$
(S5)

where $\Delta z$ is the thickness of the convective column and $\alpha$ is the scavenging efficiency.

The fraction of gas-phase OA precursors wet deposited is dictated by the liquid to gas ratio for a grid-box at any given timestep. For a soluble gas 'i', this ratio is calculated based on the following relationship:

$$\frac{C_{i,L}}{C_{i,G}} = K_i^* * L * R * T$$
(S6)

where $K_i^*$ is the effective Henry's law constant that is calculated using the van't Hoff equation (Jacob et al., 2000), L is the cloud liquid water content, R is the ideal gas constant and T is the local temperature. Each organic gas-phase species has an associated Henry's law solubility constant (in M atm$^{-1}$), volatility constant (in K) and pH correction factor which is defined in the GEOS-Chem species database. A detailed overview of the wet deposition scheme can be found in Jacob et al. (2000), Liu et al. (2001) and Amos et al. (2012).

**S4. Nomenclature: Oxygenated Primary Organic Aerosol (OPOA) vs Secondary Organic Aerosol (SOA)**

The OPOA product is formed by the oxidation of EPOA. In the simple scheme, this process is approximated by a fixed lifetime of 1 day with no direct dependence on oxidant concentrations. In the complex scheme, EPOA is oxidized with OH to form oxygenated primary organic vapors. Many previous studies in the literature have represented the aerosol formed from these vapors as Oxygenated POA (Donahue et al., 2009; Pye et al., 2010; Shrivastava et al., 2008) but the nomenclature has been the topic of some contention, with other studies preferring to use the terminology of Secondary Organic Aerosol (SOA) to represent this aerosol product (Hayes et al., 2015; Murphy et al., 2014). For the purpose of this study we have chosen to refer to aerosol resulting from the oxidation of primary organic matter that is already semi-volatile as OPOA and reserve the term SOA exclusively for aerosol formed from the oxidation of volatile organic vapors. We are further motivated to maintain these labels given that this is how they are described in the GEOS-Chem model and the relevant model paper (Pye et al., 2010). We have separated the OPOA contribution and discussion whenever possible in this study to allow the reader to interpret the results as desired.

| Campaign | Organic Aerosol | NO$_x$ | Isoprene | CO |
|---|---|---|---|---|
| ARCPAC | C-ToF-AMS (A.M. Middlebrook) | NOAA NO$_y$O$_3$ (T.B. Ryerson) | PTR-MS (J.A. de Gouw, C. Warneke) | VUV Resonance Fluorescence (J.S. Holloway) |
| ARCTAS | HR-ToF-AMS (J.L. Jimenez) | NCAR 4 channel Chemiluminescence (A.J. Weinheimer, F.M. Flocke, D.J. Knapp, D.D. Montzka, I.B. Pollack) | TOGA (E. Apel, R. Hornbrook) | DACOM (G.S. Diskin, G. Sachse) |
| EUCAARI | C-ToF-AMS (H. Coe) | | | |
| OP3 | C-ToF-AMS (H. Coe) | | | |
| CalNex | C-ToF-AMS (A.M. Middlebrook) | NOAA NO$_y$O$_3$ (T.B. Ryerson, I.B. Pollack) | PTR-MS (J.A. de Gouw, C. Warneke) | VUV Resonance Fluorescence (J.S. Holloway) |
| DC3 | HR-ToF-AMS (J.L. Jimenez) | NOAA NO$_y$O$_3$ (T.B. Ryerson, I.B. Pollack) | PTR-MS (T. Mikoviny, A. Wisthaler) | DACOM (G.S. Diskin, G Sachse) |
| SENEX | C-ToF-AMS (A.M. Middlebrook) | NOAA NO$_y$O$_3$ (T.B. Ryerson, I.B. Pollack) | PTR-MS (M. Graus) | VUV Resonance Fluorescence (J.S. Holloway) |
| SEAC4RS | HR-ToF-AMS (J.L. Jimenez) | NOAA NO$_y$O$_3$ (T.B. Ryerson, I.B. Pollack, J. Peischl) | WAS (D.R. Blake) | DACOM (G.S. Diskin, G. Sachse) |
| GoAmazon | HR-ToF-AMS (J.E. Shilling) | | PTR-MS (J.E. Shilling) | Los Gatos ICOS Analyzer (S.R. Springston) |
| FRAPPE | C-ToF-mAMS (R. Bahreini) | NCAR 2-channel Chemiluminescence (A.J. Weinheimer, D.D. Montzka) | TOGA (E. Apel, R. Hornbrook) | Aero-Laser VUV Fluorescence (T.L. Campos and F.M. Flocke) |
| KORUS-AQ | HR-AMS (J.L. Jimenez, P. Campuzano-Jost) | NCAR 4-channel Chemiluminescence (A.J. Weinheimer, D.D. Montzka) | PTR-MS (P. Eichler, L. Kaser, T. Mikoviny, M. Müller, A. Wisthaler) | DACOM (G.S. Diskin, S.E. Pusede) |
| ATom | HR-ToF-AMS (J.L. Jimenez) | NOAA NO$_y$O$_3$ (T.B. Ryerson, J. Peischl, C. Thompson) | TOGA (E. Apel, R. Hornbrook) | QCLS (B.C. Daube, S.C. Wofsy, R. Commane, E. Kort) |

**Table S2**. An overview of the instrumentation and associated primary investigators for the organic aerosol and trace gas observations used in this analysis.

Nitrogen oxides were measured using photolysis rates and NO/O$_3$ chemiluminescence techniques (Ryerson et al., 2000), carbon monoxide levels were measured using a Differential Absorption Carbon monOxide Measurement (DACOM) instrument (Sachse et al., 1987) or a VUV resonance fluorescence approach (Gerbig et al., 1999), isoprene concentrations were observed using a Proton Transfer Reaction Mass Spectrometer (de Gouw and Warneke, 2007), a Trace Organic Gas Analyzer (Apel et al., 2010) or a whole air sampling approach (Colman et al., 2001) and sulfate aerosol loadings were measured using an AMS.

| Regime | Description | Percentage of Dataset | Mean OA | Median OA | Std. Dev. OA | Mean Isoprene | Mean NO$_x$ | Mean CO |
|--------|-------------|----------------------|---------|-----------|--------------|---------------|-------------|---------|
| A | Dominant anthropogenic Influence | 39.1% | 1.9 | 0.6 | 3.2 | 0.05 | 0.96 | 144 |
| F | Dominant pyrogenic Influence | 7.3% | 4.5 | 2.7 | 5.3 | 0.13 | 0.17 | 151 |
| B | Dominant biogenic Influence | 3.6% | 3.1 | 2.6 | 2.6 | 1.46 | 0.16 | 122 |
| AF | Anthropogenic and Pyrogenic Influence | 6.8% | 3.8 | 1.6 | 5.0 | 0.05 | 0.80 | 160 |
| AB | Anthropogenic and Biogenic Influence | 14.0% | 4.1 | 2.7 | 4.0 | 0.60 | 0.35 | 115 |
| AFB | No dominant influence from any one source category. | 10.1% | 3.2 | 2.5 | 2.9 | 0.10 | 0.38 | 115 |
| R | Remote / Clean (concentrations under 0.2 μg / sm³) | 19.1% | 0.1 | 0.1 | 0.3 | 0.05 | 0.08 | 71 |
| Aggregate | | | 2.4 | 0.7 | 3.6 | 0.24 | 0.55 | 126 |

**Table S3.** An overview of the different regimes. Statistics (mean, median, standard deviation) are listed for the observational data categorized into the individual regimes. OA data is in units of μg sm$^{-3}$. Mean observations for isoprene, nitrogen oxides and carbon monoxide are in units of parts per billion (ppb).

[Figure]

**Figure S1.** Distribution in the observed organic aerosol concentrations for each campaign. The boxes denote the 25th and 75th percentile of the distribution, while the whiskers denote the 5th and 95th percentile. Observations represented here have been filtered and averaged to the model timestep. Refer to Section 3 for more details.

[Figure]

**Figure S2**. Mean vertical profiles (in km) for the observed and simulated OA and sulfate across the different regimes. The profiles are binned at 200m intervals. Observations are in black. For the OA, the complex scheme is in red while the simple scheme is in blue. Model sulfate is in purple.

[Figure]

**Figure S3**. A comparison of the simulated OA loadings averaged by grid-box over the vertical dimension. Panel (d) provides an overview of the column-averaged 'best fit' scheme based on the ability to minimize the mean bias.

[Figure]

**Figure S4.** Distribution in the ratio and bias between the observed and modelled organic aerosol concentrations for each model scheme across the 17 campaigns. The boxes denote the 25[th] and 75[th] percentile of the distribution, while the whiskers denote the 5[th] and 95[th] percentile. The ratio plots have been overlaid with violin plots describing the entire distribution.

[Figure]

**Figure S5.** Comparison of complex (red), simple (blue) and observed (grey) organic aerosol to carbon monoxide.

[Figure]

**Figure S6**. A comparison of model-observation OA bias and binned observations for a) relative humidity, b) Temperature, c) Sulfate, d) Isoprene, e) CO and f) NOₓ for the complex (red) and simple (blue) schemes across the aggregate dataset. The best fit line is shown in black.

[Figure]

**Figure S7.** A statistical evaluation of the OA model skill for the complex and simple schemes against a modified treatment that interchanges the POA and SOA from both schemes.

---

## Author Response (AR3)

We thank the referees for their consideration of our manuscript. Below are our responses (in red) to each of the **comments (in black)**, followed by the proposed changes and additions to our revised manuscript and supplementary information (with tracked changes in response to the comments highlighted in red).

**Response to the Editor**

I agree with Referee 2 that based on the decision to exclude ground-based measurements, the authors should "delete the references to health in the present paper and especially their conclusion that the simple scheme is an attractive tool for use in health studies". With these changes I will be happy to accept the manuscript for publication.

In order to address the concerns raised by the reviewer, we have removed the reference to any applicability to health studies in the manuscript and have also included a statement stressing that this study does not leverage any surface observations. We have also addressed the other comments and have outlined our changes below.

**Response to Anonymous Referee #1**

Figure 4 shows very large differences between complex and simple schemes in China, and I did not find a dedicated discussion of the reasons for this. Yet the complex and simple schemes are well correlated, and both overestimate OA, in the outflow region (if we assume Korea is the outflow region). At line 626 you might expand on your existing comments, perhaps referring specifically to this and emphasising the large uncertainties associated with IVOCs which some authors suggest could be responsible for more SOA than the Pye and Seinfeld (2010) treatment would suggest – for example, see Zhao et al, Sci Rep 2016.

Thank you for drawing attention to this point; we have added a statement at line 444 that addresses this uncertainty.

Figure S1, S4 please say in the caption what the colors represent, or remove them.

We have added a statement clarifying that the colors represent the different schemes

Fig 6,10,S2 I still think the authors should use red for sulfate and other colors which do not form part of the AMS scheme, such as purple, for their various organic schemes.

We have updated the color scheme throughout the manuscript and now use dark green for the complex scheme, light green for the simple scheme and red for sulfate, consistent with the AMS color scheme.

Initially I found the sentence "ISOA and Org-Nit are generated exclusively through the aqueous uptake pathway and do not include any 'non-aqueous' OA" confusing, because I read 'aqueous uptake' to mean 'uptake by cloud water, then cloud processing'. And Fisher et al use the term 'uptake' but not 'aqueous uptake'. Of course there is no cloud processing here– the pH of the droplets considered by Marais et al more or less precludes them being activated. But it would be good to make this clearer in the text.

We have updated the text at line 187 to explicitly state that our study does not include cloud processing of SOA.

The lab study of Brégonzio-Rozier et al (ACP 2016) cited by Marais et al as a justification for not considering aqueous formation of SOA from isoprene via cloud processing found a very small mass yield of in-cloud isoprene SOA. However, if I interpret their Figure 1 correctly, they did observe a strong enhancement compared to conditions that were first very dry, and then at 80% RH, suggesting that cloud processing could be important compared to 'aqueous uptake' in some conditions -I think it's too early to tell. I realise cloud-processing of SOA is still hugely uncertain and hard to parameterize, but perhaps the authors might briefly mention and discuss the body of research on in-cloud SOA? In-cloud SOA would presumably correlate with aqueous uptake, so I guess it is possible that when the authors find improved performance by replacing VBS with aqueous uptake for isoprene, they are in fact compensating, to some extent, for in-cloud production?

We have included a short discussion on cloud processing in the context of our study at line 554.

In the SI, it would be good to clarify that the C5-LVOC lumped product, despite its name, still doesn't partition via the VBS, instead via aqueous uptake. This is clear from the text further down, as C5-LVOC features in the long list of stuff that forms non-volatile aerosols irreversibly, but nevertheless, retaining the name C5-LVOC is potentially confusing.

We have updated the SI to explicitly reflect that the C5-LVOC product is non-volatile

Would it be possible to include a version of Figure 5 which excludes the upper troposphere as a supplementary figure, so that the surprising classification of the southeastern US as remote can be further clarified, and the different altitudes can be compared?

We have included an additional figure in the SI (Figure S2) that displays an altitude differentiated spatial mapping of the different regimes in order to address this point.

**Response to Anonymous Referee #2**

Based on their response to my first comment they should delete the references to health in the present paper and especially their conclusion that the simple scheme is an attractive tool for use in health studies. They should actually stress in the paper that their study focuses on the utility of these modeling schemes for explorations of the global OA budget. They should also caution the reader that their conclusions may not be applicable to ground-level OA concentrations.

In order to address these concerns, we have removed the reference to any applicability to health studies in the manuscript and have also included a statement stressing that this study does not leverage any surface observations.

[revised manuscript text omitted]

| Simulation | Simulation Periods | Lumped Organic Aerosol Tracers | POA Treatment | SOA Treatment |
|---|---|---|---|---|
| **Simple Scheme** | | • EPOA and OPOA (Non-volatile)
• SOA
• M-EPOA and M-OPOA | Non-volatile POA. | Non-volatile SOA. Emitted with the following emission factors:
• 1.5% SOA precursor (SOAP) and 1.5% SOA from isoprene
• 5% SOAP and 5% SOA from monoterpenes and sesquiterpenes,
• 1.3% SOAP from CO emissions from fire sources
• 6.9% of CO emissions from anthropogenic combustion sources.

SOAP converts to SOA with a fixed lifetime of 1.15 days.

The modified scheme individually simulates SOAP and SOA from each source. The default simple scheme lumps SOAP and SOA from all sources. |
| **Modified Simple Scheme** | 2007/03/01 – 2008/08/01

2009/11/01 – 2010/07/01

2011/12/01 – 2012/07/01 | • A-EPOA, A-OPOA and ASOA (Non-volatile anthropogenic OA)
• F-EPOA, F-OPOA and FSOA (Non-volatile pyrogenic OA)
• M-EPOA and M-OPOA (Non-volatile marine POA)
• ISOA (Non-volatile isoprene SOA)
• TSOA (Non-volatile terpene SOA) | 50% is emitted as fresh hydrophobic OA (EPOA) and 50% is emitted directly as aged hydrophilic OA (OPOA).

EPOA is aged to OPOA in the atmosphere with a fixed lifetime of 1 day. | |
| **Pure VBS Scheme** | 2012/06/01 – 2014/01/01

2013/09/01 – 2014/02/01

2014/02/01 – 2014/11/01

2015/11/01 – 2016/09/01

2016/08/01 – 2017/03/01 | • EPOA and OPOA
• ASOA (VBS anthropogenic SOA)
• ISOA (VBS isoprene SOA)
• TSOA (VBS terpene SOA)
• M-EPOA and M-OPOA (Non-volatile marine POA) | Semi-volatile. 49% is emitted as $EPOG_1$ with a saturation concentration (C\*) of 1646 µg m$^{-3}$ and 51% is emitted as $EPOG_2$ with C\* of 20 µg m$^{-3}$. $EPOG_1$ and $EPOG_2$ reversibly partition to $EPOA_1$ and $EPOA_2$.

$EPOG_1$ and $EPOG_2$ are aged in gas-phase via reaction with OH radical to $OPOG_1$ and $OPOG_2$ with C\* of 16.46 µg m$^{-3}$ and 0.2 µg m$^{-3}$ respectively. | Gas-phase SOA precursors (aromatics, IVOCs, terpenes and isoprene) are oxidized with oxidants OH, $O_3$ to form alkyl peroxy ($RO_2$) radicals that react with either $HO_2$ or NO depending on the NOx regime.

The resulting products are classified based on the origins of their precursors into Anthropogenic SOA (ASOA), Isoprene SOA (ISOA) and Terpene SOA (TSOA), that dynamically partition between the aerosol and gas phases based on their saturation vapor pressures and ambient aerosol concentrations. Aerosol formed from intermediate volatility organic compounds (IVOCs) is modelled using naphthalene as a proxy which, when oxidized, contributes to the ASOA lumped product. |
| **Complex Scheme** | | • EPOA and OPOA (Semi-volatile)
• ASOA (VBS anthropogenic SOA)
• ISOA (Aqueous isoprene SOA)
• TSOA (VBS terpene SOA)
• OrgNit (Organic Nitrates)
• M-EPOA and M-OPOA (Non-volatile marine POA) | | The Complex scheme builds on VBS framework but replaces VBS isoprene SOA with isoprene-derived OA formed irreversibly from the aqueous phase reactive uptake of isoprene oxidation products. It also includes an explicit formation mechanism for organo-nitrates from isoprene and monoterpene oxidation pathways. |

**Table S1**. A brief description of the various simulations presented in this study

**S1. Model Sampling with the 'Planeflight Diagnostic'**

Latitude, longitude and timestamp information was extracted from the aircraft campaign data and used in conjunction with the default GEOS-Chem 'Planeflight Diagnostic' to sample the appropriate model gridbox at the appropriate spatial and temporal spot. Model transport timestep was set for 10 minute intervals and chemistry timestep was set at 20 minutes. Diagnostic output from the planeflight sampling was averaged in cases where multiple observations were conducted within the span of a single model timestep within a certain gridbox.

**S2. Organic Aerosol in the Complex Scheme**

**S2.1 Absorptive Partitioning**

The complex scheme simulates both primary and secondary OA as semi-volatile using an absorptive partitioning model (Chung and Seinfeld, 2002; Pye et al., 2010), with each class of organic compound (i) associated with a saturation vapor pressure ($C_i^*$) that determines the fraction of the tracer in both gas and aerosol phase using the following relationship:

$$C_i^* = \frac{[G_i][M_o]}{[A_i]} \tag{S1}$$

$$[M_o] = \sum [A_i] \tag{S2}$$

Where [Gi] and [Ai] are the concentrations of the semi-volatile i in the gas and aerosol phase respectively and [Mo] is the concentration of the particle-phase absorptive material into which the semi-volatile i can partition. The saturation vapor pressure is temperature dependent and is dynamically calculated using the following equation:

$$\frac{C_i^*(T_2)}{C_i^*(T_1)} = \frac{T_2}{T_1} \exp\left(\frac{\Delta H_i}{R}\left(\frac{1}{T_2} - \frac{1}{T_1}\right)\right) \tag{S3}$$

An enthalpy of vaporization of 50 kJ mol$^{-1}$ is assumed to estimate C* over a range of ambient temperatures.

**S2.2 POA**

49% of POA is emitted as EPOG$_1$ with a saturation concentration (C*) of 1646 µg m$^{-3}$ and 51% is emitted as EPOG$_2$ with C* of 20 µg m$^{-3}$. EPOG$_1$ and EPOG$_2$ reversibly partition to EPOA$_1$ and EPOA$_2$. EPOG$_1$ and EPOG$_2$ are aged in gas-phase via reaction with the OH radical ($k_{OH}$ of 2 x 10$^{-11}$) to OPOG$_1$ and OPOG$_2$ with C* of 16.46 µg m$^{-3}$ and 0.2 µg m$^{-3}$ and respectively (Grieshop et al., 2009; Pye et al., 2010)

**S2.3 SOA from Aromatic VOCs and Terpenes (Pye et al., 2010)**

Gas-phase anthropogenic and select biogenic VOCs are oxidized (with oxidants - OH, O$_3$) to form alkyl peroxy (RO$_2$) radicals that then react with either HO$_2$ or NO to form second-generation aerosol products depending on the NOx regime – with high and low NOx yields and partitioning coefficients based on experimental fits from laboratory studies (See Table 1 in Pye et al., 2010). These second-generation products are assigned volatilities with C* ranging from 0.1,1,10 and 100 ug m$^{-3}$ and partition between aerosol and gas phase based on the equations listed above. This framework is referred to as the 'Volatility Basis Set' (VBS) and its implementation in the GEOS-Chem model is outlined in Pye et al. (2010). Aromatic VOCs are simulated using benzene, toluene and xylene, which are oxidized to form 4 lumped semi-volatile products. Terpenoids (monoterpenes and sesquiterpenes) are also oxidized to form 4 lumped products with C* of 0.1,1,10,100. A detailed overview of the second-generation yields can be found in Pye et al. (2010).

**S2.4 SOA from IVOCs (Pye et al., 2010)**

Intermediate Volatility Organic Compounds (IVOCs) such as alcohols and phenols have been shown to form SOA on oxidation (Chan et al., 2009; Pye et al., 2010). Phenol and substituted phenol compounds have been shown to be major contributors to IVOC emissions (Schauer et al., 2001) and exhibit similar behavior to naphthalene in terms of their aerosol yields. Thus, IVOCs are represented as a naphthalene-like surrogate (Pye et al., 2010) and assumed to form SOA in accordance with the parameters derived from the chamber studies of Chan et al. 2009. Global IVOC emissions are uncertain but are assumed to have the spatial distribution of naphthalene. For biofuel and biomass burning, naphthalene emissions are approximated using CO as a proxy, with an emission ratio of 0.0602 and 0.0701 mmol naphthalene / mol CO for biomass and biofuel burning respectively (Andreae and Merlet, 2001; Pye et al., 2010). Anthropogenic IVOC emissions are estimated from the CEDS Inventory and were scaled from benzene emissions using the same scale factors used by Pye et al. (2010).

**S2.5 Explicit Mechanism for SOA from Isoprene (Marais et al., 2016)**

Isoprene oxidation occurs through an explicit mechanism outlined in Marais et al. (2016). In this mechanism most of the isoprene undergoes oxidation via OH to form a peroxy radical which in turn reacts with $HO_2$, NO, other peroxy radicals ($RO_2$) or undergoes isomerization. The $HO_2$ reaction pathway leads to the formation of hydroxyhydroperoxides (ISOPOOH) that are oxidized by OH to isoprene epoxydiol (IEPOX) and several low-volatility products, that are represented in the model as the C5-LVOC lumped product which, despite its name is assumed to be non-volatile. The high-NOx (NO) pathway results in $C_5$ hydroxy carbonyls, methyl vinyl ketone, methacrolein, and first-generation isoprene nitrates (ISOPN). The first three products react with OH to produce glyoxal (GLYX) and methylglyoxal (MGLY). ISOPN is oxidized with OH to form dihydroxy dinitrates (DHDN) and IEPOX. Reaction of the peroxy radical with $RO_2$ is a minor pathway that ultimately leads to the formation of $C_4$ hydroxyepoxides (MEPOX) as well as GLYX and MGLY. Isomerization is a similarly minor pathway that leads to the formation of a hydroperoxyaldehyde that forms GLYX and MGLY when photolyzed. IEPOX also forms GLYX and MGLY on oxidation with OH.

In addition to the processes above, isoprene also undergoes ozonolysis and reaction with $NO_3$, forming MGLY and second generation hydroxy-nitrates (NT-ISOPN). IEPOX, GLYX, MGLY, C5-LVOC, MEPOX, ISOPN, DHDN, NT-ISOPN form non-volatile aerosols through an irreversible aqueous reactive uptake parametrization. A more detailed overview of the relevant mechanism, yields, reaction rates, branching ratios and uptake coefficients can be found in Marais et al. (2016).

**S2.6 Explicit Mechanism for Organo-nitrates from Terpenes (Fisher et al., 2016)**

Terpene species also form aerosol-phase organo-nitrates through an explicit mechanism defined in Fisher et al. (2016). During the day, terpene precursors react with OH to form peroxy radicals which then react with NO to form first generation monoterpene nitrates with a yield of 18%. These are then further oxidized to form second-generation monoterpene nitrates. At night, these terpenes react with $NO_3$ to form nitrooxy peroxy radicals that either decompose or form a more stable organo-nitrate with a predefined branching ratio based on the precursor. Formation of nonvolatile aerosol from gas-phase organo-nitrate is modelled using an irreversible reactive uptake parameterization, followed by particle-phase hydrolysis. A more detailed overview of the relevant mechanism, yields, reaction rates, branching ratios and uptake coefficients can be found in Fisher et al. (2016).

**S3. OA Loss Processes: Dry and Wet Deposition**

Organic Aerosol is deposited from the atmosphere through both wet deposition and dry deposition. Dry deposition is estimated using a parametrization described in Zhang et al. (2001) that calculates particle deposition velocities as a function of particle size, density and relevant meteorology and accounts for turbulent transfer, Brownian diffusion, impaction, interception, gravitational settling and particle rebound. Particle diameter and density is assumed to be 0.5 μm and 1500 kg m$^{-3}$ respectively. Deposition velocities are calculated using the following relationship:

$$V_d = V_g + \frac{1}{(R_a + R_s)}$$

(S4)

where $V_g$ is the gravitational settling velocity, $R_a$ is the aerodynamic resistance above the canopy and $R_s$ is the surface resistance. A more detailed derivation of the individual terms can be found in Section 2 of Zhang et al. (2001).

Wet deposition occurs through two processes – 'Rainout' defined by in-cloud scavenging and 'Washout' defined by below-cloud scavenging. Rainout scavenges aerosols efficiently and is sensitive to the fraction of the grid-box that experiences precipitation. This fraction is calculated online using the grid-scale precipitation formation rate ($Q_k$), cloud condensed water content (L), the duration of the model timestep, the duration of precipitation over the time step ($T_c$) and rate constant for conversion of cloud water to precipitation ($C_1$). See Liu et al. (2001) for more details. Below-cloud scavenging is calculated using a washout rate applied to the precipitation fraction described above. The model also simulates the release of aerosol during the re-evaporation of precipitation as it falls to the ground. Scavenging of aerosols is also modelled from cloud updrafts in moist convection and the fraction of aerosol tracer scavenged by the convective precipitation in the updraft is defined by the following relationship:

$$Conv_{frac} = 1 - e^{-\alpha \Delta z}$$

(S5)

where $\Delta z$ is the thickness of the convective column and $\alpha$ is the scavenging efficiency.

The fraction of gas-phase OA precursors wet deposited is dictated by the liquid to gas ratio for a grid-box at any given timestep. For a soluble gas 'i', this ratio is calculated based on the following relationship:

$$\frac{C_{i,L}}{C_{i,G}} = K_i^* * L * R * T$$

(S6)

where $K_i^*$ is the effective Henry's law constant that is calculated using the van't Hoff equation (Jacob et al., 2000), L is the cloud liquid water content, R is the ideal gas constant and T is the local temperature. Each organic gas-phase species has an associated Henry's law solubility constant (in M atm$^{-1}$), volatility constant (in K) and pH correction factor which is defined in the GEOS-Chem species database. A detailed overview of the wet deposition scheme can be found in Jacob et al. (2000), Liu et al. (2001) and Amos et al. (2012).

**S4. Nomenclature: Oxygenated Primary Organic Aerosol (OPOA) vs Secondary Organic Aerosol (SOA)**

The OPOA product is formed by the oxidation of EPOA. In the simple scheme, this process is approximated by a fixed lifetime of 1 day with no direct dependence on oxidant concentrations. In the complex scheme, EPOA is oxidized with OH to form oxygenated primary organic vapors. Many previous studies in the literature have represented the aerosol formed from these vapors as Oxygenated POA (Donahue et al., 2009; Pye et al., 2010; Shrivastava et al., 2008) but the nomenclature has been the topic of some contention, with other studies preferring to use the terminology of Secondary Organic Aerosol (SOA) to represent this aerosol product (Hayes et al., 2015; Murphy et al., 2014). For the purpose of this study we have chosen to refer to aerosol resulting from the oxidation of primary organic matter that is already semi-volatile as OPOA and reserve the term SOA exclusively for aerosol formed from the oxidation of volatile organic vapors. We are further motivated to maintain these labels given that this is how they are described in the GEOS-Chem model and the relevant model paper (Pye et al., 2010). We have separated the OPOA contribution and discussion whenever possible in this study to allow the reader to interpret the results as desired.

| Campaign | Organic Aerosol | NOx | Isoprene | CO |
|---|---|---|---|---|
| ARCPAC | C-ToF-AMS (A.M. Middlebrook) | NOAA NO$_y$O$_3$ (T.B. Ryerson) | PTR-MS (J.A. de Gouw, C. Warneke) | VUV Resonance Fluorescence (J.S. Holloway) |
| ARCTAS | HR-ToF-AMS (J.L. Jimenez) | NCAR 4 channel Chemiluminescence (A.J. Weinheimer, F.M. Flocke, D.J. Knapp, D.D. Montzka, I.B. Pollack) | TOGA (E. Apel, R. Hornbrook) | DACOM (G.S. Diskin, G. Sachse) |
| EUCAARI | C-ToF-AMS (H. Coe) | | | |
| OP3 | C-ToF-AMS (H. Coe) | | | |
| CalNex | C-ToF-AMS (A.M. Middlebrook) | NOAA NO$_y$O$_3$ (T.B. Ryerson, I.B. Pollack) | PTR-MS (J.A. de Gouw, C. Warneke) | VUV Resonance Fluorescence (J.S. Holloway) |
| DC3 | HR-ToF-AMS (J.L. Jimenez) | NOAA NO$_y$O$_3$ (T.B. Ryerson, I.B. Pollack) | PTR-MS (T. Mikoviny, A. Wisthaler) | DACOM (G.S. Diskin, G Sachse) |
| SENEX | C-ToF-AMS (A.M. Middlebrook) | NOAA NO$_y$O$_3$ (T.B. Ryerson, I.B. Pollack) | PTR-MS (M. Graus) | VUV Resonance Fluorescence (J.S. Holloway) |
| SEAC4RS | HR-ToF-AMS (J.L. Jimenez) | NOAA NO$_y$O$_3$ (T.B. Ryerson, I.B. Pollack, J. Peischl) | WAS (D.R. Blake) | DACOM (G.S. Diskin, G. Sachse) |
| GoAmazon | HR-ToF-AMS (J.E. Shilling) | | PTR-MS (J.E. Shilling) | Los Gatos ICOS Analyzer (S.R. Springston) |
| FRAPPE | C-ToF-mAMS (R. Bahreini) | NCAR 2-channel Chemiluminescence (A.J. Weinheimer, D.D. Montzka) | TOGA (E. Apel, R. Hornbrook) | Aero-Laser VUV Fluorescence (T.L. Campos and F.M. Flocke) |
| KORUS-AQ | HR-AMS (J.L. Jimenez, P. Campuzano-Jost) | NCAR 4-channel Chemiluminescence (A.J. Weinheimer, D.D. Montzka) | PTR-MS (P. Eichler, L. Kaser, T. Mikoviny, M. Müller, A. Wisthaler) | DACOM (G.S. Diskin, S.E. Pusede) |
| ATom | HR-ToF-AMS (J.L. Jimenez) | NOAA NO$_y$O$_3$ (T.B. Ryerson, J. Peischl, C. Thompson) | TOGA (E. Apel, R. Hornbrook) | QCLS (B.C. Daube, S.C. Wofsy, R. Commane, E. Kort) |

**Table S2**. An overview of the instrumentation and associated primary investigators for the organic aerosol and trace gas observations used in this analysis.

Nitrogen oxides were measured using photolysis rates and NO/O$_3$ chemiluminescence techniques (Ryerson et al., 2000), carbon monoxide levels were measured using a Differential Absorption Carbon monOxide Measurement (DACOM) instrument (Sachse et al., 1987) or a VUV resonance fluorescence approach (Gerbig et al., 1999), isoprene concentrations were observed using a Proton Transfer Reaction Mass Spectrometer (de Gouw and Warneke, 2007), a Trace Organic Gas Analyzer (Apel et al., 2010) or a whole air sampling approach (Colman et al., 2001) and sulfate aerosol loadings were measured using an AMS.

| Regime | Description | Percentage of Dataset | Mean OA | Median OA | Std. Dev. OA | Mean Isoprene | Mean NO$_x$ | Mean CO |
|--------|-------------|----------------------|---------|-----------|--------------|---------------|--------------|---------|
| A | Dominant anthropogenic Influence | 39.1% | 1.9 | 0.6 | 3.2 | 0.05 | 0.96 | 144 |
| F | Dominant pyrogenic Influence | 7.3% | 4.5 | 2.7 | 5.3 | 0.13 | 0.17 | 151 |
| B | Dominant biogenic Influence | 3.6% | 3.1 | 2.6 | 2.6 | 1.46 | 0.16 | 122 |
| AF | Anthropogenic and Pyrogenic Influence | 6.8% | 3.8 | 1.6 | 5.0 | 0.05 | 0.80 | 160 |
| AB | Anthropogenic and Biogenic Influence | 14.0% | 4.1 | 2.7 | 4.0 | 0.60 | 0.35 | 115 |
| AFB | No dominant influence from any one source category. | 10.1% | 3.2 | 2.5 | 2.9 | 0.10 | 0.38 | 115 |
| R | Remote / Clean (concentrations under 0.2 µg / sm³) | 19.1% | 0.1 | 0.1 | 0.3 | 0.05 | 0.08 | 71 |
| Aggregate | | | 2.4 | 0.7 | 3.6 | 0.24 | 0.55 | 126 |

**Table S3.** An overview of the different regimes. Statistics (mean, median, standard deviation) are listed for the observational data categorized into the individual regimes. OA data is in units of µg sm$^{-3}$. Mean observations for isoprene, nitrogen oxides and carbon monoxide are in units of parts per billion (ppb).

[Figure]

**Figure S1.** Distribution in the observed organic aerosol concentrations for each campaign. The boxes denote the 25[th] and 75[th] percentile of the distribution, while the whiskers denote the 5[th] and 95[th] percentile. Observations represented here have been filtered and averaged to the model timestep. The bars are colored by campaign. Refer to Section 3 for more details.

[Figure]

**Figure S2.** Flight tracks colored by regime type and differentiated by altitude. The Regimes are as follows – Anthropogenic (A), Pyrogenic (F), Biogenic (B), Anthropogenic + Pyrogenic (AF), Anthropogenic + Biogenic (AB), Mixed (AFB) and Remote / Marine (R). Refer to Sect. 3 for details on model sampling and averaging.

[Figure]

[Figure]

**Figure S32**. Mean vertical profiles (in km) for the observed and simulated OA and sulfate across the different regimes. The profiles are binned at 200m intervals. Observations are in black. For the OA, the complex scheme is in dark greenred while the simple scheme is in light greenblue. Model sulfate is in redpurple.

[Figure]

**Figure S4**. A comparison of the simulated OA loadings averaged by grid-box over the vertical dimension. Panel (d) provides an overview of the column-averaged 'best fit' scheme based on the ability to minimize the mean bias.

[Figure]

**Figure S5.** Distribution in the ratio and bias between the observed and modelled organic aerosol concentrations for each model scheme across the 17 campaigns. The boxes denote the 25[th] and 75[th] percentile of the distribution, while the whiskers denote the 5[th] and 95[th] percentile. The ratio plots have been overlaid with violin plots describing the entire distribution. The box and ratio plots are colored by campaign.

[Figure]

[Figure]

**Figure S65.** Comparison of complex (dark green), simple (light green) and observed (grey) organic aerosol to carbon monoxide.

[Figure]

[Figure]

**Figure S76**. A comparison of model-observation OA bias and binned observations for a) relative humidity, b) Temperature, c) Sulfate, d) Isoprene, e) CO and f) NOx for the complex (left panels – dark greenred) and simple (right panels – light greenblue) schemes across the aggregate dataset. The best fit line is shown in black.

[Figure]

**Figure S87.** A statistical evaluation of the OA model skill for the complex and simple schemes against a modified treatment that interchanges the POA and SOA from both schemes.